# KMT2A associates with PHF5A-PHF14-HMG20A-RAI1 subcomplex in pancreatic cancer stem cells and epigenetically regulates their characteristics

Mai Abdel Mouti [1], Siwei Deng [1], Martin Pook[1,2], Jessica Malzahn[1], Aniko Rendek[3], Stefania Militi[1], Reshma Nibhani [1], Zahir Soonawalla[4], Udo Oppermann[1], Chang-il Hwang [5] & Siim Pauklin [1]✉

Pancreatic cancer (PC), one of the most aggressive and life-threatening human malignancies, is known for its resistance to cytotoxic therapies. This is increasingly ascribed to the subpopulation of undifferentiated cells, known as pancreatic cancer stem cells (PCSCs), which display greater evolutionary fitness than other tumor cells to evade the cytotoxic effects of chemotherapy. PCSCs are crucial for tumor relapse as they possess 'stem cell-like' features that are characterized by self-renewal and differentiation. However, the molecular mechanisms that maintain the unique characteristics of PCSCs are poorly understood. Here, we identify the histone methyltransferase KMT2A as a physical binding partner of an RNA polymerase-associated PHF5A-PHF14-HMG20A-RAI1 protein subcomplex and an epigenetic regulator of PCSC properties and functions. Targeting the protein subcomplex in PCSCs with a KMT2A-WDR5 inhibitor attenuates their self-renewal capacity, cell viability, and in vivo tumorigenicity.

Pancreatic cancer (PC) remains one of the most resilient and inevitably fatal malignancies, currently ranking third on the list of leading causes of death among men and women[1]. Pancreatic ductal adenocarcinoma (PDAC) is the most common type of PC, accounting for more than 90% of all pancreatic malignancies[2]. In contrast, pancreatic adenosquamous carcinoma (PASC) is a rare pathological subtype of PC, which accounts for only 0.5–4% of cases. It is often difficult to differentiate PASC from PDAC, since both display the same clinical manifestations, including weight loss, abdominal pain, and jaundice. Currently, both neoplasms share similar treatments, but patients diagnosed with PASC tend to have a worse survival rate than those with PDAC[3].

PC is undeniably one of the most notorious cancers that are highly resistant to cytotoxic chemotherapy; often associated with an initial response to treatment that is rapidly followed by drug resistance and tumor recurrence[4]. Accumulating evidence suggests that therapy resistance in PC is largely driven by pancreatic cancer stem cells (PCSCs), also known as tumor-initiating cells, which are evolutionary fitter than other tumor cells to evade the cytotoxic effects of chemotherapy[5]. Despite accounting for only a small fraction of the total tumor cell population, those cells play a significant role in tumor relapse and metastatic growth due to their unique stem cell-like properties of self-renewal and differentiation which allow them to recapitulate the heterogeneity of the original tumor[6]. However, what molecular mechanisms maintain the distinctive characteristics of PCSCs are currently poorly understood.

[1]Botnar Research Centre, Nuffield Department of Orthopaedics, Rheumatology, and Musculoskeletal Sciences, University of Oxford, Oxford, UK. [2]Institute of Biomedicine and Translational Medicine, Faculty of Medicine, University of Tartu, Tartu, Estonia. [3]Department of Histopathology, Oxford University Hospitals NHS Foundation Trust, Oxford, UK. [4]Department of Hepatobiliary and Pancreatic Surgery, Oxford University Hospitals NHS, Oxford, UK. [5]Department of Microbiology and Molecular Genetics, University of California Davis, Davis, USA. ✉e-mail: siim.pauklin@ndorms.ox.ac.uk

The plant homeodomain (PHD) finger domain protein 5A (PHF5A), a member of the PHD-finger superfamily, is known for its important role in the epigenetic regulation of stem cell maintenance and pluripotency in glioblastoma multiforme stem cells (GSCs)[7], human embryonic stem cells (hESCs)[8], and PCSCs[9]. However, the epigenetic mechanisms by which PHF5A regulates PCSC properties and functions have not been adequately explored.

In this study, we investigate the protein interactome of PHF5A in PCSCs derived from PDAC and PASC. Our study employs two parallel strategies: First, proteomic identification of transcriptional coregulators that bind to nuclear PHF5A in PCSCs using liquid chromatography coupled with tandem mass spectrometry (LC-MS/MS) and the search tool for retrieval of interacting genes/proteins (STRING) proteomic database[10]. This reveals a physical association between PHF5A and retinoic acid-induced 1 (RAI1) complex subunits, including PHF14, high mobility group 20A (HMG20A), and RAI1. Second, a small molecule compound screening targeting epigenetic modulating enzymes to identify epigenetic regulators of PCSC properties and functions. This identifies lysine methyltransferase 2A (KMT2A) as an epigenetic regulator of PCSC stemness. Furthermore, our interactome analysis shows that KMT2A physically associates with PHF5A, PHF14, HMG20A, and RAI1 to form an RNA polymerase II (RNA Pol II)-associated protein subcomplex, referred to as PHF5A subcomplex, that is specific to the CSC population of PC cells. We combine our proteomics data with chromatin immunoprecipitation sequencing (ChIP-seq) analysis of PHF5A subcomplex subunits and analysis of the transcriptional changes by RNA-sequencing (RNA-seq) upon the functional impairment of the enzymatic activity of KMT2A in PCSCs. Collectively, we demonstrate here a regulatory transcriptional mechanism of PCSC properties and functions that could be therapeutically targeted to attenuate their self-renewal capacity, cell viability, and tumorigenicity.

## Results

### Optimizing the platform for characterizing regulatory transcriptional complexes in PCSCs

To set up an optimal system for characterizing regulatory transcriptional complexes in PCSCs by proteomic, genomic, and transcriptomic methods, we first examined the enrichment of CSC surface markers in pancreatic tumorspheres to identify the CSC population of cells. Here, a 3-dimensional (3D) in vitro culturing technique, also known as the sphere-forming assay[11], of monolayer PC cells was used to identify PCSCs by flow cytometry (see Methods). As illustrated in Fig. 1a, monolayer PC cells derived from PASC (L3.6pl and L3.6sl) and PDAC (A13A and A13B) were cultured under CSC-enriching culture conditions and analyzed by flow cytometry for enriched CSC surface markers using a panel of fluorochrome-conjugated antibodies against putative PCSC surface markers. These included the following: ATP binding cassette subfamily G member 2 (ABCG2), epithelial cell adhesion molecule (EPCAM), CD44, CD24, prominin 1 (PROM1), stage-specific embryonic antigen-4 (SSEA4), and C-X-C motif chemokine receptor 4 (CXCR4)[12]. The gating strategy for the identification of enriched CSC surface markers is shown in Supplementary Fig. S1a.

Screening of those CSC surface markers by flow cytometry revealed a significant enrichment of ABCG2 in day 6 L3.6pl (Fig. 1b; Supplementary Fig. S1b), L3.6sl (Fig. 1c; Supplementary Fig. S1c), and A13B (Fig. 1e) tumorspheres, in addition to SSEA4 in day 6 A13A (Fig. 1d; Supplementary Fig. S1d) and A13B (Fig. 1e; Supplementary Fig. S1e) tumorspheres as compared to their respective parental monolayer cultures. We also observed that only a few A13A and A13B monolayer cells express CXCR4, but CXCR4-expressing cells were significantly enriched in day 6 A13A (Fig. 1d) and A13B (Fig. 1e) tumorspheres. Brightfield images of day 6 anchorage-independent L3.6pl, L3.6sl, A13A, and A13B tumorspheres are shown in Fig. 1f–i, respectively.

A13B is derived from a primary PDAC tumor from which A13A cells metastasized locally in the pancreas[13], whereas L3.6pl (pancreas-liver) and L3.6sl (spleen-liver) are metastatic variants of FG cells, PC cells derived from metastatic pancreatic adenocarcinoma[14], which are obtained after 3 rounds of in vivo selection in athymic nude mice[15]. This might explain our observation of the slower proliferation rate and smaller size of A13B tumorspheres (Fig. 1i) as compared to other tumorspheres generated from cell lines with a metastatic phenotype (Fig. 1f–h) which often selects for more aggressive and fast-growing cell variants in a parental tumor cell population.

Based on our flow cytometry screening data, PC cells were phenotypically sorted using a magnetic-activated cell sorting (MACS) technique that employs magnetic microbeads conjugated to monoclonal antibodies against relevant CSC surface markers. Following MACS, sorted cells were grown in a 3D culture under CSC-enriching culture conditions and analyzed by flow cytometry to examine the efficiency of cell sorting and enrichment of the CSC population. Representative examples of flow cytometry analysis of MACS-sorted cells (Fig. 1j–m) demonstrate the enrichment of ABCG2-expressing cells in ABCG2-enriched (ABCG2+) L3.6pl (Fig. 1j) and ABCG2+ L3.6sl (Fig. 1k) tumorspheres and SSEA4-expressing cells in SSEA4-enriched (SSEA4+) A13A (Fig. 1l) and SSEA4+ A13B (Fig. 1m) tumorspheres.

Using quantitative real-time polymerase chain reaction (qRT-PCR) analysis, we assessed the expression of self-renewal and pluripotency genes, including SRY-Box Transcription Factor 2 (SOX2), NANOG, Krüppel-like factor 4 (KLF4), signal transducer and activator of transcription 3 (STAT3), and POU domain, class 5, transcription factor 1 (POU5F1) in CSC-enriched pancreatic tumorspheres using the human pancreatic duct epithelial cell line HPDE6c7 that possesses stem cell-like characteristics[16] as a control. In all MACS-sorted pancreatic tumorspheres, SOX2 was significantly upregulated as compared to HPDE6c7 (Fig. 1n–q). We also observed a significant increase in the expression of NANOG in ABCG2+ L3.6pl (Fig. 1n) and ABCG2+ L3.6sl (Fig. 1o) tumorspheres, in addition to KLF4 in ABCG2+ L3.6sl (Fig. 1o), SSEA4+ A13A (Fig. 1p), and SSEA4+ A13B (Fig. 1q) tumorspheres, and STAT3 in ABCG2+ L3.6pl tumorspheres (Fig. 1n) as compared to HPDE6c7.

To verify our gene expression data, we analyzed SOX2 protein expression levels in CSC-enriched pancreatic tumorspheres using flow cytometry, since SOX2 is the self-renewal and pluripotency-associated gene that shows the most substantial variance (significant increase) in all MACS-sorted pancreatic tumorspheres as compared to HPDE6c7. This revealed a significant increase in the mean fluorescence intensity (MFI) of SOX2, a readout of SOX2 protein expression levels, in all CSC-enriched pancreatic tumorspheres versus HPDE6c7 (Fig. 1r). Representative flow cytometry histograms of SOX2 protein expression in ABCG2+ L3.6pl, ABCG2+ L3.6sl, SSEA4+ A13A, and SSEA4+ A13B tumorspheres versus HPDE6c7 are shown in Fig. 1s–v, respectively.

Additionally, we compared the expression levels of self-renewal and pluripotency genes in CSC-enriched pancreatic tumorspheres to their respective CSC-depleted monolayer cells sorted by MACS (Supplementary Fig. S1f). Our data show a significant increase in the expression of SOX2, NANOG, KLF4, STAT3, and POU5F1 in ABCG2+ L3.6pl (Supplementary Fig. S1g) and ABCG2+ L3.6sl (Supplementary Fig. S1h) tumorspheres as compared to their respective ABCG2-depleted (ABCG2-) monolayer cells, in addition to a significant increase in the expression of POU5F1 in SSEA4+ A13A tumorspheres (Supplementary Fig. S1i) and KLF4 in SSEA4+ A13B (Supplementary Fig. S1j) tumorspheres as compared to their respective SSEA4-depleted (SSEA4-) monolayer cells.

Collectively, these experiments for identifying and isolating the CSC population of PC cells followed by their validation as stem cell-like cells enabled us to verify that this system is suitable for downstream large-scale proteomic experiments for analyzing protein-protein interactions (PPIs) in PCSCs.

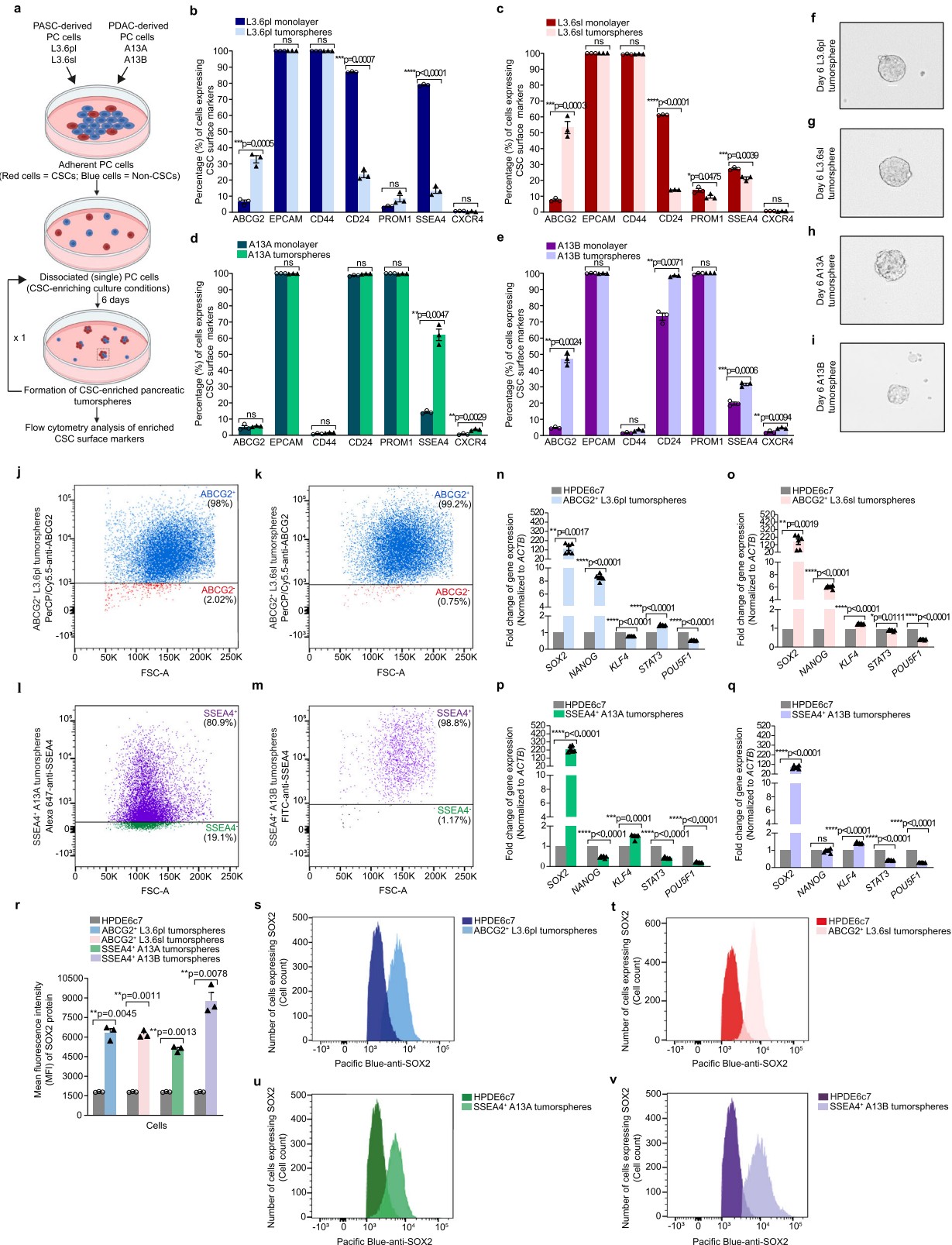

## PHF14 is a nuclear binding partner of PHF5A in PCSCs

SMAD2/3 transcription factors play a key role in regulating the tumorigenic properties and functions of PC cells[17]. Our previous research investigated the molecular mechanisms by which SMAD2/3 regulate PC through LC-MS/MS analysis of SMAD2/3 co-immunoprecipitates (Co-IPs) from nuclear protein extracts of L3.6pl

monolayer cells treated with inhibin subunit beta A (INHBA) at a concentration of 100 ng/ml for 24 h (Fig. 2a). This identified PHF5A as a nuclear SMAD2/3 binding partner in INHBA-treated L3.6pl monolayer cells, as demonstrated by the mass spectrum of a single tryptic PHF5A peptide (highlighted in the protein sequence coverage data) in a SMAD2/3 Co-IP sample (Fig. 2b). Western blot analysis of nuclear

**Fig. 1 | Identification, selective enrichment, and characterization of PCSCs.** **a** Schematic diagram depicting the sphere-forming assay for analyzing enriched CSC surface markers in pancreatic tumorspheres by flow cytometry. The schematic diagram was created with Biorender scientific illustration software. **b**–**e** Flow cytometry analysis of enriched CSC surface markers in day 6 L3.6pl (**b**), L3.6sl (**c**), A13A (**d**), and A13B (**e**) tumorspheres versus their respective parental monolayer cultures. Data are presented as the mean value ± standard error of the mean (SEM; $n = 3$ biologically independent experiments). $P$ values were calculated using a two-tailed $t$-test with Welch's correction for unequal variances. **f**–**i** Brightfield images of day 6 L3.6pl (**f**), L3.6sl (**g**), A13A (**h**), and A13B (**i**) tumorspheres. Data are representative of 3 biologically independent experiments with similar results. Scale bar, 50 μm. **j**, **k** Flow cytometry analysis of ABCG2 enrichment in MACS-sorted ABCG2$^+$ L3.6pl (**j**) and ABCG2$^+$ L3.6sl (**k**) tumorspheres. **l**, **m** Flow cytometry analysis of SSEA4 enrichment in MACS-sorted SSEA4$^+$ A13A (**l**) and SSEA4$^+$ A13B (**m**) tumorspheres.

**n**–**q** qRT-PCR analysis of self-renewal and pluripotency genes in ABCG2$^+$ L3.6pl (**n**), ABCG2$^+$ L3.6sl (**o**), SSEA4$^+$ A13A (**p**), and SSEA4$^+$ A13B (**q**) tumorspheres versus HPDE6c7. *ACTB* was used for the normalization of mRNA expression levels. Data are representative of 3 biologically independent experiments (6 technical replicates per biological replicate) and presented as the mean value ± SEM. $P$ values were calculated using a two-tailed $t$-test with Welch's correction for unequal variances. **r** Mean fluorescence intensity of SOX2 protein expression in ABCG2$^+$ L3.6pl, ABCG2$^+$ L3.6sl, SSEA4$^+$ A13A, and SSEA4$^+$ A13B tumorspheres as compared to HPDE6c7. Data are presented as the mean value ± SEM ($n = 3$ biologically independent experiments). Statistical analysis was performed using a two-tailed $t$ test with Welch's correction for unequal variances. **s**–**v** Flow cytometry histograms of intracellular SOX2 protein expression in ABCG2$^+$ L3.6pl (**s**), ABCG2$^+$ L3.6sl (**t**), SSEA4$^+$ A13A (**u**), and SSEA4$^+$ A13B (**v**) tumorspheres versus HPDE6c7. Data are representative of 3 biologically independent experiments.

PHF5A Co-IPs in INHBA (100 ng/ml)-treated L3.6pl and L3.6sl monolayer cells verified the physical association between SMAD2/3 and PHF5A (Fig. 2c). However, SMAD2/3-PHF5A physical interaction was not observed in ABCG2$^+$ L3.6pl and ABCG2$^+$ L3.6sl tumorspheres treated with INHBA (100 ng/ml) for 24 h (Fig. 2d). These findings indicate that even within the same cancer cell type, CSCs and non-CSCs have different PHF5A PPIs, highlighting the importance of analyzing the differential transcriptional mechanisms of PHF5A in different cell types in order to accurately assess its biological activity and functional relevance in PC.

This study aims at identifying epigenetic regulators that bind to PHF5A in PCSCs. Thus, we first performed an LC-MS/MS analysis of PHF5A Co-IPs from nuclear protein extracts of ABCG2$^+$ L3.6pl tumorspheres to identify nuclear binding partners of PHF5A in PCSCs (Fig. 2e). Immunoprecipitation (IP) of PHF5A in 3 biological replicates of nuclear PHF5A Co-IP versus control IgG Co-IP in ABCG2$^+$ L3.6pl tumorspheres was first examined by western blotting prior to LC-MS/MS analysis (Fig. 2f) then verified by LC-MS/MS data. A representative mass spectrum of a single tryptic peptide of PHF5A (highlighted and inboxed in the PHF5A protein sequence coverage data) in 1 biological replicate of PHF5A Co-IP in ABCG2$^+$ L3.6pl tumorspheres is shown in Fig. 2g.

Our LC-MS/MS analysis identified a total of 52 potential binding partners of PHF5A in 3 biological replicates of nuclear PHF5A Co-IP in ABCG2$^+$ L3.6pl tumorspheres which were absent in the control IgG Co-IP sample. Based on normalized total spectral counts, PIBF1, AMOTL2, IRF2BPL, RBBP8, PON1, PHF14, C5, SLC4A1AP, CCDC97, and IRAK4 constituted the top 10 potential binding partners of PHF5A (Fig. 2h). In this study, we primarily focused on PHF14, a member of the PHD-finger protein family, which is encoded by a highly conserved gene located on the 19p13.2 chromosome. PHF14 shares with PHF5A the ability to recognize modified histone codes and assign biological consequences based on these modifications, indicating their crucial role in regulating cellular epigenetic changes[18], however, its exact biological functions remain largely unknown. In the 3 biological replicates of nuclear PHF5A Co-IP in ABCG2$^+$ L3.6pl tumorspheres, LC-MS/MS data supporting the physical interaction between PHF5A and PHF14 include normalized total spectral counts of PHF5A (Fig. 2i) and PHF14 (Fig. 2j). A representative peptide fragmentation mass spectrum of a single tryptic peptide of PHF14 protein (highlighted and inboxed in the PHF14 protein sequence coverage data) is shown in Fig. 2k. The physical association between PHF5A and PHF14 was verified by western blot analyses of nuclear Co-IPs of PHF5A (Fig. 2l) and PHF14 (Fig. 2m) in ABCG2$^+$ L3.6pl, ABCG2$^+$ L3.6sl, and SSEA4$^+$ A13A tumorspheres.

Furthermore, subcellular fractionation of protein extracts from SSEA4$^+$ A13B tumorspheres derived from primary PDAC (Fig. 2n) and SSEA4$^+$ A13A tumorspheres derived from metastatic PDAC (Fig. 2o) showed that both PHF5A and PHF14 are primarily localized in the nucleus, which is consistent with their epigenetic role(s) in regulating

cellular functions. The nuclear membranes of SSEA4$^+$ A13B tumorspheres and, to a lesser extent, SSEA4$^+$ A13A tumorspheres are also abundant for PHF5A.

Encouraged by our LC-MS/MS data, we simulated a docking between PHF5A and PHF14 proteins using HDOCK[19] (Supplementary Fig. S2a). This revealed a favored physical association between PHF5A and PHF14, with a putative binding energy of −288.23 kcal/mol which further validates our findings. A list of PHF5A-PHF14 interface residue pairs is shown in Supplementary Fig. S2b.

The physical association between PHF5A and PHF14 was then examined in monolayer cultures lacking CSC enrichment, including L3.6pl, L3.6sl, HPDE6c7, and the differentiated mouse embryonic fibroblasts NIH-3T3 by western blot analysis of nuclear PHF5A Co-IPs in those cells (Supplementary Fig. S2c). As a result, we observed that PHF5A physically associates with PHF14 in monolayer cultures of L3.6pl, L3.6sl (Supplementary Fig. S2d), and HPDE6c7 (Supplementary Fig. S2e). According to these findings, we conclude that the physical association between PHF5A and PHF14 is not specific to PCSCs. In NIH-3T3 cells, however, PHF5A does not bind to PHF14 (Supplementary Fig. S2f), suggesting that the physical association between the two proteins is cell type-dependent to regulate specific cellular properties and functions.

Next, PHF5A and PHF14 ChIP-seq analyses were performed in ABCG2$^+$ L3.6pl tumorspheres to map the occupancy of PHF5A and PHF14 across the genome and identify their target DNA binding sites (Fig. 3a). This revealed a total of 200 genes occupying common genomic binding sites between PHF5A and PHF14 in 2 biological replicates of ABCG2$^+$ L3.6pl tumorspheres per each ChIP-seq experiment (Fig. 3b) which further supports the physical link between the 2 proteins; while hinting at their potential functional cooperation. Genomic distribution analysis of common DNA binding sites occupied by PHF5A and PHF14 showed that 63.9% of those target sites are situated within the gene body, compared to 25% at the promoter region and 8.3% downstream of genes (Fig. 3c).

Our pathway enrichment analysis of genes cooccupied by PHF5A and PHF14 in ABCG2$^+$ L3.6pl tumorspheres revealed the over-representation of pathways involved in the following: regulation of self-renewal and pluripotency of stem cells (Wnt signaling pathway), activation of HOX genes during differentiation, regulation of transcription by RUNX1 during the differentiation of hematopoietic stem cells, chromatin remodeling, epigenetic regulation of gene expression, in addition to pathways involved in the regulation of key functions that are required for stem cell maintenance, such as telomere maintenance, packaging of telomere ends, and cellular response to stress (Fig. 3d). Representative ChIP-seq tracks of *PAK3* (Fig. 3e), *FLT4* (Fig. 3f), *LINGO2* (Fig. 3g), and *COL6A1* (Fig. 3h) genes demonstrate PHF5A and PHF14-enriched regions (peaks) at different genomic locations in PHF5A and PHF14 ChIP-seq samples versus input control of ABCG2$^+$ L3.6pl tumorspheres and highlight the common target sites occupied by PHF5A and PHF14 in ChIP-seq samples.

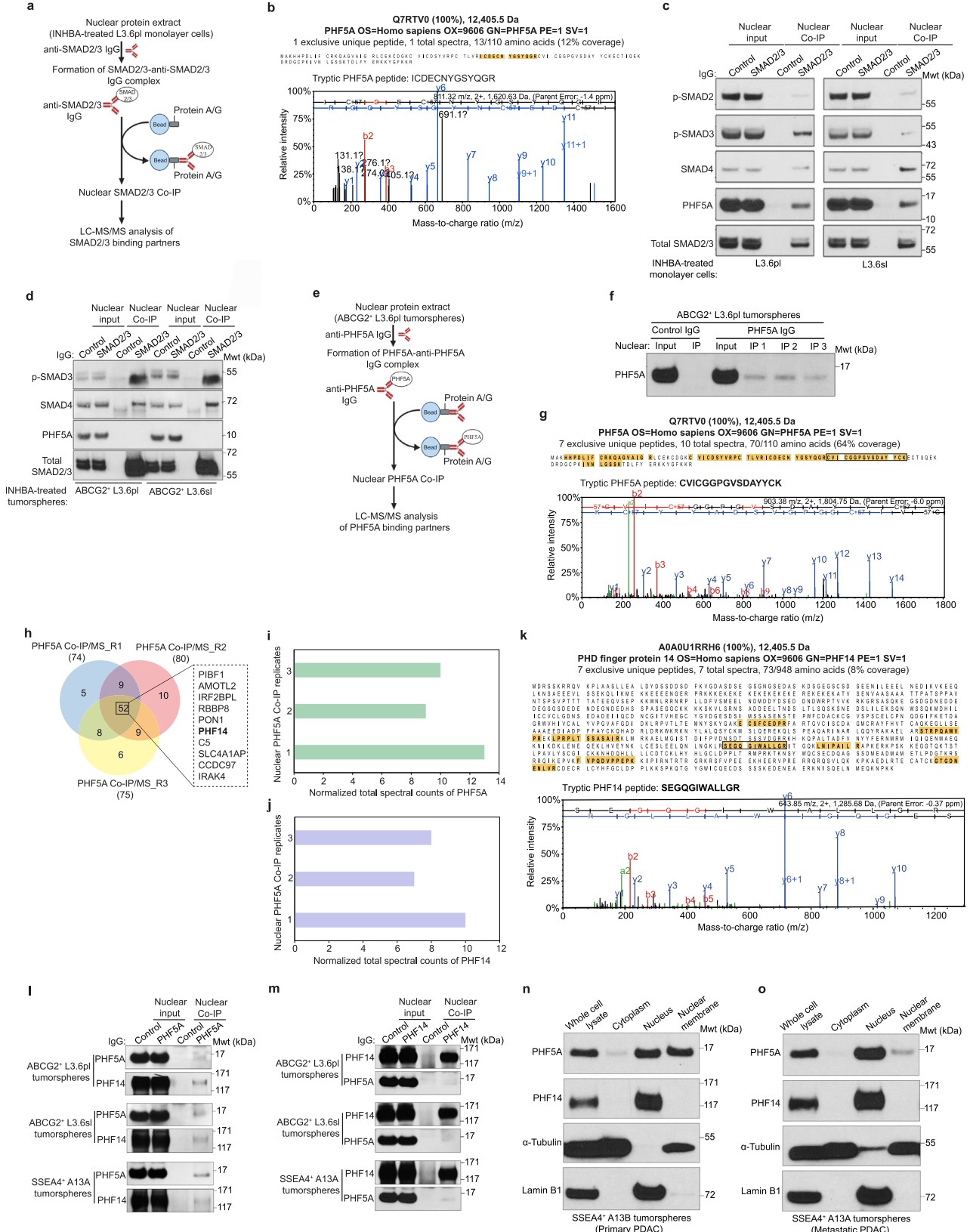

Overall, our proteomics, molecular docking, and genomics data support the physical association between PHF5A and PHF14 in PCSCs, and suggest their cooperation in regulating cellular functions through their cooccupancy of binding sites within or near transcription regulatory regions.

## LC-MS/MS-based proteomic analysis reveals the physical association between PHF5A and members of the RAI1 complex in PCSCs

The PHF5A-PHF14 protein subcomplex is likely to cooperate with other transcriptional regulatory proteins in the nucleus. To identify other

**Fig. 2 | Protein interactome analysis of PHF5A in PCSCs. a** Schematic illustration of nuclear SMAD2/3 Co-IP in L3.6pl monolayer cells treated with INHBA (100 ng/ml) for 24 h for LC-MS/MS analysis of SMAD2/3 binding partners. The schematic illustration was created with Biorender scientific illustration software. **b** Representative mass spectrum of a single tryptic peptide of PHF5A (highlighted in PHF5A protein sequence coverage data) detected upon LC-MS/MS analysis of nuclear SMAD2/3 Co-IP in INHBA (100 ng/ml)-treated L3.6pl monolayer cells ($n = 2$ biologically independent experiments) for 24 h. The mass spectrum demonstrates the relative intensity of fragment ions separated in a mass spectrometer versus their mass-to-charge ratio (m/z). Protein identifications were accepted if they could be established at greater than 99.0% probability and contained at least 2 identified peptides. **c** Western blot analysis of SMAD2/3 versus control IgG Co-IP from nuclear protein extracts of L3.6pl and L3.6sl monolayer cells treated with INHBA (100 ng/ml) for 24 h. Data are representative of 2 biologically independent experiments with similar results. **d** Western blot analysis of SMAD2/3 versus control IgG Co-IP from nuclear protein extracts of ABCG2⁺ L3.6pl and ABCG2⁺ L3.6sl tumorspheres treated with INHBA (100 ng/ml) for 24 h. Data are representative of 2 biologically independent experiments with similar results. **e** Schematic illustration of PHF5A Co-IP from nuclear protein extracts of ABCG2⁺ L3.6pl tumorspheres for LC-MS/MS analysis of PHF5A nuclear binding partners. The schematic illustration was created with Biorender scientific illustration software. **f** Western blot analysis of PHF5A IP from 3 biological replicates of PHF5A versus control IgG Co-IP from nuclear protein extracts of ABCG2⁺ L3.6pl tumorspheres. **g** Representative mass spectrum of a single tryptic peptide of PHF5A (highlighted and inboxed in the PHF5A protein sequence coverage data) detected upon LC-MS/MS analysis of PHF5A Co-IP from nuclear protein extracts of ABCG2⁺ L3.6pl tumorspheres ($n = 3$ biologically independent experiments). The mass spectrum demonstrates the relative intensity of fragment ions separated in a mass spectrometer versus their m/z). Protein

identifications were accepted if they could be established at greater than 99.0% probability and contained at least 2 identified peptides. **h** Venn diagram demonstrating the total number of potential nuclear binding partners of PHF5A detected upon LC-MS/MS analysis of 3 biological replicates of PHF5A versus control IgG Co-IP from nuclear protein extracts of ABCG2⁺ L3.6pl tumorspheres. Based on normalized total spectral counts, the top 10 candidates in 3 biological replicates are inboxed, including PHF14 (highlighted in bold). **i, j** Bar charts demonstrating the normalized total spectral counts of PHF5A (**i**) and PHF14 (**j**) peptides detected upon LC-MS/MS analysis of 3 biological replicates of nuclear PHF5A versus control IgG Co-IP from nuclear protein extracts of ABCG2⁺ L3.6pl tumorspheres.
**k** Representative mass spectrum of a single tryptic peptide of PHF14 (highlighted and inboxed in the PHF14 protein sequence coverage data) detected upon LC-MS/MS analysis of nuclear PHF5A Co-IP from nuclear protein extracts of ABCG2⁺ L3.6pl tumorspheres ($n = 3$ biologically independent experiments). The mass spectrum demonstrates the relative intensity of fragment ions separated in a mass spectrometer versus their m/z. Protein identifications were accepted if they could be established at greater than 99.0% probability and contained at least 2 identified peptides. **l** Western blot analysis of nuclear Co-IPs of PHF5A versus control IgG in ABCG2⁺ L3.6pl, ABCG2⁺ L3.6sl, and SSEA4⁺ A13A tumorspheres. Data are representative of 2 biologically independent experiments with similar results. **m** Western blot analysis of nuclear Co-IPs of PHF14 versus control IgG in ABCG2⁺ L3.6pl, ABCG2⁺ L3.6sl, and SSEA4⁺ A13A tumorspheres. Data are representative of 2 biologically independent experiments with similar results. **n, o** Western blot analysis of subcellular localization of PHF5A and PHF14 in fractionated protein extracts (cytoplasm, nucleus, and nuclear membrane) from SSEA4⁺ A13B (**n**) and SSEA4⁺ A13A (**o**) tumorspheres. α-Tubulin and Lamin B1 were used as internal controls for cytoplasmic and nuclear fractions, respectively. Data are representative of 2 biologically independent experiments with similar results.

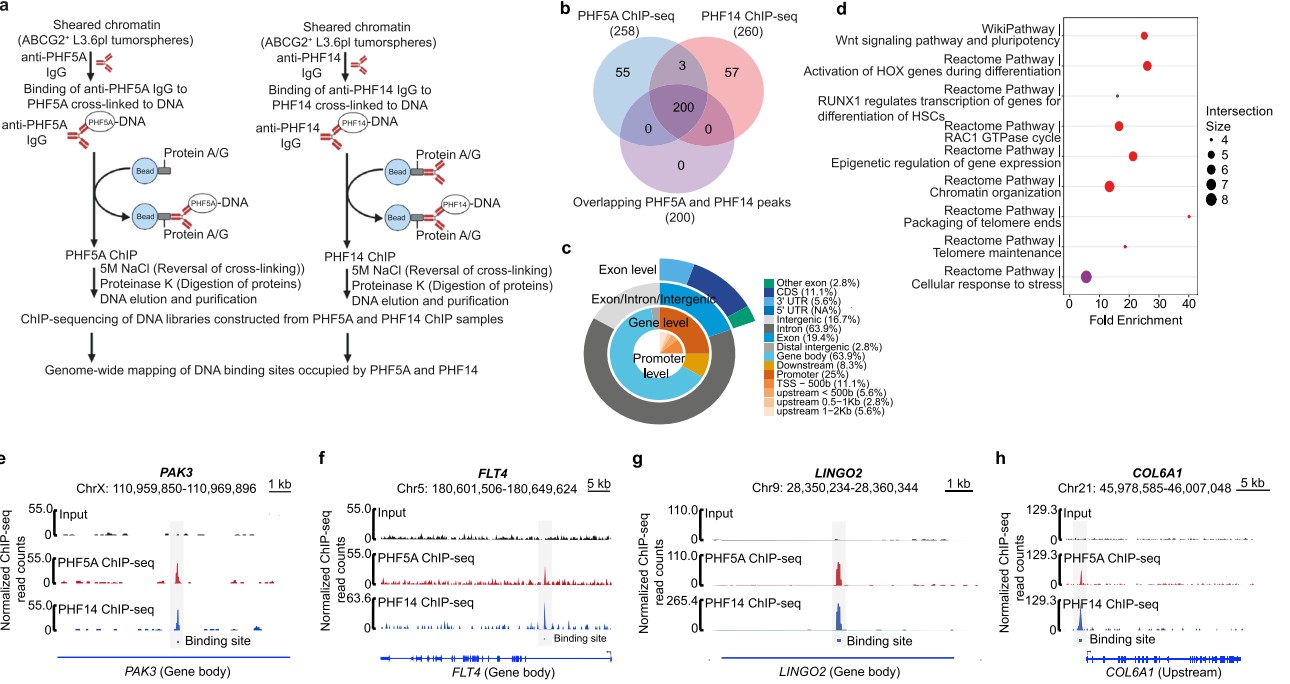

**Fig. 3 | PHF14 occupies common DNA binding sites with PHF5A in PCSCs. a** Schematic illustration of the workflow for ChIP-seq analysis of DNA binding sites occupied by PHF5A and PHF14 in ABCG2⁺ L3.6pl tumorspheres. The schematic illustration was created with Biorender scientific illustration softwar. **b** Venn diagram demonstrating the total number of genes occupied by PHF5A and PHF14 and genes with overlapping regions of peak enrichment for PHF5A and PHF14 in ABCG2⁺ L3.6pl tumorspheres. Data are representative of 2 biological replicates per each ChIP-seq experiment. **c** Pie chart illustrating the genomic distribution of common DNA binding sites occupied by PHF5A and PHF14 in 2 biological replicates

of ABCG2⁺ L3.6pl tumorspheres per each ChIP-seq experiment. **d** Dotplot pathway enrichment map illustrating the most significantly over-represented pathways (Wiki and Reactome) for genes cooccupied by PHF5A and PHF14 in biological duplicates of ABCG2⁺ L3.6pl tumorspheres per each ChIP-seq experiment. The size and color of the dots represent the number of genes enriched in the pathway of interest and the significance of enrichment, respectively. **e–h** Representative ChIP-seq tracks highlighting common DNA binding sites occupied by PHF5A (red) and PHF14 (blue) in *PAK3* (**e**), *FLT4* (**f**), *LINGO2* (**g**), and *COL6A1* (**h**) genes in ChIP-seq samples versus input control (black) of ABCG2⁺ L3.6pl tumorspheres.

members of the PHF5A-PHF14 protein subcomplex in PCSCs, we performed a physical subnetwork analysis of the proteins identified by our LC-MS/MS analysis as potential nuclear binding partners of PHF5A in ABCG2+ L3.6pl tumorspheres using the STRING proteomic database which integrates data from text mining and curated databases, along with experimental evidence to identify PPIs (Fig. 4a). The STRING network analysis revealed several PPIs among LC-MS/MS-identified nuclear binding partners of PHF5A (Fig. 4b). Here, we primarily focused on the physical association between PHF5A and components of the RAI1 complex[20], including PHF14, HMG20A, RAI1, and transcription factor 20 (TCF20). Normalized total spectral counts of LC-MS/MS-identified HMG20A, RAI1, and TCF20 peptides in 3 biological replicates of PHF5A Co-IP in ABCG2+ L3.6pl tumorspheres are displayed in Fig. 4c−e, respectively. In addition, we show representative peptide fragmentation mass spectra of a single tryptic peptide of each of HMG20A, RAI1, and TCF20 in Fig. 4f–h, respectively. Using western blot analysis, we verified the physical association between PHF5A and RAI1 complex members, including PHF14, HMG20A, and RAI1, but not TCF20 in ABCG2+ L3.6pl tumorspheres (Fig. 4i).

## Small molecule compound screening identifies KMT2A as an epigenetic regulator of PCSCs

The PHF5A-PHF14-HMG20A-RAI1 protein subcomplex on its own is unable to regulate gene expression, but requires the activity of epigenetic regulatory enzymes. To identify epigenetic regulators that could be important for the maintenance of PCSCs, we performed a small molecule compound screening targeting a broad range of epigenetic regulatory enzymes in PC cells. The chemical screening was performed in FG cells, representing a heterogeneous population of CSCs and non-CSCs, which allowed us to assess the effects of the chemical compounds on both subpopulations. Cells were treated in biological triplicate with DMSO (control) and each small molecule drug inhibitor for 5 days, followed by flow cytometry analysis of the surface expression of the CSC marker SSEA4 and assessment of the sphere-forming capacity and cell viability of PC cells (Fig. 5a). We used a library of 142 small molecule compounds verified to be active and targeting specific regulatory epigenetic modulating enzymes (Supplementary Table 1), such as bromodomain inhibitors, histone deacetylase (HDAC) inhibitors, histone methyltransferase (HMT) inhibitors, and compounds targeting methyllysine binders (Fig. 5b).

Using this large unbiased screening approach, we identified KMT2A as a potential epigenetic regulator of PCSCs, as OICR-9429 (Fig. 5c), the chemical probe for WDR5 that leads to KMT2A inhibition[21–23], significantly decreased the percentage (%) of cells expressing the CSC surface marker SSEA4 at a concentration of 10 μM as compared to DMSO (control; Fig. 5d). Next, we examined the effect of OICR-9429 (10 μM) on the self-renewal capacity of PCSCs. As shown in Fig. 5e, OICR-9429 significantly reduced tumorsphere formation by FG cells as compared to DMSO (control) treatment, which suggests a role for KMT2A in the regulation of cell stemness. Western blot analysis of H3K4me3 protein levels in ABCG2+ L3.6sl cells treated with different concentrations (0–50 μM) of OICR-9429 showed reduced H3K4me3 protein levels starting at 10 μM (Supplementary Fig. S3a), which verifies the impairment of the HMT activity of KMT2A.

Additionally, treatment of FG cells with different concentrations (0–200 μM) of OICR-9429 compound resulted in increased cell death in non-CSCs in a dose-dependent manner (Fig. 5f), suggesting that KMT2A may serve as a potential therapeutic target for both subpopulations of PC cells, especially since gene expression data derived from TCGA clinical datasets show a significant increase in *KMT2A* expression levels in PDAC patients as compared to the normal pancreas (Fig. 5g).

In order to validate the role of KMT2A in the maintenance of PCSCs, we tested the effects of OICR-9429 and MM-401[24], a small molecule inhibitor of KMT2A-WDR5 interaction, on the self-renewal capacity of SSEA4+ A13A cells. As shown in Supplementary Fig. S3b, tumorsphere formation was significantly impaired when SSEA4+ A13A cells were treated with 10 μM of each of OICR-9429 and MM-401 versus DMSO (control). We also examined the effects of 10 μM of each of OICR-9429 and MM-401 on the viability of SSEA4+ A13A cells which were pretreated with each compound for 5 days before staining with PE-Annexin V and 7-Aminoactinomycin D (7-AAD) for flow cytometry analysis. This revealed a significant decrease in the percentages (%) of viable and early apoptotic SSEA4+ A13A cells treated with either OICR-9429 or MM-401 as compared to DMSO (control; Supplementary Fig. S3c, lower panel) which is demonstrated by the lower percentages (%) of viable (Annexin^Lo/7AAD^Lo) and early apoptotic (Annexin^Hi/7AAD^Lo) SSEA4+ A13A cells treated with either OICR-929 or MM-401 as compared DMSO in the representative flow cytometry pseudocolor plots (Supplementary Fig. S3c, upper panel). The analysis also showed a significant increase in the percentages (%) of late apoptotic and necrotic SSEA4+ A13A cells treated with either OICR-9429 or MM-401 as compared to DMSO (control; Supplementary Fig. S3c, lower panel). These data are demonstrated by the higher percentages (%) of late apoptotic (Annexin^Hi/7AAD^Hi) and necrotic (Annexin^Lo/7AAD^Hi) SSEA4+ A13A cells treated with either OICR-9429 or MM-401 as compared DMSO (control) in the representative flow cytometry pseudocolor plots (Supplementary Fig. S3c, upper panel).

Collectively, our data from the small molecule compound screening in PC cells and in vitro cell-based functional assays involving the inhibition of the HMT activity of KMT2A in PCSCs indicate that KMT2A plays an important epigenetic role in regulating PCSC stemness and survival, which renders this epigenetic modulating enzyme an ideal candidate for further investigation.

## Interactome analysis reveals the physical association between KMT2A and PHF5A-PHF14-HMG20A-RAI1 protein subcomplex in PCSCs

RAI1 protein complex subunits were initially identified in murine brain, liver, and kidney tissues in the H3K4me3-repelled fraction of tissue eluates pre-filtered through columns containing non-methylated H3K4 and H3K4me3 peptides prior to MS analysis, indicating that the recognition of the RAI1 complex to methylated H3K4 requires the incorporation of an HMT enzyme[20]. Based on our findings from the small molecule compound screening and in vitro cell-based assays which identified KMT2A as an epigenetic regulator of PCSCs, in addition to our data from the pull-down assay of biotinylated modified histone peptides in PCSCs (Supplementary Figure S3d) demonstrating that PHF5A and PHF14 bind to biotinylated H3K4me2 and H3K4me3 peptides in ABCG2+ L3.6sl tumorspheres (Supplementary Figure S3e), we hypothesized that there could be functional cooperation via a physical association between PHF5A-PHF14-HMG20A-RAI1 protein subcomplex and KMT2A enzyme that would deposit methylated histone marks that are then recognized by protein subcomplex members, such as PHF5A and PHF14 (Fig. 5h), especially since earlier research in neurons has demonstrated a role for HMG20A in enhancing H3K4 trimethylation through KMT2A recruitment for activating neuronal-specific genes[25]. Western blot analysis of PHF5A Co-IP in nuclear protein extracts from ABCG2+ L3.6sl tumorspheres revealed a physical association between KMT2A and RNA Pol II-associated PHF5A-PHF14-HMG20A-RAI1 protein subcomplex (Fig. 5i). This finding was also observed in SSEA4+ A13A tumorspheres, but RAI1 did not appear to be a subunit of the protein subcomplex (Fig. 5j). A schematic diagram illustrating subunits of PHF5A-PHF14-HMG20A-RAI1-KMT2A protein subcomplex (PHF5A subcomplex) is shown in Fig. 5k. However, whether KMT2A is recruited through HMG20A or other members of the PHF5A subcomplex remains unknown and requires future investigation.

To further investigate the formation of PHF5A subcomplex in other PCSCs expressing distinct CSC surface markers, we isolated

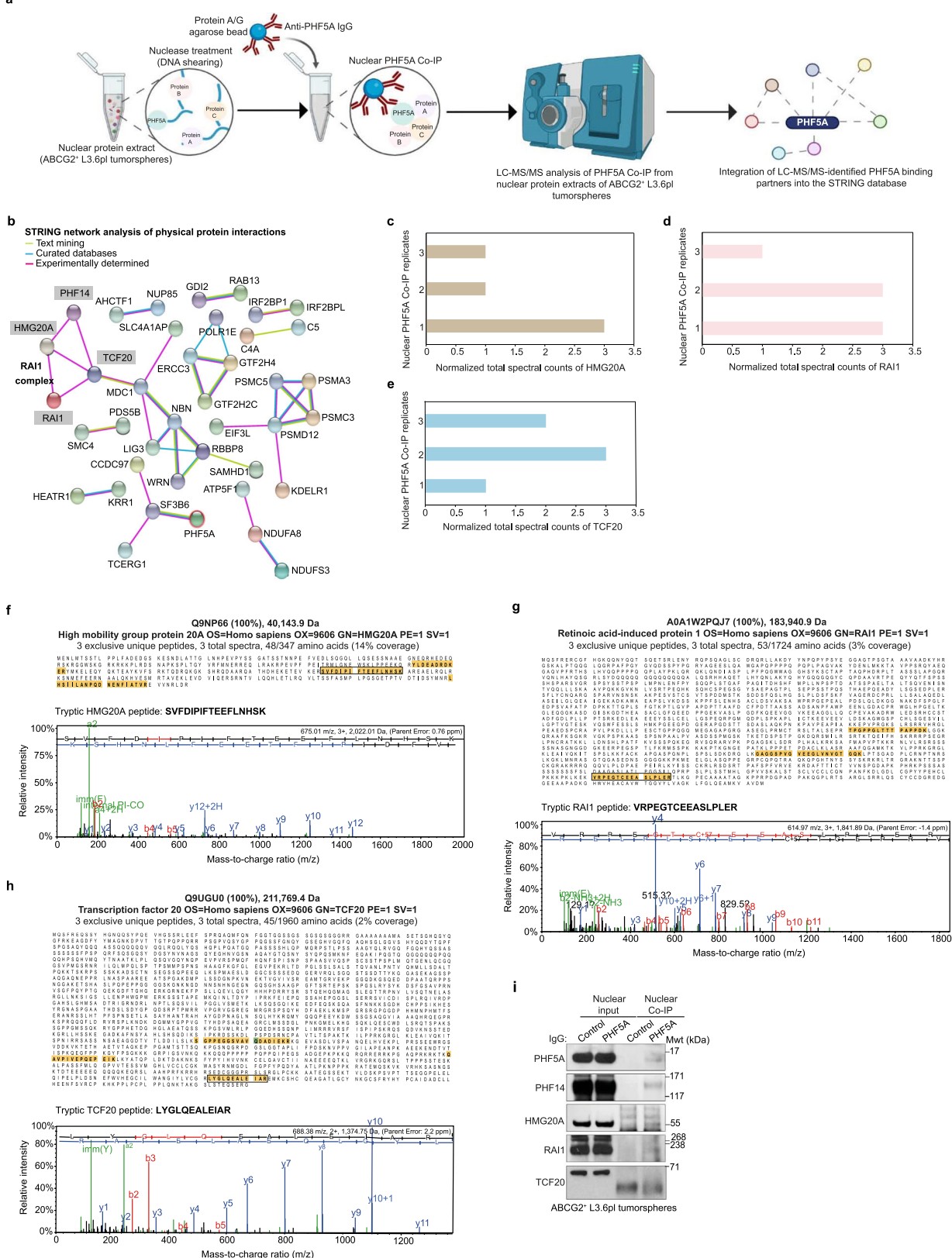

**a**

Nuclear protein extract (ABCG2⁺ L3.6pl tumorspheres) → Nuclease treatment (DNA shearing) → Protein A/G agarose bead, Anti-PHF5A IgG → Nuclear PHF5A Co-IP → LC-MS/MS analysis of PHF5A Co-IP from nuclear protein extracts of ABCG2⁺ L3.6pl tumorspheres → Integration of LC-MS/MS-identified PHF5A binding partners into the STRING database

**b** STRING network analysis of physical protein interactions
- Text mining
- Curated databases
- Experimentally determined

**c** Normalized total spectral counts of HMG20A

**d** Normalized total spectral counts of RAI1

**e** Normalized total spectral counts of TCF20

**f** Q9NP66 (100%), 40,143.9 Da
High mobility group protein 20A OS=Homo sapiens OX=9606 GN=HMG20A PE=1 SV=1
3 exclusive unique peptides, 3 total spectra, 48/347 amino acids (14% coverage)
Tryptic HMG20A peptide: SVFDIPIFTEEFLNHSK

**g** A0A1W2PQJ7 (100%), 183,940.9 Da
Retinoic acid-induced protein 1 OS=Homo sapiens OX=9606 GN=RAI1 PE=1 SV=1
3 exclusive unique peptides, 3 total spectra, 53/1724 amino acids (3% coverage)
Tryptic RAI1 peptide: VRPEGTCEEASLPLER

**h** Q9UGU0 (100%), 211,769.4 Da
Transcription factor 20 OS=Homo sapiens OX=9606 GN=TCF20 PE=1 SV=1
3 exclusive unique peptides, 3 total spectra, 45/1960 amino acids (2% coverage)
Tryptic TCF20 peptide: LYGLQEALEIAR

**i** ABCG2⁺ L3.6pl tumorspheres

CD44⁺ and CD133⁺ cells from their respective parental L3.6sl and A13A monolayer cultures, respectively, using MACS which was followed by nuclear PHF5A Co-IP and western blot analysis. Our data demonstrate the physical association between PHF5A, PHF14, HMG20A, RAI1, and KMT2A in CD44⁺ L3.6sl (Supplementary Fig. S3f) and CD133⁺ A13A

(Supplementary Fig. S3g) tumorspheres, providing further support for the role of the identified protein subcomplex in PCSCs.

We also assessed the specificity of the PHF5A protein subcomplex to PCSCs by performing a western blot analysis of KMT2A Co-IP in nuclear protein extracts from L3.6pl monolayer cells and ABCG2⁺

**Fig. 4 | Physical association between PHF5A and components of RAI1 complex in PCSCs. a** Schematic diagram illustrating the integration of LC-MS/MS-identified nuclear binding partners of PHF5A in ABCG2⁺ L3.6pl tumorspheres into the STRING network database. The schematic diagram was created with Biorender scientific illustration software. **b** Physical subnetwork analysis of LC-MS/MS-identified interacting partners of PHF5A in ABCG2⁺ L3.6pl tumorspheres using the STRING network database. Physical association data derived from text mining, curated databases, and experimental evidence are represented by light green, turquoise, and fuchsia lines, respectively. Highlighted proteins including PHF14, HMG20A, RAI1, and TCF20 are components of the RAI1 complex. **c**–**e** Bar charts demonstrating normalized total spectral counts of HMG20A (**c**), RAI1 (**d**), and TCF20 (**e**) peptides detected upon LC-MS/MS analysis of 3 biological replicates of PHF5A Co-

IP from nuclear protein extracts of ABCG2⁺ L3.6pl tumorspheres.
**f**–**h** Representative mass spectra of a single tryptic peptide of each of HMG20A (**f**), RAI1 (**g**), and TCF20 (**h**) (highlighted and inboxed in the respective protein sequence coverage data) detected upon LC-MS/MS analysis of PHF5A Co-IP from nuclear protein extracts of ABCG2⁺ L3.6pl tumorspheres ($n = 3$ biologically independent experiments). The mass spectrum demonstrates the relative intensity of fragment ions separated in a mass spectrometer versus their m/z. Protein identifications were accepted if they could be established at greater than 99.0% probability and contained at least 2 identified peptides. **I**, Western blot analysis of PHF5A versus control IgG Co-IP from nuclear protein extracts of ABCG2⁺ L3.6pl tumorspheres. Data are representative of 2 biologically independent experiments with similar results.

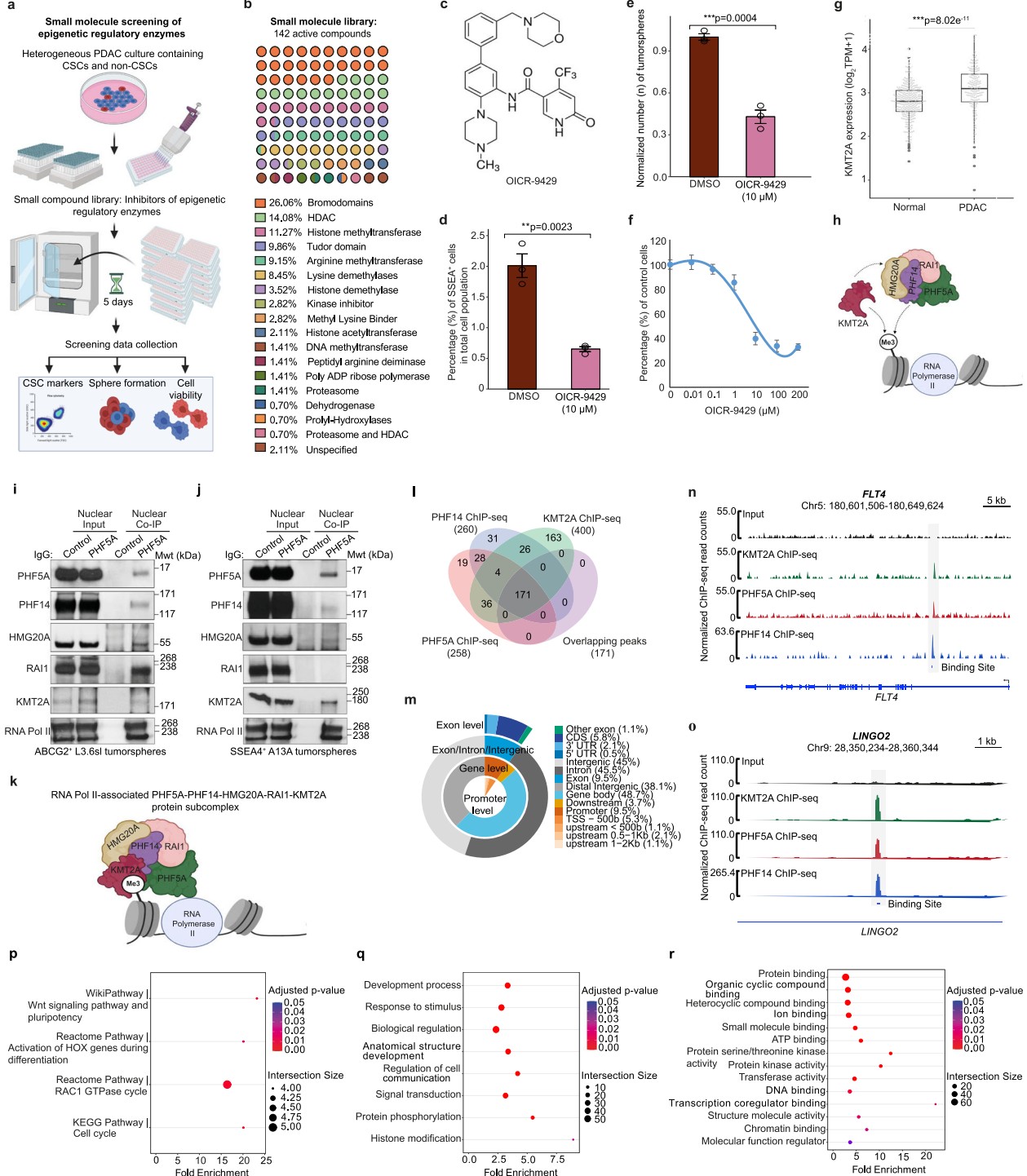

**Fig. 5 | KMT2A epigenetically regulates PC cells and physically associates with RNA Pol II-associated PHF5A-PHF14-HMG20A-RAI1 protein subcomplex in PCSCs. a** Schematic illustration of the small molecule compound screening of epigenetic regulatory enzymes in a heterogeneous population of FG cells. The schematic illustration was created with Biorender scientific illustration software. **b** Schematic illustration of the small molecule library including 142 active compounds targeting specific epigenetic modifying enzymes. The schematic illustration was created with Biorender scientific illustration software. **c** Chemical structure of OICR-9429. **d** Effect of OICR-9429 on the percentage (%) of SSEA4⁺ FG cells (CSCs) as compared to DMSO-treated (control) cells. Data are presented as the mean value ± SEM (*n* = 3 biologically independent experiments). Statistical analysis was performed using a two-tailed *t* test with Welch's correction for unequal variances. **e** Effect of OICR-9429 on tumorsphere formation in FG cells as compared to DMSO-treated (control) cells. Data are presented as the mean value ± SEM (*n* = 3 biologically independent experiments). Statistical analysis was performed using a two-tailed *t* test with Welch's correction for unequal variances. **f** Concentration curve analysis demonstrating the effect of OICR-9429 on the viability of FG cells (n = 3 biologically independent experiments). Data are presented as the mean value ± standard deviation of the mean (SD) and as the percentage (%) of control cells treated with DMSO. Statistical analysis was performed using a two-tailed *t* test with Welch's correction for unequal variances. **g** Box plot comparing gene expression levels of *KMT2A* in PDAC tumors versus normal pancreatic tissues. Data were obtained from TCGA PAAD (PDAC patients, *n* = 183) and GTEx (normal pancreas, *n* = 328) clinical datasets derived from the National Cancer Institute GDC data portal. Statistical analysis was performed using a two-tailed *t* test. Box plots show the median (center line), upper and lower quartiles (box), and range of the data

excluding outliers (whiskers). The upper and lower whiskers represent the maximum (non-outlier) and minimum (non-outlier) values, respectively. **h** Schematic illustration depicting the potential association of KMT2A with PHF5A-PHF14-HMG20A-RAI1 protein subcomplex in PCSCs. The schematic illustration was created with Biorender scientific illustration software. **i, j**, Western blot analysis of PHF5A versus control IgG Co-IP from nuclear protein extracts of ABCG2⁺ L3.6sl (**i**) and SSEA4⁺ A13A (**j**) tumorspheres. Data are representative of 2 biologically independent experiments with similar results. **k** Schematic illustration of the RNA Pol II-associated PHF5A protein subcomplex in PCSCs, including KMT2A, PHF5A, PHF14, HMG20A, and RAI1. The schematic illustration was created with Biorender scientific illustration software. **l** Venn diagram illustrating the total number of genes bound by KMT2A, PHF5A, and PHF14, in addition to overlapping DNA binding sites in KMT2A, PHF5A, and PHF14 ChIP-seq biological duplicates of ABCG2⁺ L3.6pl tumorspheres. **m** Pie chart illustrating the genomic distribution of target sites cooccupied by KMT2A, PHF5A, and PHF14 in ChIP-seq biological duplicates of ABCG2⁺ L3.6pl tumorspheres. **n, o** Representative ChIP-seq tracks demonstrating common DNA binding sites occupied by KMT2A, PHF5A, and PHF14 within *FLT4* (**n**) and *LINGO2* (**o**) genes in KMT2A, PHF5A, and PHF14 ChIP-seq samples versus input control of ABCG2⁺ L3.6pl tumorspheres (*n* = 2 biologically independent experiments per each ChIP-seq experiment). **p–r** Pathway (**p**), biological process (**q**), and molecular function (**r**) enrichment analyses of genes cooccupied by KMT2A, PHF5A, and PHF14 in ABCG2⁺ L3.6pl tumorspheres. Data are representative of biological duplicates per each ChIP-seq experiment. The size and color of the dots represent the number of genes enriched in the pathway (**p**), biological process (**q**), or molecular function (**r**) of interest and the significance of enrichment, respectively.

L3.6pl tumorspheres. In L3.6pl monolayer cells, KMT2A was found to only associate with HMG20A and WDR5 (Supplementary Fig. S3h). In ABCG2⁺ L3.6pl tumorspheres, however, we observed a physical association between KMT2A, PHF5A, PHF14, HMG20A, RAI1, and WDR5 (Supplementary Fig. S3i), indicating the specificity of the identified protein subcomplex to PCSCs.

Encouraged by our protein interactome data, we mapped the occupancy of KMT2A, PHF5A, and PHF14 across the genome in ABCG2⁺ L3.6pl tumorspheres by ChIP-seq analysis. The results revealed a total of 171 genes occupying common regions of peak enrichment in KMT2A, PHF5A, and PHF14 ChIP-seq samples of ABCG2⁺ L3.6pl tumorspheres (2 biological replicates per each ChIP-seq experiment; Fig. 5l). Among those, cancer stemness-associated genes were identified, including *LINGO2, PAK3, JADE3, MLLT3, SRPK1, HIST1H2BF, CTBP2, TARBP2, CREBBP, COL6A1*, and *FAM83F*. Genomic distribution analysis revealed that 48.7% of the cooccupied DNA-binding sites are located within the gene body, 38.1% are distal intergenic, and only 9.5% are located at gene promoters (Fig. 5m). Representative ChIP-seq tracks highlighting common regions of peak enrichment for KMT2A, PHF5A, and PHF14 at the introns of *FLT4* and *LINGO2* genes in ChIP-seq samples versus input control of ABCG2⁺ L3.6pl tumorspheres are shown in Fig. 5n, o, respectively. In addition, data from pathway (Fig. 5p), biological process (Fig. 5q), and molecular function (Fig. 5r) enrichment analyses of genes cooccupied by KMT2A, PHF5A, and PHF14 indicate their potential involvement in the regulation of various cellular functions including pluripotency of stem cells.

## KMT2A epigenetically regulates self-renewal and tumorigenicity of PCSCs through the deposition of H3K4me3 marks

The activity of KMT2A as an epigenetic regulator of PCSCs was further validated using MM-102[26,27] (Fig. 6a), a specific inhibitor of the HMT activity of KMT2A by blocking WDR5-KMT2A interaction. Single cells of CSC-enriched pancreatic tumorspheres were treated with different concentrations (0–75 μM) of MM-102 for 5 days (Fig. 6b), followed by western blot analysis of changes in H3K4me2 and H3K4me3 protein levels. This revealed a gradual decrease in H3K4me3 protein levels with increasing drug concentrations as compared to DMSO-treated (control) cells, but only a subtle decrease in H3K4me2 protein levels in SSEA4⁺ A13A cells (Fig. 6c), suggesting that the epigenetic activity of

KMT2A in PCSCs is mainly mediated through the deposition of H3K4me3 marks.

The sphere limiting dilution analysis showed a significant decrease in the total number of tumorspheres formed by ABCG2⁺ L3.6sl cells when treated with MM-102 at a concentration of 75 μM versus DMSO (control) with increasing serial dilutions (1:2) of treated cells (Fig. 6d), verifying the effect of KMT2A inhibition on the impairment of the self-renewal capacity of PCSCs. Brightfield images captured 5 days post-treatment show 3D tumorspheres formed by DMSO-treated (control) ABCG2⁺ L3.6sl (Fig. 6e) and SSEA4⁺ A13A (Fig. 6g) cells. However, tumorsphere formation by cells treated with 75 μM MM-102 was significantly impaired, resulting in only a few small-sized spheres of ABCG2⁺ L3.6sl (Fig. 6f) and SSEA4⁺ A13A (Fig. 6h) cells, as well as cell shrinkage that suggests apoptosis.

To validate our observation of potential apoptosis induced by KMT2A inhibition, we examined the effect of MM-102 treatment on chromatin condensation by flow cytometry analysis of pretreated cells that were stained with cell-permeable Vybrant DyeCycle Violet and cell-impermeable SYTOX AADvanced dyes for identifying apoptotic cells with condensed chromatin and late apoptotic/necrotic cells, respectively. As illustrated in Fig. 6i (lower left panel), MM-102 (75 μM) resulted in a significant decrease in the percentage (%) of viable cells as compared to the DMSO-treated (control) cells. This is demonstrated in representative flow cytometry contour plots by the lower percentage (%) of Vybrant DyeCycle Violet^Lo/SYTOX AADvanced^Lo cells in MM-102-treated cells (Fig. 6i, upper right panel) as compared to the control group (Fig. 6i, upper left panel). We also observed a significant increase in the percentage (%) of necrotic/late apoptotic cells as compared to the control group (Fig. 6i, lower right panel) which is demonstrated in representative flow cytometry contour plots by the higher percentage (%) of Vybrant DyeCycle Violet^Hi/SYTOX AADvanced^Hi cells in MM-102 treated cells (Fig. 6i, upper right panel) as compared to the control group (Fig. 6i, upper left panel).

To test whether MM-102-induced cell death is mediated through apoptosis or necrosis, we stained DMSO (control) and 75 μM MM-102-treated ABCG2⁺ L3.6sl cells with PE-Annexin V and 7-AAD for flow cytometry analysis. This showed a significant decrease in the percentage (%) of viable cells and a significant increase in the percentage (%) of late apoptotic cells in the MM-102-treated group versus the control

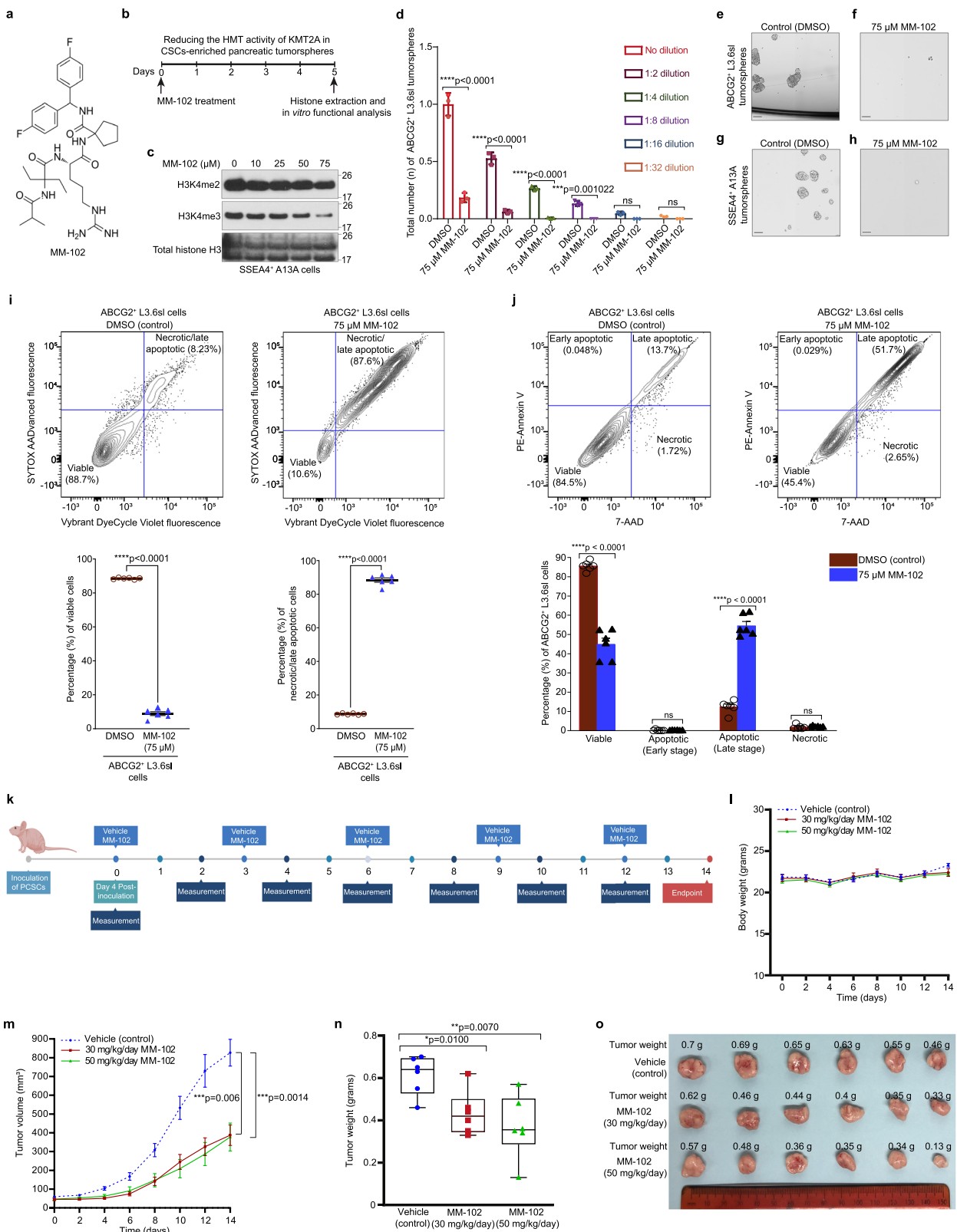

group, with no significant changes observed in the percentages (%) of early apoptotic or necrotic cells (Fig. 6j, lower panel). Representative flow cytometry contour plots of the percentages (%) of viable (PE-Annexin $V^{Lo}$/7-AAD$^{Lo}$), necrotic (PE-Annexin $V^{Lo}$/7-AAD$^{Hi}$), early apoptotic (PE-Annexin $V^{Hi}$/7-AAD$^{Lo}$), and late apoptotic (PE-Annexin $V^{Hi}$/7-AAD$^{Hi}$) cells in the DMSO (control) and 75 μM MM-102-treated ABCG2$^+$

L3.6sl cells are shown in Fig. 6j upper left and upper right panels, respectively.

Next, we investigated the effect of KMT2A inhibition using MM-102 on the in vivo tumorigenicity of PCSCs (ABCG2$^+$ L3.6pl cells) that were subcutaneously xenografted into athymic nude mice homozygous for Foxn1<nu >. First, we performed a pilot study to

**Fig. 6 | MM-102 impairs self-renewal, induces apoptosis, and attenuates in vivo tumorigenicity of PCSCs. a** Chemical structure of MM-102. **b** Schematic illustration depicting the treatment timeline of individual cells of CSC-enriched pancreatic tumorspheres with DMSO (control) and MM-102 for the assessment of H3K4me2 and H3K4me3 protein levels and in vitro functional analysis. The schematic illustration was created with Biorender scientific illustration software. **c** Western blot analysis of H3K4me2 and H3K4me3 protein levels in SSEA4+ A13A cells following treatment with different doses (0 – 75 μM) of MM-102 for 5 days. Total Histone H3 was used as an internal loading control for extracted histones. Data are representative of 2 biologically independent experiments with similar results. **d** Sphere limiting dilution analysis to examine the sphere-forming capacity of ABCG2+ L3.6sl cells treated with 75 μM MM-102 versus DMSO. Data are presented as the mean value ± SD (*n* = 3 biologically independent experiments). Statistical analysis was performed using two-way ANOVA with multiple comparisons with Tukey correction. **e–h** Brightfield images demonstrating the effect of MM-102 treatment at a concentration of 75 μM on the sphere-forming ability of ABCG2+ L3.6sl (**f**) and SSEA4+ A13A (**h**) cells versus ABCG2+ L3.6sl (**e**) and SSEA4+ A13A (**g**) cells treated with DMSO (control). Scale bar, 500 μm. **i, upper panel:** Representative contour plots of flow cytometry analysis of chromatin condensation in ABCG2+ L3.6sl cells treated with DMSO (**left panel**) and 75 μM MM-102 (**right panel**) for 5 days followed by Vybrant DyeCycle Violet and SYTOX AADvanced staining. **i, lower panel:** Scatter graphs demonstrating the percentages (%) of viable (**left panel**) and necrotic/late apoptotic (**right panel**) cells, as determined by flow cytometry analysis of SYTOX AADvanced and Vybrant DyeCycle Violet-stained ABCG2+ L3.6sl cells following DMSO and MM-102 treatments. Data are presented as the mean value ± SEM (n = 6 biologically independent experiments). Statistical analysis was performed using a two-tailed *t* test with Welch's correction for unequal variances. **j, upper panel:** Representative contour plots of flow cytometry analysis of apoptosis following PE-Annexin V and 7-AAD staining of ABCG2+ L3.6sl cells treated with DMSO (**left panel**)

and 75 μM MM-102 (**right panel**) for 5 days. **j, lower panel:** Graphical presentation of the percentages (%) of viable, early apoptotic, late apoptotic, and necrotic ABCG2+ L3.6sl cells treated with DMSO and 75 μM MM-102 for 5 days, as determined by flow cytometry analysis of PE-Annexin V and 7-AAD-stained ABCG2+ L3.6sl cells. Data are presented as the mean value ± SEM (*n* = 6 biologically independent experiments). Statistical analysis was performed using a two-tailed *t* test with Welch's correction for unequal variances. **k** Experimental timeline for testing the in vivo effects of MM-102 in subcutaneous xenografts of ABCG2+ L3.6pl cells in athymic mice homozygous for Foxn1<nu>. The schematic illustration was created with Biorender scientific illustration software. **l**, Body weights (grams) of athymic mice xenografted subcutaneously with ABCG2+ L3.6pl cells and intraperitoneally injected with either vehicle or MM-102 (30 or 50 mg/kg/day) every 3 days. Data are presented as the mean value ± SEM (*n* = 6 mice). Statistical analysis was performed using a two-tailed *t* test with Welch's correction for unequal variances. **m** Volume (mm³) of subcutaneous pancreatic tumors, as assessed by a digital caliper and calculated using the formula: $V = 1/2$ (Length × Width²). Data are presented as the mean value ± SEM (*n* = 6 mice). Statistical analysis was performed using a two-tailed *t* test with Welch's correction for unequal variances. **n** Weight (grams) of subcutaneous pancreatic tumors resected from euthanized mice. Data are presented as the mean value ± SEM (*n* = 6 mice). Statistical analysis was performed using a two-tailed *t* test with Welch's correction for unequal variances. Box plots show the median (center line), upper and lower quartiles (box), and range of the data excluding outliers (whiskers). The upper and lower whiskers represent the maximum (non-outlier) and minimum (non-outlier) values, respectively. **o** Image of resected subcutaneous pancreatic tumors from euthanized mice (*n* = 6) following treatment with either vehicle or MM-102 (30 or 50 mg/kg/day) every 3 days for 14 days. Corresponding tumor weight (grams) is annotated above each resected tumor.

---

determine the maximal tolerated dose (MTD) of MM-102 in mice (*n* = 3) using the following doses: 30, 50, 75, and 100 mg/kg/day. The MTD of MM-102 in mice was 50 mg/kg/day, so we used this dose of MM-102 for drug treatment in vivo. However, to avoid potential toxicity upon prolonged drug administration, we included an additional dose of 30 mg/kg/day. The treatment timeline, including the frequency of drug administration and body weight/tumor volume measurements, is shown in Fig. 6k and described in more detail in the Methods section. No significant changes were observed in the average body weight of mice (*n* = 6) treated with either 30 or 50 mg/kg/day of MM-102 as compared to the vehicle-treated (control) group (Fig. 6l). Average tumor volume was significantly reduced in mice (*n* = 6) treated with either 30 or 50 mg/kg/day of MM-102 as compared to the control group, whereas no significant changes were observed between the 2 treatment doses of MM-102 (Fig. 6m). The average weight of pancreatic tumors resected from euthanized mice (*n* = 6) was also significantly decreased in groups treated with either 30 or 50 mg/kg/day of MM-102 as compared to the control group, with no significant changes observed between the 2 treatment doses of MM-102 (Fig. 6n). An image of resected pancreatic tumors and their corresponding weights is shown in Fig. 6o.

## KMT2A regulates the expression of self-renewal and pluripotency genes in PCSCs

To understand how KMT2A epigenetically regulates PCSCs, we performed an RNA-seq analysis in MM-102-treated ABCG2+ L3.6pl cells to assess global transcriptomic changes upon KMT2A inhibition. However, to avoid the potential effect of drug-induced apoptosis on the quality of sequencing data, cells used for sequencing were treated with a suboptimal concentration of MM-102 (50 μM) for 5 days which preserves the viability of PCSCs (Supplementary Fig. S4a).

We compared the changes in gene expression levels upon partial inhibition of the enzymatic activity of KMT2A using MM-102 to gemcitabine monotherapy, as well as combined MM-102 and gemcitabine in ABCG2+ L3.6pl cells. A summary of the workflow for RNA-seq

analysis in ABCG2+ L3.6pl cells is shown in Fig. 7a. The analysis included DMSO (control), MM-102 (50 μM), gemcitabine (100 nm), and combined MM-102 (50 μM) and gemcitabine (100 nm) treatments. Principal component analysis (PCA) of RNA-seq data shows an obvious difference in the transcriptomic changes between groups treated with MM-102 as a single agent or when combined with gemcitabine and the ones treated with DMSO (control) or gemcitabine. This is demonstrated by PC1 which accounts for 80.04% of data variability (Fig. 7b), indicating the distinct changes induced by MM-102 in the gene expression profile of PCSCs.

As compared to the DMSO-treated (control) group, MM-102 monotherapy resulted in a significant upregulation of 768 genes and a significant downregulation of 1317 genes (Fig. 7c). There were a few transcriptomic changes associated with gemcitabine treatment, as only 11 genes were upregulated and 5 genes were downregulated significantly (Fig. 7d), however, when combined with MM-102, 668 genes were significantly upregulated, along with 1193 genes being significantly downregulated (Fig. 7e).

We found that only 6% of genes cooccupied by KMT2A, PHF5A, and PHF14 were differentially expressed upon partial inhibition of KMT2A's enzymatic activity. In addition, our differential PHF5A and PHF14 ChIP-seq analyses in ABCG2+ L3.6pl cells treated with MM-102 (50 μM) revealed a significant decrease in the binding of PHF14 to its target DNA binding sites in only a few number of genes (7% [12/171] of KMT2A, PHF5A, and PHF14-associated genes; Supplementary Fig. S4b) as compared to DMSO-treated (control) cells, whereas no significant changes were observed in the binding profile of PHF5A. ChIP-qPCR analysis showed a significant decrease in the enrichment of H3K4me3 and PHF14 at the target binding site within the intronic region of *PAK3* (identified by our ChIP-seq analysis), with no significant changes observed in the enrichment of PHF5A or RNA Pol II in MM-102 versus DMSO-treated (control) ABCG2+ L3.6pl cells (Supplementary Fig. S4c). A representative ChIP-seq track illustrating the significant decrease in PHF14 peak enrichment at the highlighted target site within an intronic region of *PAK3* gene in ABCG2+ L3.6pl cells treated with 50 μM MM-102 versus DMSO is shown in Supplementary Fig. S4d. Collectively, these

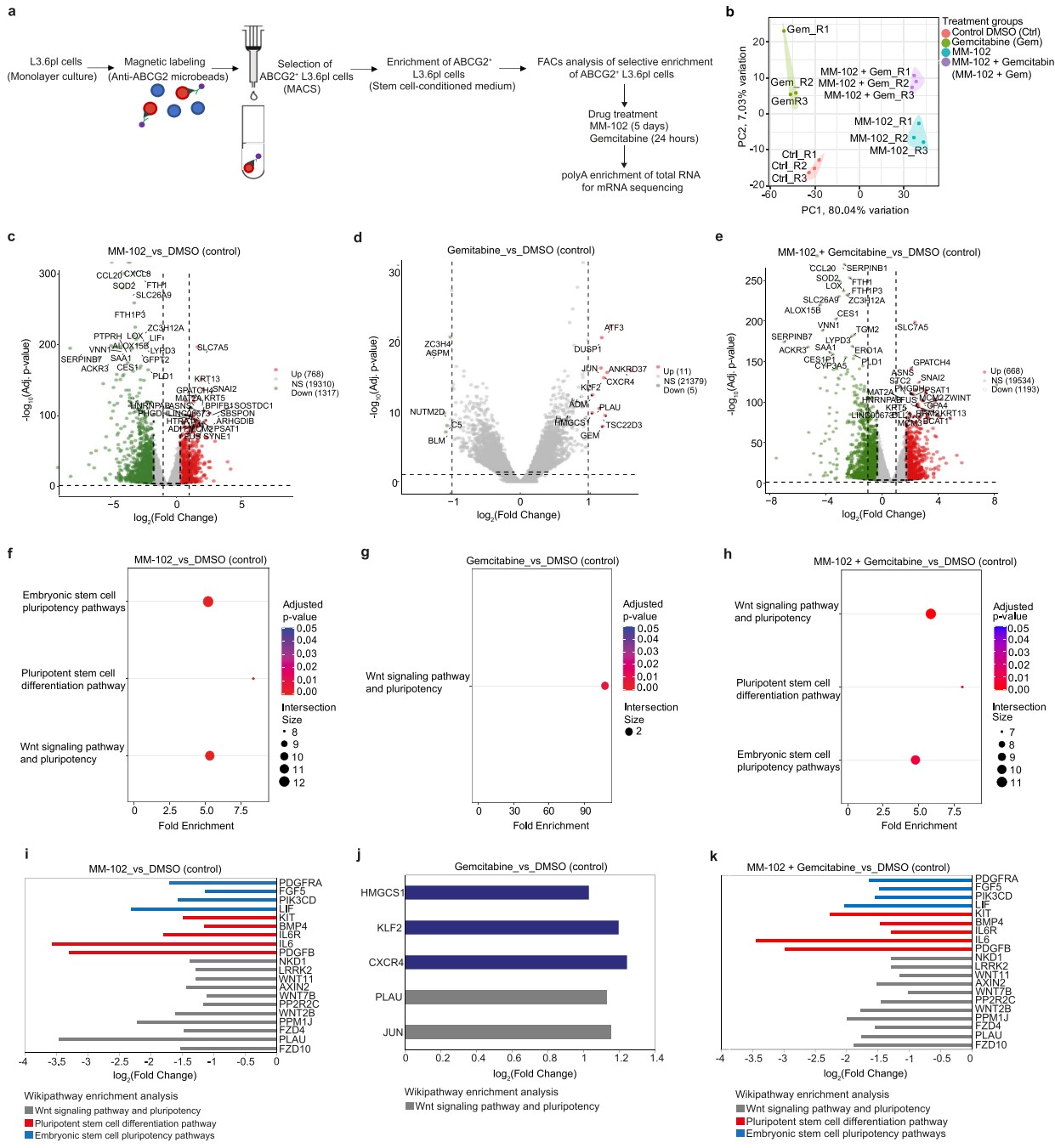

**Fig. 7 | Transcriptomic changes upon partial KMT2A inhibition and gemcitabine treatment in PCSCs. a** Schematic illustration of the workflow for RNA-seq analysis in ABCG2⁺ L3.6pl cells treated with MM-102 (50 μM) for 5 days and gemcitabine (100 nm) for 24 h. The schematic illustration was created with Biorender scientific illustration software. **b** Principal component analysis (PCA) of normalized RNA-seq data of ABCG2⁺ L3.6pl cells treated with DMSO (control), MM-102 (50 μM), gemcitabine (100 nm), and combined MM-102 (50 μM) and gemcitabine (100 nm), demonstrating the clustering of RNA-seq samples by treatment. Each treatment group consists of 3 biological replicates. PC1 and PC2 account for 80.04% and 7.03% of data variability, respectively. **c–e** Volcano scatter plots demonstrating significantly upregulated (red) and significantly downregulated (green) genes in ABCG2⁺ L3.6pl cells treated with MM-102 (**c**), gemcitabine (**d**), and combined MM-102 and gemcitabine (**e**) as compared to the DMSO-treated (control) group (*n* = 3 biologically independent experiments). The top 20 significantly downregulated and upregulated genes are annotated. Significant differential gene expression was defined as |log2 Fold change| > 1 for upregulated genes and <−1 for downregulated genes at an adjusted *p*-value < 0.05. The statistical test used was two-sided. The

effect size was shrunk using the ashr method and *p*-values were adjusted using the independent hypothesis weighting (IHW) method. **f–h** WikiPathway enrichment analyses of significantly DEGs in ABCG2⁺ L3.6pl cells treated with MM-102 as a single agent (**f**), gemcitabine (**g**), and combined MM-102 and gemcitabine (**h**) as compared to the DMSO-treated (control) group (*n* = 3 biologically independent experiments). The size and color of the dots represent the number of genes enriched in the pathway of interest and the significance of enrichment, respectively. The statistical test was performed using the method provided by g:GOSt which uses the well-proven cumulative hypergeometric test. The multiple testing correction was performed using the default g:SCS (Set Counts and Sizes) correction method. **i–k** Bar charts illustrating log2(fold change) of expression levels of significantly DEGs involved in the regulation of self-renewal and pluripotency in ABCG2⁺ L3.6pl cells treated with MM-102 (**i**), gemcitabine (**j**), and combined MM-102 and gemcitabine (**k**) versus DMSO-treated (control) group (*n* = 3 biologically independent experiments). Significant differential gene expression was defined as |log2 Fold change| > 1 for upregulated genes and <− 1 for downregulated genes at an adjusted *p*-value < 0.05.

findings may provide one possible explanation for the few differentially expressed genes (DEGs) that are cooccupied by KMT2A, PHF5A, and PHF14 upon partial KMT2A inhibition in PCSCs.

In terms of cell stemness and pluripotency regulation, pathway enrichment analysis of DEGs revealed that genes enriched in Wnt signaling, pluripotent stem cell differentiation, and embryonic stem cell pluripotency pathways are significantly downregulated in ABCG2$^+$ L3.6pl cells treated with MM-102 as a single agent (Fig. 7f) or when combined with gemcitabine (Fig. 7h), but significantly upregulated in the gemcitabine-treated cells (Fig. 7g) as compared to the DMSO-treated (control) cells. Log2 fold changes of some of those enriched genes in MM102, gemcitabine, and combined MM-102 and gemcitabine-treated groups are shown in Fig. 7i–k, respectively.

Based on the pathway enrichment analysis of other genes significantly downregulated in MM-102 (Supplementary Fig. S5a) and combined gemcitabine and MM-102 (Supplementary Fig. S5b) treatment groups as compared to the DMSO-treated (control) group, many of those genes were found to be significantly enriched in oncogenic signaling pathways, including for instance PI3K-AKT, JAK-STAT, MAPK, RAS, and Hippo signaling pathways. In the gemcitabine-treated group, however, significantly upregulated genes were those found to be enriched in oncogenic signaling pathways, such as the Interleukin-18 (IL-18) and TGF-β signaling pathways (Supplementary Fig. S5c).

Furthermore, MM-102 treatment as a single agent (Supplementary Fig. S5a) or in combination with gemcitabine (Supplementary Fig. S5b) resulted in a significant downregulation of genes involved in nuclear factor erythroid 2 related factor 2 (NRF2) signaling. Gene set enrichment analysis (GSEA) ridge plots demonstrate the downregulation of enriched NRF2 genes in MM-102 versus control, MM-102 and gemcitabine versus gemcitabine, as well as MM-102 and gemcitabine versus control treatment groups, but not in the gemcitabine versus control group which shows upregulation of enriched NRF2 genes (Supplementary Fig. S5d). Furthermore, GSEA ridge plots for genes involved in epithelial-mesenchymal transition (Supplementary Fig. S5e) and MAPK signaling (Supplementary Fig. S5f) show the downregulation of those genes in MM-102 versus control, MM-102 and gemcitabine versus gemcitabine, in addition to MM-102 and gemcitabine versus control treatment groups, but their upregulation in the gemcitabine versus the control group. Altogether, these findings suggest that reduced HMT activity of KMT2A has the effect of blocking multiple cellular mechanisms and signaling pathways that regulate the maintenance, tumorigenesis, and chemoresistance of PCSCs.

## Discussion

*PHF5A* gene encodes a small and structurally conserved chromatin-associated protein composed of 110 amino acid subunits, with a characteristic PHD zinc finger domain[28] that is essential for its binding to histone H3[29]. As a subunit of the splicing factor 3b (SF3b), a component of the U2 small nuclear ribonucleoprotein (U2 snRNP) splicing complex, PHF5A facilitates the interactions between the U2 snRNP complex and ATP-dependent helicases[30]. Additionally, it is implicated in the progression of colorectal[31], breast[29], and lung[32,33] cancers. An RNAi-based screen conducted by Hubert et al. in patient-derived GSCs revealed the role of PHF5A in regulating the viability and growth of tumor-initiating GSCs by facilitating the recognition of 3′ splice sites with C-rich polypyrimidine tracts in tumor-initiating GSCs[7]. Furthermore, an RNAi-based screen in hESCs identified PHF5A as a regulator of stem cell maintenance and pluripotency by regulating transcriptional elongation and RNA Pol II pause release of pluripotency genes. In hESCs, PHF5A associates with components of RNA polymerase II-associated factor (PAF1) complex, including PAF1, LEO, CTR9, and cell division cycle 73 (CDC73), and plays an important role in regulating the stability of the PHF5A-PAF1-LEO-CTR9-CDC73 complex and its recruitment to the regulatory regions of pluripotency genes[8]. In PCSC subpopulations studied by Karmakar et al, however, PHF5A does not interact with LEO1, CTR9, or CDC73, but is a part of a protein subcomplex containing PAF1 and DDX3. Inhibiting the activity of DDX3 by RK-33, a small molecule inhibitor of the helicase activity of DDX3, reduced the localization of DDX3 and PAF1 on the *Nanog* promoter. Functionally, RK-33 decreased the expression of PAF1 and CSC markers in PCSCs, but only had a minimal effect on their expression in non-CSCs. DDX3 inhibition also impaired the sphere-forming capacity of PCSCs and triggered their apoptosis, with less pronounced cell death observed in normal human fibroblasts[9]. Altogether, these data indicate a critical role for PHF5A in diverse cellular contexts and biological processes, which likely entails the involvement of multiple binding cofactors.

LC-MS/MS analysis of nuclear Co-IPs of PHF5A in PCSCs enabled the identification of several binding partners of PHF5A, including a physical association with PHF14 which possesses similar abilities to PHF5A in recognizing modified histone signatures and regulating epigenetic mechanisms[18]. ChIP-seq analysis revealed their predominant cooccupancy of genomic target sites within gene bodies. This could involve potential association with distal cis-regulatory elements (i.e., enhancer regions), known to be prevalent in intronic regions, and which play a significant role in regulating gene transcription by contacting gene promoters via tracking, linking, and chromatin looping mechanisms[34]. Integrating our LC-MS/MS findings of nuclear binding partners of PHF5A in PCSCs into the STRING network database allowed us to identify a physical association between PHF5A and RAI1 protein complex subunits, including PHF14, HMG20A, and RAI1[20]. However, our LC-MS/MS analysis of nuclear PHF5A Co-IPs in ABCG2$^+$ L3.6pl tumorspheres indicates that PHF5A does not associate with any of the nuclear binding partners identified by Strikoudis et al. [8] or Karmakar et al. [9], suggesting that hESCs and different subpopulations of PCSCs maintain their stemness features and functions through distinct PPIs that involve PHF5A to fulfill cell type-specific requirements and functions.

Our small molecule compound screening of epigenetic modulating enzymes and in vitro cell-based functional assays identified KMT2A as an epigenetic regulator of PCSCs, while interactome analysis revealed its physical association with RNA Pol II-associated PHF5A-PHF14-HMG20A-RAI1 protein subcomplex, which supports previously proposed models suggesting the physical association between KMT2A and RAI1 complex subunits to access target genomic binding sites[35]. This protein subcomplex was identified in two different types of PCSCs, derived from PASC and PDAC neoplasms, which were phenotypically sorted based on their surface expression of CSC markers, including ABCG2, CD44, and CD133. In addition, the complex was detected in SSEA4$^+$ A13A tumorspheres, but lacked the RAI1 subunit, suggesting a cell-type-dependent association of the RAI1 subunit. However, further investigation is required to assess the physical association of RAI1 with the PHF5A subcomplex in different CSC populations as well as its functional role within the protein subcomplex.

While Aldefluor and Hoechst 33342 exclusion assays have been used in various studies to identify and isolate CSCs, our experiments did not show significant enrichment of aldehyde dehydrogenase-bright (ALDH$^{br}$) or side-population (SP) cells in PCSCs versus their parental monolayers lacking CSC enrichment. This may be attributed to the heterogeneous nature of CSCs and their associated markers which are not universal among all CSC populations[36]. Accordingly, we did not specifically study our identified protein subcomplex in ALDH$^{br}$ or SP cells.

Our proteomics data were further verified by ChIP-seq analysis which demonstrated common genomic loci occupied by KMT2A, PHF5A, and PHF14 in 171 genes. The predominant cooccupancy of KMT2A, PHF5A, and PHF14 at genomic sites within the gene body and intergenic regions extends our previous observation of the genomic distribution of DNA binding sites cooccupied by PHF5A and PHF14 and

suggests that the PHF5A protein subcomplex may be associated with enhancer regions containing H3K4me3 marks[37]. Future studies involving sequencing techniques, such as chromatin interaction analysis and paired-end tag (chIA-PET)[38] sequencing may be able to identify potential long-range chromatin interactions between distal regulatory regions occupied by the PHF5A protein subcomplex and gene promoters, which could provide additional information about genes that are coregulated by those factors, since genes identified by our ChIP-seq analysis may not be regulated by the PHF5A protein subcomplex, especially when associated with enhancer regions.

In vitro cell-based assays in PCSCs using different KMT2A inhibitors corroborated our findings from the small molecule screening, as they demonstrated the association of KMT2A inhibition in PCSCs with impaired self-renewal capacity and induction of apoptosis. In addition, our in vivo data demonstrate the effect of KMT2A inhibition using MM-102 on impairing the tumorigenicity of PCSCs. Future follow-up studies will evaluate the in vivo effects of additional KMT2A inhibitors and compare their pharmacokinetics data by measuring drug concentrations achieved in blood and tumors in order to select the best candidate for clinical evaluation.

While WDR5 is highly conserved in many protein complexes[39], a study conducted by Cao et al. showed that WDR5 is essential for the stability and HMT activity of KMT2A, but not for other members of the KMT2 family proteins[40], indicating the selectivity and therapeutic applicability of KMT2A-WDR5 inhibitors.

Our RNA-seq analysis revealed an association between reduced HMT activity of KMT2A and transcriptional changes in PCSCs. However, only 6% of the genes whose target sites are cooccupied by KMT2A, PHF5A, and PHF14 were found to be differentially expressed upon partial KMT2A inhibition. There are two possible mechanisms of how KMT2A's enzymatic activity relates to the PHF5A-PHF14-HMG20A-RAI1 protein subcomplex: Firstly, as we have proposed, KMT2A enzyme catalyzes methylated histone marks that are recognized by the protein subcomplex subunits, which would impact their chromatin binding. To validate this hypothesis, we treated PCSCs with MM-102 at a concentration of 50 μM. This concentration is not associated with an apoptotic activity, which is critical for ensuring the quality of sequencing data. A significant decrease in peak enrichment for PHF14 was observed in only 7% (12/171) of genes cooccupied by KMT2A, PHF5A, and PHF14. This may be due to the partial inhibition of the enzymatic activity of KMT2A at 50 μM, resulting in only a subtle decrease in H3K4me3 protein levels which may not be sufficient to significantly modulate the binding of PHF14 and/or PHF5A to their target genomic sites or to impair the viability of cells. This limitation may be overcome, however, by the generation of inducible KMT2A knockouts which preserve the stemness of PCSCs across multiple passages through enhanced CSC-enriching conditions, as well as developing more potent small molecule drug inhibitors of KMTA. As a second point, KMT2A may catalyze the methylation of histone marks in complex with PHF5A-PHF14-HMG20A-RAI1. This may be true for most loci, where inhibition of KMT2A has no significant effect on the stability of the complex or binding of its components to target genomic sites, but impacts other aspects such as gene expression. There is also a possibility that not all genes identified by our ChIP-seq analysis are regulated by the PHF5A subcomplex, as discussed earlier.

While MM-102 resulted in a significant downregulation of pluripotency and oncogenic genes, we observed a significant upregulation of self-renewal and pluripotency-associated genes, such as CXCR4, Kruppel-like factor 2 (KLF2), and 3-hydroxy-3-methylglutaryl-CoA synthase 1 (HMGCS1) in response to gemcitabine treatment. Consistent with our findings, Singh et al. reported the association of gemcitabine with a dose and time-dependent increase in CXCR4 expression levels in PC cells and also demonstrated the role of the Chemokine Ligand-2 (CXCL2)/CXCR4 signaling axis in driving resistance to gemcitabine-induced apoptosis[41]. In mouse embryonic stem cells (mESCs), KLF2 was found to be the most effective among other KLF proteins in sustaining their ground pluripotent state[42]. Moreover, the mevalonate precursor enzyme HMGCS1 has been identified as a CSC marker in both luminal and basal tumor subtypes of breast cancer, with expression levels strongly linked to high tumor grades. The HMGCS1 gene was also found to support various CSC activities in breast cancer by serving as a gatekeeper for dysregulated mevalonate metabolism[43]. Interestingly, those genes were suppressed in PCSCs treated with MM-102 as a single agent and in combination with gemcitabine. Altogether, these findings suggest that KMT2A inhibition could reverse the resistance of PCSCs to the cytotoxic effects of gemcitabine which lends support for preclinical investigation.

In summary, our study identifies KMT2A as an epigenetic regulator of PCSC properties and functions, as well as a key member of the RNA Pol II-associated PHF5A-PHF14-HMG20A-RAI1 protein subcomplex. Epigenetic targeting of the protein subcomplex by interfering with the enzymatic activity of KMT2A attenuates the stemness and tumorigenic properties of PCSCs. In addition, our RNA-seq data suggest that KMT2A may be involved in epigenetic regulatory processes beyond its association with the PHF5A subcomplex. Accordingly, KMT2A inhibition could be investigated as an adjunct to chemotherapy in future preclinical and clinical studies to improve the clinical outcomes of PC patients.

## Methods

### Animal work

Female Foxn1[nu] athymic nude mice (6 weeks old) were purchased from the Jackson Laboratory and housed in a pathogen-free environment under ambient temperature and standard light-dark cycle conditions at the Genome and Biomedical facility at the University of California Davis (USA). For subcutaneous engraftment, $0.19 \times 10^6$ ABCG2$^+$ L3.6pl cells suspended in 100 μl of matrigel (Fisher Scientific, CB-40230C) were injected subcutaneously into the left flank of mice. Four days post-inoculation, xenografted mice were randomized into 3 groups (6 mice per group) and intraperitoneally injected with either vehicle (90% sulfobutylether-β-Cyclodextrin (SBE-β-CD) in normal saline + 10% DMSO) or MM-102 (Tocris Bioscience, 5307) suspended in vehicle (90% SBE-β-CD in normal saline + 10% DMSO) at 30 or 50 mg/kg/day every 3 days for 14 days, with measurements of body weight and tumor volume recorded every 2 days. In our experiments, the maximum tumor size at the largest diameter was 12.5 mm which is lower than the maximum tumor size permitted by the Institutional Animal Care and Use Committee (IACUC; 20 mm). Tumor volume (mm$^3$) was measured with a Vernier caliper and calculated using the formula: $V = 1/2$ (Length × Width$^2$)[44]. Following the completion of the drug administration course, mice were euthanized in a chamber that was gradually filled with 100% $CO_2$ from a compressed gas cylinder, then tumors were carefully resected for imaging and measurement of tumor weight. All experiments were conducted in accordance with procedures approved by IACUC.

### Cell culture

L3.6pl, L3.6sl, and FG PC cell lines, all with a KRAS; Simple; p.Gly12Asp (c.35G>A) genetic background, were purchased from the MD Anderson Cancer Center (USA). A13A and A13B PC cell lines were kindly gifted by Prof. Christine A. Iacobuzio-Donahue (Memorial Sloan Kettering Cancer Center, USA). The genetic backgrounds of A13A and A13B cell lines are copy number gain of GATA-6 and cTAGE1; KRAS G12V; TpS3 WT; SMAD4 WT, and copy number gain of GATA-6 and cTAGE1, respectively. Monolayer cultures of L3.6pl, L3.6sl, FG, A13A, and A13B were maintained in Dulbecco's Modified Eagle Medium (DMEM; Thermo Fisher Scientific, 31966047) supplemented with 10% heat-inactivated fetal bovine serum (FBS; Sigma-Aldrich, F9665), 1x minimal essential medium (MEM) vitamins (Thermo Fisher Scientific, 11120052), 1x MEM non-essential amino acids (Thermo Fisher

Scientific, 11140035), and 1% Penicillin-Streptomycin (Thermo Fisher Scientific, 15140122). The NIH-3T3 cell line, kindly gifted by Prof. Udo Oppermann (Botnar Research Centre, UK), was maintained in DMEM (Thermo Fisher Scientific, 31966047) supplemented with 10% heat-inactivated FBS (Sigma-Aldrich, F9665) and 1% Penicillin-Streptomycin (Thermo Fisher Scientific, 15140122). MACS-enriched pancreatic tumorspheres were maintained in serum-free, stem cell-conditioned culture medium consisting of 1:1 DMEM/Nutrient Mixture F-12 Ham (Thermo Fisher Scientific, 21041025) supplemented with 2% B-27 supplement (Thermo Fisher Scientific, 17504001), thermostable recombinant human basic fibroblast growth factor (Proteintech, HZ-1285) at a final concentration of 20 ng/ml, and 1% Penicillin/Streptomycin. The HPDE6c7 cell line (Kerafast, ECA001-FP), with normal *KRAS, Tp53*, c-*myc*, and *p16*$^{INK4A}$ genotypes, was maintained in keratinocyte serum-free medium supplemented with human recombinant epidermal growth factor and bovine pituitary extract (Thermo Fisher Scientific, 17005042). All cell lines were maintained under standard culture conditions in a humidified cell culture incubator at 37 °C and 5% $CO_2$. Short Tandem Repeat DNA profiling was used to authenticate cell lines. A PCR-based method for the detection of *Mycoplasma* contamination in cell cultures was routinely performed using the following primer sequences: forward primer; 5'-GGGAGCAAACAGGATTAGA-TACCCT-3' and reverse primer; 5'-TGCACCATCTGTCACTCTGT-TAACCTC-3'.

## Tumorsphere formation assay

PC cells, including L3.6pl, L3.6sl, A13A, and A13B were seeded in ultra-low attachment 6-well plates at a density of 5 ×10³ cells/well in serum-free, stem cell-conditioned culture medium. Pancreatic tumorspheres were harvested on day 6 post-seeding using a 40 μm cell strainer (pluriSelect, 43-50040-51) and dissociated with TrypLE Express enzyme (Thermo Fisher Scientific, 12604021). Single cells were reseeded at their initial cell density under the same CSC-enriching culture conditions for 6 additional days, then finally recollected for flow cytometry analysis of enriched CSC surface markers. Brightfield images of pancreatic tumorspheres were acquired using Celigo Imaging Cytometer (Nexcelom Bioscience, Version 2.1).

To assess the effect of reduced HMT activity of KMT2A on the sphere-forming capacity of PCSCs, single cells were seeded in ultra-low attachment 6-well plates at a density of 5 ×10³ cells/well in stem cell-conditioned culture medium and treated with either DMSO (Sigma-Aldrich, D8418), 75 μM MM-102 (Tocris Bioscience, 5307), 10 μM OICR-9429 (Sigma-Aldrich, SML1209), or 10 μM MM-401 (MedChemExpress, HY-19554A) for 5 days in a humidified cell culture incubator at 37 °C and 5% $CO_2$. Following treatment, pancreatic tumorspheres (100 μm colony diameter) were counted using Celigo Imaging Cytometer (Nexcelom Bioscience, Version 2.1).

## Flow cytometry

**Enrichment of CSC surface markers.** Single cells were suspended in ice-cold cell staining buffer composed of Dulbecco's phosphate buffered saline (DPBS; Thermo Fisher Scientific, 14190169), 10% heat-inactivated FBS (Sigma-Aldrich, F9665), and 0.1% sodium azide (Sigma-Aldrich, S2002) then incubated with the Fc receptor blocking reagent human TrueStain FcX (Biolegend, 422301) for 10 min at room temperature. Cells were labeled with fluorochrome-conjugated antibodies, including PerCP/Cy5.5 anti-ABCG2 (BioLegend, 332024; 1:50), APC anti-EPCAM (BioLegend, 324208; 1:50), PE anti-CD44 (BD Biosciences, 555479; 1:10), BV421 anti-CD24 (BD Biosciences, 562789; 1:50), BV786 anti-PROM1 (BD Biosciences, 747640; 1:50), FITC anti-SSEA4 (BD Biosciences, 560126; 1:10), and BV510 anti-CXCR4 (BioLegend, 306535; 1:50) for 30 min on ice in the dark. FMO and isotype control antibodies, including PerCP-Cy5.5 mouse IgG2b, k (BD Biosciences, 558020; 1:50), APC mouse IgG2b, k (BioLegend, 402206; 1:50), PE mouse IgG2a, k (BD Biosciences, 555574; 1:10), BV421 mouse IgG2a, k (BD Biosciences,

562439; 1:50), BV786 mouse IgG1, k (BD Biosciences, 563330; 1:50), FITC mouse IgG3, k (BD Biosciences, 556658; 1:10), and BV510 mouse IgG2a, k (BioLegend, 400267; 1:50) were used to gate and identify the positive cell population. Following labeling, cells were washed 3 times with the cell staining buffer then incubated with 4', 6-diamidino-2-phenylindole (DAPI; BD Biosciences, 564907) at a final concentration of 0.1 μg/ml in cell staining buffer for 15 min at room temperature in the dark for the assessment of cell viability. Flow cytometry data were acquired using BD LSRFortessa cell analyzer (BD Biosciences), collected with FACSDiva software (BD, Version8.0.1), and analyzed using FlowJo v10.8 software (BD Life Sciences).

**Analysis of SOX2 protein expression levels.** We analyzed SOX2 protein expression levels by flow cytometry using the True-Nuclear transcription factor buffer set (BioLegend, 424401) and Pacific Blue anti-SOX2 antibody (BioLegend, 656111; 1:20) as per the manufacturer's instructions.

**Apoptosis detection assays.** Day 6 CSC-enriched pancreatic tumorspheres were harvested using a 40 μm cell strainer (pluriSelect, 43-50040-51) and dissociated with TrypLE Express enzyme (Thermo Fisher Scientific, 12604021). Single cells were seeded in ultra-low attachment 6 well plates at a density of 0.25 × 10⁶ cells per well in stem cell-conditioned culture medium then treated with either DMSO (Sigma-Aldrich, D8418), MM-102 (Tocris Bioscience, 5307) at a final concentration of 50 or 75 μM, 10 μM OICR-9429 (Sigma-Aldrich, SML1209), or 10 μM MM-401 (MedChemExpress, HY-19554A) for 5 days in a humidified cell culture incubator at 37 °C and 5% $CO_2$. To assess chromatin condensation as a readout of apoptosis, cells were stained with Vybrant DyeCycle Violet and SYTOX AADvanced dyes (Thermo Fisher Scientific, A35135) as per the manufacturer's instructions. To distinguish between viable, early apoptotic, late apoptotic, and necrotic cells, pretreated PCSCs were stained with PE-Annexin V and 7-AAD (Biolegend, 640934) according to the manufacturer's instructions. Flow cytometry data were acquired using BD LSRFortessa cell analyzer (BD Biosciences), collected with FACSDiva software (BD, Version 8.0.1), and analyzed using FlowJo v10.8 software (BD Life Sciences).

## Magnetic-activated cell sorting (MACS)

PC cells were suspended in ice-cold MACS buffer consisting of PBS, pH 7.2 (Thermo Fisher Scientific, 10010023), 0.5% bovine serum albumin (BSA; Sigma-Aldrich, A7906), 2 mM ethylenediaminetetraacetic acid (EDTA; Thermo Fisher Scientific, 15575020), and RevitaCell supplement (Thermo Fisher Scientific, A2644501) then incubated with magnetic microbeads conjugated with anti-ABCG2 (Miltenyi Biotec, 130-107-680; 1:5) and anti-SSEA4 (Miltenyi Biotec, 130-097-855; 1:5) antibodies for 15 min at 4 °C. Magnetically labeled cells were then sorted using an LS column (Miltenyi Biotec, 130-042-401) inserted into a MediMACS separator (Miltenyi Biotec, 130-042-302) as per the manufacturer's instructions.

## Quantitative real-time PCR (qRT-PCR)

Total RNA was extracted with TRIzol reagent (Thermo Fisher Scientific, 15596018) and purified using a Direct-zol RNA miniprep kit (Zymo Research, R2050) according to the manufacturer's instructions. Reverse transcription of 0.5-1 ug of total RNA was performed using a High-Capacity RNA-to-cDNA kit (Thermo Fisher Scientific, 4387406) as per the manufacturer's instructions. Gene-specific primers, spanning exon junctions, were designed using DNADynamo software (version 1.0) and custom synthesized at Thermo Fisher Scientific. Sequences of primer pairs used for qRT-PCR analysis are listed in Supplementary Table 2. The reaction mixture contained Power SYBR Green PCR master mix (Thermo Fisher Scientific, 4367659), cDNA, primers, and nuclease-free water. Gene expression analysis was performed using a

ViiA7 real-time PCR system (Applied Biosystems) with QuantStudio software v1.6.1 (Applied Biosystems). qRT-PCR data were normalized to *ACTB* and graphically presented as the fold change of gene expression in the test samples as compared to the control sample.

## Co-immunoprecipitation (Co-IP)

For SMAD2/3 nuclear Co-IP experiments, cells were pretreated with recombinant INHBA protein (R and D Systems, 338-AC-050/CF) at a final concentration of 100 ng/ml for 24 h. Nuclear protein complexes were extracted using the nuclear complex Co-IP kit (Active Motif, 54001) following the manufacturer's instructions. Nuclear protein extracts were incubated with target-specific antibodies, including anti-SMAD2/3 (R and D Systems, AF3797; 1:200), anti-PHF5A (Proteintech, 15554-1-AP; 1:80), anti-PHF14 (Proteintech, 24787-1-AP; 1:260), and anti-KMT2A (Abcam, ab272023; 1:200), as well as isotype control antibodies, including normal goat IgG (R and D Systems, AB-108-C; 1:200) and rabbit (DA1E) mAb IgG XP (Cell Signaling Technology, 3900S; 1:500) for 2 h at 4 °C with agitation, followed by overnight incubation with Protein A/G Plus-agarose beads (Santa Cruz Biotechnology, sc-2003) at 4 °C with agitation. Agarose beads were washed 3 times with 1x PBS then suspended in 1x Laemmli sample buffer supplemented with 2-mercaptoethanol (Sigma-Aldrich, M6250) at a final concentration of 2.5%. Samples were boiled at 95 °C for 5 min to elute co-immunoprecipitated proteins for western blotting and LC-MS/MS analysis.

## Mass spectrometry (MS)

LC-MS/MS analysis of nuclear PHF5A Co-IP samples was performed at the Cambridge Centre for Proteomics (Cambridge, UK) using a Dionex Ultimate 3000 RSLC nanoUPLC (Thermo Fisher Scientific Inc, Waltham, MA, USA) system and a Q Exactive Orbitrap mass spectrometer (Thermo Fisher Scientific Inc, Waltham, MA, USA). Separation of peptides was performed by reverse-phase chromatography at a flow rate of 300 nl/min and a Thermo Scientific reverse-phase nano Easy-spray column (Thermo Scientific PepMap C18, 2 μm particle size, 100 A pore size, 75 μm i.d. × 50 cm length). Peptides were loaded onto a pre-column (Thermo Scientific PepMap 100 C18, 5 μm particle size, 100 A pore size, 300 μm i.d. × 5 mm length) from the Ultimate 3000 auto-sampler with 0.1% formic acid for 3 min at a flow rate of 15 μl/min. After this period, the column valve was switched to allow the elution of peptides from the pre-column onto the analytical column. Solvent A was water + 0.1% formic acid, and solvent B was 80% acetonitrile, 20% water + 0.1% formic acid. The linear gradient employed was 2–40% B in 90 min (the total run time including column washing and re-equilibration was 120 min). The LC eluant was sprayed into the mass spectrometer by means of an Easy-spray source (Thermo Fisher Scientific Inc.). All *m/z* values of eluting ions were measured in an Orbitrap mass analyzer, set at a resolution of 35,000 and scanned between *m/z* 380–1500. Data-dependent scans (Top 20) were employed to automatically isolate and generate fragment ions by higher energy collisional dissociation (HCD, Normalized collision energy (NCE):25%) in the HCD collision cell, and measurement of the resulting fragment ions was performed in the Orbitrap analyzer, set at a resolution of 17,500. Singly charged ions and ions with unassigned charge states were excluded from being selected for MS/MS, and a dynamic exclusion of 60 s was employed. Post-run, all MS/MS data were converted to mgf files which were then submitted to the Mascot search algorithm (Matrix Science, London UK, version 2.6.0) and searched against a common contaminants database (cRAP_20190401.fasta) and the UniProt human database (CCP_UniProt_homo sapiens_proteome_20180409 database (93734 entries)), assuming the digestion enzyme trypsin. Mascot was searched with a fragment ion mass tolerance of 0.100 Da and a parent ion tolerance of 20 PPM. Carbamidomethyl of cysteine was specified in Mascot as a fixed modification. Deamidation of asparagine and glutamine and oxidation of methionine were specified in Mascot as variable modifications. Scaffold (version Scaffold_4.10.0, Proteome software Inc., Portland, OR) was used to validate MS/MS-based peptide and protein identifications. Peptide identifications were accepted if they could be established at greater than 95.0% probability by the Peptide Prophet algorithm with Scaffold delta-mass correction. Protein identifications were accepted if they could be established at greater than 99.0% probability and contained at least 2 identified peptides. Protein probabilities were assigned by the Protein Prophet algorithm. Proteins that contained similar peptides and could not be differentiated based on MS/MS analysis alone were grouped to satisfy the principles of parsimony. Proteins sharing significant peptide evidence were grouped into clusters. The normalization of total spectral counts involved 3 steps: (1) calculation of the total number of spectra in each biosample; (2) calculation of the average number of spectra across all biosamples; and (3) multiplying each spectrum count in each biosample by the average count over the biosample's total spectral count.

## Protein extraction and western blotting

Whole cell lysates were extracted with ice-cold radio-immunoprecipitation assay (RIPA) lysis buffer (150 mM NaCl, 1% Triton X-100, 0.5% sodium deoxycholate, 0.1% sodium dodecyl sulfate (SDS), and 50 mM Tris, pH 8.0) supplemented with 1x EDTA-free protease inhibitor cocktail (Roche, 04693159001) and 1x phosphatase inhibitor cocktail (Roche, 4906845001). Nuclear proteins were extracted with nuclear digestion buffer (Active Motif, 54001) supplemented with protease and phosphatase inhibitors. Histones were extracted by lysing cells in Triton extraction buffer (PBS containing 0.5% Triton X-100, 2 mM phenylmethylsulfonyl fluoride (PMSF), and 0.02% sodium azide) at a density of $10^7$ cells per ml on ice for 10 min followed by centrifugation at 6500*g* for 10 min at 4 °C. The supernatant was then discarded, and the nuclei were washed in half the volume of Triton extraction buffer and recentrifuged at 6500 x *g* for 10 min at 4 °C. The resulting supernatant was discarded, and the pellet was suspended in 0.2 N HCl ($4 \times 10^7$ nuclei per ml) and incubated overnight at 4 °C for the extraction of histones. To pellet cell debris, samples were centrifuged at 6500 x *g* for 10 min at 4 °C. Finally, the resulting supernatant containing histone proteins was neutralized with 2 M NaOH at 1/10th the volume of the supernatant. The concentration of all extracted proteins was determined using Pierce BCA protein assay kit (Thermo Fisher Scientific, 23250).

Protein samples, suspended in 1x Laemmli sample buffer (Biorad, 1610747) supplemented with 2-mercaptoethanol (Sigma-Aldrich; M6250) at a final concentration of 2.5%, were loaded into the wells of 3-8% Tris-acetate protein gels (Thermo Fisher Scientific, EA0378BOX) for the separation of large molecular weight proteins and 4-12% Bis-Tris protein gels (Thermo Fisher Scientific, NP0321BOX) for the separation of medium to low molecular weight proteins. Following sodium dodecyl sulfate-polyacrylamide gel electrophoresis (SDS-PAGE), proteins were transferred to polyvinylidene fluoride (PVDF) membranes (Sigma-Aldrich, GE10600021) using Trans-Blot Turbo Transfer System. Membranes were then blocked with 5% non-fat dry milk (Santa Cruz Biotechnology, sc-2325) in 1x Tris-buffered saline with 0.1% Tween 20 (TBST) for 1 h at room temperature with agitation followed by over-night incubation at 4 °C with primary antibodies against SMAD2/3 (Cell Signaling Technology, 3102S; 1:2000), p-SMAD2 (Ser465/Ser467) (Thermo Fisher Scientific, 44-244 G; 1:1000), p-SMAD3 (Ser423/Ser425) (Abcam, ab52903; 1:2000), SMAD4 (Proteintech, 10231-1-AP; 1:2000), PHF5A (Proteintech, 15554-1-AP; 1:2000), PHF14 (Proteintech, 24787-1-AP; 1:2000), α-Tubulin (Proteintech, 66031-1-Ig; 1:6000), Lamin B1 (Proteintech, 12987-1-AP; 1:2000), HMG20A (Proteintech, 12085-2-AP; 1:2000), RAI1 (Abcam, ab86599; 1:2000), TCF20 (Novus Biologicals, NBP2-83631; 1:1000), KMT2A (Proteintech, 29278-1-AP; 1:1000), RNA polymerase II CTD repeat YSPTSPS (Abcam, ab26721; 1:2000), H3K4me2 (Abcam, cat. ab32356, dilution 1:2000), H3K4me3

(Abcam, ab213224; 1:2000), Histone H3 (Abcam, ab1791; 1:6000), and WDR5 (Proteintech, 15544-1-AP; 1:2000). To remove residual primary antibody, membranes were washed 3 times, 5 min each, with 1x TBST then incubated with anti-rabbit (Sigma-Aldrich, A0545; 1:10,000) or anti-mouse (Sigma-Aldrich, A9044; 1:10,000) horseradish peroxidase (HRP)-conjugated secondary antibodies for 1 h at room temperature with agitation. Following the final washing steps (3 times with 1x TBST, 5 min each), blots were incubated with a luminol reagent (Santa Cruz Biotechnology, sc-2048) for 1 min at room temperature. Generated signals were captured on x-ray films (Thermo Fisher Scientific, 34089) and processed using AGFA Curix 60 x-ray film processor (Photon Surgical Systems Ltd). Uncropped and unprocessed scans of western blots are provided in the source data file.

### Cellular fractionation

Cytoplasmic, nuclear, and membrane proteins were extracted using a cell fractionation kit (Cell Signaling Technology, 9038) as per the manufacturer's instructions.

### ChIP-sequencing (ChIP-seq) and ChIP-quantitative PCR (ChIP-qPCR)

$ABCG2^+$ L3.6pl tumorspheres were dissociated into single cells using TrypLE Express enzyme (Thermo Fisher Scientific, 12-604-021) prior to fixation with 36% formaldehyde solution (Sigma-Aldrich, 47608) at a final concentration of 1% for 15 min at room temperature with agitation. Formaldehyde was then quenched with 2.5 M glycine (final concentration of 125 mM) for 5 min at room temperature with agitation. Cells were washed 3 times with ice-cold PBS, 5 min each, then suspended in ChIP lysis buffer (50 mM 4-(2-hydroxyethyl)-1-piperazineethanesulfonic acid-potassium hydroxide (HEPES-KOH) pH 7.5, 140 mM NaCl, 1 mM EDTA pH 8, 0.1% sodium deoxycholate, 0.1% SDS, and 1% Triton X-100) freshly supplemented with 1x EDTA-free protease inhibitor cocktail (Roche, 04693159001) and 1x phosphatase inhibitor cocktail (Roche, 4906845001). 20 cycles (30 s on/30 s off) were used to shear chromatin at 4 °C using Bioruptor Pico sonication device (Diagenode). Fragmentation of sheared chromatin was assessed on 1.5% agarose gel in 1x Tris-acetate-EDTA (TAE) buffer (40 mM Tris, 20 mM glacial acetic acid, and 1 mM EDTA). A total of 10 ug of sheared chromatin per ChIP was incubated with each of the following antibodies: anti-PHF5A (Proteintech, 15554-1-AP; 1:200), anti-PHF14 (Proteintech, 24787-1-AP; 1:200), and anti-KMT2A (Proteintech, 29278-1-AP; 1:200) for 2 h at 4 °C with agitation, followed by overnight incubation with Pierce ChIP-Grade Protein A/G Plus agarose beads (Thermo Fisher Scientific, 26159) at 4 °C with agitation. Agarose beads were then washed 3 times, 5 min each, with ChIP wash buffer (0.1% SDS, 1% Triton X-100, 2 mM EDTA, 20 mM Tris-HCl pH 8.0, and 150 mM NaCl). DNA was eluted using ChIP elution buffer (1% SDS and 100 mM NaHCO3) at 65 °C for 30 min with agitation at 1000 rpm. To reverse cross-linking and degrade RNA, DNA samples were incubated overnight at 65 °C with 5 M NaCl and RNAse A (Active Motif, 53040) with agitation at 400 rpm. Proteinase K (Active Motif, 53040) was then added to digest proteins at 60 °C for 1 h with agitation at 400 rpm. ChIPped DNA was purified with a Qiaquick PCR purification kit (Qiagen, 28104) and quantified by Qubit 2.0 fluorometer (Thermo Fisher Scientific) using Qubit 1x dsDNA HS assay kit (Thermo Fisher Scientific, Q33230). Library preparation and quality control were both conducted at CD Genomics (New York, USA) along with ChIP-sequencing of biological duplicates per each ChIP experiment and input samples using Illumina NovaSeq6000 S4 and BCL convert v4.0.3 software.

For ChIP-qPCR analysis, the reaction mixture contained Power SYBR Green PCR master mix (Thermo Fisher Scientific, 4367659), either input or ChIPped DNA, *PAK3* primer pair with the following sequences: forward; 5′-CTGTGCCACGTCTGAAGAACA-3′ and reverse; 5′-GCCTTAGAACACCAAATGGGC-3′, and nuclease-free water. ChIP-qPCR analysis was performed using LightCycler 480

System (Roche). Percent (%) of input was used for ChIP-qPCR data normalization.

### Small molecule compound screening

FG cells were seeded in biological triplicate per each screened compound at a density of $10^4$ cells/well of a 96-well plate. Small molecule compounds (Supplementary Table 1) were added individually to each well, including OICR-9429 (Sigma-Aldrich, SML1209). The cells were cultured for 5 days in a humidified cell culture incubator at 37 °C and 5% $CO_2$, with fresh compounds added on the second and fourth day of the culture. To determine the effect of drug treatment on the percentage (%) of SSEA4-expressing CSCs in the total cell population by flow cytometry, cells were washed with 1% BSA/PBS and then incubated with primary antibodies, including Alexa Fluor 647 anti-SSEA4 (BD Biosciences, 560796; 1:40) and its isotype control Alexa Fluor 647 Mouse IgG3, k (BD Biosciences, 560803; 1:40) for 40 min on ice, followed by washing 3 times with 1% BSA/PBS. For the determination of cell viability, cells were suspended in 1% BSA/PBS supplemented with DAPI (0.5 μg/ml final concentration). Flow cytometry data were acquired using BD LSRFortessa cell analyzer (BD Biosciences), collected with FACSDiva software (BD, Version8.0.1), and analyzed using FlowJo v10.8 software (BD Life Sciences). The total number of pancreatic tumorspheres (100 μm colony diameter) formed by DMSO (Sigma-Aldrich, D8418) and small molecule inhibitor-treated FG cells were counted using Celigo Imaging Cytometer (Nexcelom Bioscience, Version 2.1).

### OICR-9429 concentration curve

FG cells were seeded in biological triplicate in 96-well plates and treated with different concentrations of OICR-9429 (0–200 μM) for 5 days in a humidified cell culture incubator at 37 °C and 5% $CO_2$. The culture medium was replaced every 48 h with fresh medium supplemented with OICR-9429. A FluoStar Omega microplate reader (BMG Labtech) was used to measure the number of live cells stained with PrestoBlue cell viability reagent (Thermo Fisher Scientific, A13261) as per the manufacturer's instructions.

### Biotin-modified histone peptide pull-down assay

Streptavidin agarose beads (Sigma-Aldrich, 16-126) were prewashed with binding buffer (50 mM Tris pH 7.5, 150-200 mM NaCl, and 0.05-0.1% Non-idet P-40) then incubated with biotinylated H3K4me2 (Active motif, 81041) and H3K4me3 (Active motif, 81042) peptides for 2 h at 4 °C with agitation. To remove unbound peptides, beads were washed 3 times with the binding buffer. Nuclear protein extracts from $ABCG2^+$ L3.6sl cells were incubated with streptavidin agarose beads-biotinylated H3K4me2 and H3K4me3 peptides conjugates overnight at 4 °C with agitation. Following 3 washes with the binding buffer, 1x Laemmli sample buffer (Biorad, 1610747) supplemented with 2-mercaptoethanol (Sigma-Aldrich, M6250) at a final concentration of 2.5% was added to the beads and boiled at 95 °C for 5 min to elute co-bound proteins for western blot analysis.

### Limiting dilution assay

Serial two-fold dilutions of $ABCG2^+$ L3.6sl cells were seeded in ultra-low attachment 6-well plates in serum-free, stem cell-conditioned culture medium and treated with either DMSO (Sigma-Aldrich, D8418) or MM-102 (Tocris Bioscience, 5307) at a concentration of 75 μM for 5 days in a humidified cell culture incubator at 37 °C and 5% $CO_2$. Following treatment, pancreatic tumorspheres (100 μm colony diameter) were counted using Celigo Imaging Cytometer (Nexcelom Bioscience, Version 2.1).

### RNA-sequencing (RNA-seq)

Total RNA was extracted with Trizol reagent (Thermo Fisher Scientific, 15596018) and purified using a Direct-zol RNA miniprep kit (Zymo

Research, R2050) according to the manufacturer's instructions. Ribosomal RNA (rRNA) was depleted using NEBNEXT rRNA Depletion Kit v2 (Human/Mouse/Rat) (New England Biolabs, E7400S) as per the manufacturer's instructions. RNA-seq libraries were prepared using NEBNEXT Ultra II Directional RNA Library Prep Kit for Illumina (New England Biolabs, E7760S) according to the manufacturer's instructions. The fragment size distribution of cDNA libraries was determined using Agilent 4150 TapeStation system (Agilent Technologies). RNA-sequencing of 3 biological replicates per each treatment condition was performed at CD Genomics (New York, USA) using an Illumina Nova-Seq6000 S4 sequencing platform and BCL convert v4.0.3 software.

## Bioinformatics analysis

**ChIP-seq analysis.** Raw reads were cleaned using fastp v0.23.2 with default parameters[45]. Cleaned reads were then confirmed high-quality using FastQC v0.11.9[46]. Burrows-Wheeler Aligner v0.7.17[47] was used to map cleaned reads to the human genome hg38. Duplicated reads, reads mapped to the ENCODE blacklisted regions[48], and reads with a mapping quality lower than 30 were removed, and only properly paired reads were retained. Peaks were then called using MACS2 v2.2.7.1[49]. Common peaks within replicates were annotated with genes located within 5 kb upstream to 3 kb downstream of the gene body using ChIPpeakAnno v3.30.0[50]. Differential binding analysis was performed using DiffBind v3.6.1[51], and peaks with adjusted $p$-value < 0.01 were considered significant.

**RNA-seq analysis.** Raw reads were cleaned using fastp v0.23.2 with default parameters[45]. Cleaned reads were confirmed high-quality using FastQC v0.11.9[46] and then mapped to the human genome hg38 using STAR v2.7.3a[52]. Mapped reads were quantified using featureCounts v2.0.0[53] and analyzed using DESeq2 v1.34.0[54]. For differential gene expression analysis, the effect size was shrunk using the ashr method[55], and $p$-values were adjusted using the independent hypothesis weighting (IHW) method[56]. Significant differential gene expression was defined as |log2 Fold change| > 1 for upregulated genes and |log2 Fold change | <−1 for downregulated genes at an adjusted $p$-value < 0.05.

**Functional enrichment analysis.** For RNA-seq data, GSEA was performed using fgsea v1.22.0[57]. The gene sets used in GSEA were obtained from the Molecular Signatures Database v7.5.1[58]. RNA-seq and ChIP-seq over-representation analyses was performed using g:GOSt[59].

## Molecular docking
Human PHF5A 3D protein structure was measured using the X-ray diffraction method[60]. Since there is no standalone 3D protein structure for PHF14, the corresponding molecular structure was predicted using AlphaFold2[61]. The protein-protein docking was simulated using HDOCK[19], and the predicted structures with the lowest possible binding energy were selected.

## Statistical analysis
Statistical analysis was performed using GraphPad Prism 8 software. Data are either presented as mean ± SEM or mean ± SD. P values were calculated using either a two-tailed Student's $t$ test with Welch's correction in case of significantly different variances as analyzed with the F-test or two-way ANOVA with multiple comparisons with Tukey correction.

## Schematic illustrations
All schematic illustrations were created with Biorender.com

## Reporting summary
Further information on research design is available in the Nature Portfolio Reporting Summary linked to this article.

## Data availability
Raw and processed ChIP-seq and RNA-seq data (BioProject PRJNA887833) are available in the Sequence Read Archive (SRA) under accession code SRP403939 and the Gene Expression Ominbus (GEO) under accession code GSE217332, with no restrictions on data availability. The mass spectrometry proteomics data have been deposited to the ProteomeXchange Consortium via the PRIDE [1] partner repository with the dataset identifier PXD038378 (Project Webpage: ; FTP Download: ftp://ftp.pride.ebi.ac.uk/pride/data/archive/2023/08/PXD038378). The remaining data are available within the Article, Supplementary Information or Source Data file. Source data are provided with this paper.

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

## Acknowledgements

This study was funded by Cancer Research UK Career Development Fellowship (No. C59392/A25064, SP). We acknowledge the European Association for cancer research (EACR) for providing financial support through a travel fellowship awarded to MAM to conduct preclinical experiments at the University of California Davis, USA. This work was also supported by the China Scholarship Council - University of Oxford

Scholarship, SFF2122_CSCUO_1284663 (No. 202108330024, SD). Animal studies were funded by the National Cancer Institute (No. R37CA29007, CH). We thank Mike Deery and Yagnesh Umrania from the Cambridge Centre for Proteomics, Cambridge, UK for their assistance with the analysis of mass spectrometry data. We also thank Suyakarn Archasappawat from the University of California Davis, USA for providing technical assistance with the animal experiments.

## Author contributions

M.A.M.: Conceptualization, data curation, formal analysis, validation, investigation, visualization, methodology, writing the original draft of the manuscript, and manuscript revision; S.D.: Bioinformatic analysis, data deposition, manuscript revision, and writing review; M.P.: Methodology and formal analysis; J.M.: Methodology; A.R.: Formal analysis; S.M.: Methodology; R.N.: Bioinformatic analysis; Z.S.: Methodology; U.O.: Supervision; C.H.: Methodology, resources, and provided suggestions and comments on the manuscript; and S.P.: Supervision, methodology, resources, manuscript revision and writing review.

## Competing interests

The authors declare no competing interests.
