## [Peer Review File · Nature Communications]

KMT2A associates with PHF5A-PHF14-HMG20A-RAI1 subcomplex in pancreatic cancer stem cells and epigenetically regulates their characteristics

REVIEWER COMMENTS

Reviewer #1, expertise in PCSCs (Remarks to the Author):

Cancer stem cells (CSC) are critical players in tumor initiation, progression, recurrence, and metastasis. Several biomarkers for detecting CSC populations in pancreatic cancer have been found. Several studies were conducted to investigate the processes of CSC maintenance. This study focused on identifying the mechanism of histone methyltransferase KMT2A and its subcomplex formation in pancreatic CSC maintenance. The established pancreatic CSCs were validated and used to identify PHF5A binding partners to maintain stemness potential. They identified RNA polymerase-associated PHF5A-PHF14-HMG20A-RAI1-KMT2A transcriptional subcomplex involved in the maintenance of pancreatic CSCs. Specially KMT2A showed self-renewal and tumorigenicity of pancreatic CSCs. The KMT2A inhibitor MM-102 showed decreased tumorigenicity and stemness potential of pancreatic CSCs. This manuscript is well-written; however, several conceptual and technical questions must be answered.

- The 3D model of the CSC establishment depicted a population pool of CSCs. According to current research, numerous heterogeneous populations of CSC have distinct mechanisms for self-renewal and tumorigenicity. Identify this PHF5A-PHF14-HMG20A-RAI1-KMT2A complex in distinct CSC populations (CD44+, CD133+, Alde-flour, and SP).
- Previous studies have found that PHF5A interacts with a number of subcomplexes in pluripotent stem cells and pancreatic cancer stem cells. How the PHF5A-PHF14-HMG20A-RAI1-KMT2A complex differs from earlier mechanisms for pancreatic CSC maintenance.
- This work shows that the PHF5A-PHF14-HMG20A-RAI1-KMT2A-mediated pathway is important in pancreatic cell line models. Using multiplexed immunofluorescence, demonstrate the expression of PHF5A-PHF14-HMG20A-RAI1-KMT2A complex in pancreatic cancer tissues solely in the CSC population.
- What is the pattern of KMT2A expression in pancreatic tumor samples?
- All of the self-renewal biomarkers were examined at the mRNA level. The self-renewal markers showing considerable variance may have been evaluated at the protein level.

- The self-renewal of CSC populations was demonstrated using a serial dilution or limited dilution assay. KMT2A inhibitor treatment and serial dilution experiment is required to demonstrate its impact on pancreatic CSC self-renewal.

Reviewer #2, expertise in epigenetic regulation in pancreatic cancer (Remarks to the Author):

In this manuscript, Mai et al. investigate the molecular mechanisms that maintain the unique characteristics of pancreatic cancer stem cells. They first establish culture conditions to generate CSCs-enriched pancreatic tumorspheres with four pancreatic cancer immortalized cell lines, L3.6pl, L3.6sl, A13A, and A13B. Using large-scale proteomic assay, they identify a physical association between PHF5A and PHF14 in PCSCs. Subsequently, they reveal a transcriptional subcomplex, PHF5A-PHF14-HMG20A-RAI1, as a regulator of PCSC characteristics. In the second part of the study, the authors use a small molecule compound screening platform and demonstrate that inhibition of KMT2A decreases SSEA4-expressing cells and tumorsphere formation. Since KMT2A is co-immunoprecipitated PHF5A, the authors propose that KMT2A-mediated H3K4me3 which is recognized by the PHF5A-PHF14-HMG20A-RAI1 protein subcomplex. The manuscript is clearly written and most of the conclusions are supported by reasonable experimental evidence. Although the interaction between PHF5A and PHF14 is demonstrated with Co-IPs coupled with LC/MS-MS analysis, the question of whether the PHF5A-PHF14-HMG20A-RAI1-KMT2A protein subcomplex functions as an epigenetic regulator in pancreatic cancer stem cells is interesting. Unfortunately, the authors fail to provide enough experimental evidence to support this claim. Several key points require clarification or additional evidence to make the work suitable for publication. In particular, the physical interaction between PHF5A and PHF14 needs to be verified and characterized. Furthermore, the inhibitory effect of KMT2A is unique to suppress PCSCs is unconvincing.

Major comments:

1. The authors propose that PHF14 is a novel binding partner of PHF5A in PCSCs based on LC-MS/MS analysis of PHF5A Co-IP in ABCG2+ L3.6pl tumorspheres. Even though protein-protein docking with HDOCK favors a potential physical association between PHF14 and PHF5A, none of the experiments and analyses address whether the interaction is specific to PCSCs and whether the interaction is a direct physical interaction. Given that PHF5A and PHF14 belong to the PHD-finger superfamily and may act as a chromatin-associated protein, it is possible that PHF5A and PHF14 simply bind to the same chromatin regions rather than interact specifically with each other. A Nuclease (like benzonase) treatment should be included to support the physical interaction. A more specific measurement (like Isothermal titration calorimetry) to quantify the interaction between PHF5A and PHF14 will also support their conclusion. In

addition, such interaction should also be evaluated in the monolayer cultures which show no enrichment of PCSCs.

2. ChIP-seq for PHF5A and PHF14 resulted in less than 300 targeted regions being identified. It would be important to show quality controls for the specificity of the PHF5A and PHF14 antibodies.

3. It is well demonstrated that KMT2A mediates H3K4me3 and actively associates with RNA polymerase II. It is therefore expected to find that inhibition of KMT2A with either OICR-9429 or MM-102 can lead to decreasing tumorsphere formation and increasing cell death. The authors make the argument that a physical association between PHF5A-PHF14-HMG20A-RAI1 protein subcomplex and KMT2A, in which KMT2A enzyme catalyzes methylated histone marks recognized by PHF5A-PHF14-HMG20A-RAI1 protein subcomplex members. This claim is not supported by examining changes in PHF5A/PHF14 deposition upon MM-102 treatment (Supplementary figure 3e). Furthermore, the ChIP-seq for KMT2A identifies only 400 targets which is much less than the targets demonstrated in other studies. It would be important to show quality controls for the KMT2A antibody.

Minor points:

1. The authors examine the enrichment of CSC surface markers in pancreatic tumorspheres. Even if ABCG2 shows higher expression levels in tumorsphere cultures, other typical cancer stem cell markers, such as CD24, are expressed much less in the tumorsphere culture than in the monolayer cultures for L3.6pl and L3.6sl lines. The enrichment of ABCG2 could simply be a reflection of the cultural conditions. Therefore, it is important to determine whether the selected ABCG2+ cells are indeed functionally cancer stem cells. For example, whether the ABCG2+ are more enriched in tumor-propagating cells compared with monolayer culture cells in transplantation assays. Do the ABCG2+ cells present higher self-renewal properties compared with monolayer culture cells in serial transplantations?

2. Specific interaction of KMT2A with PHF5A-PHF14-HMG20A-RAI1 protein subcomplex (Figure h and i) needs to be shown more convincingly. It is unclear if this interaction is direct and specific toward KMT2A in PCSCs. KMT2A is likely to interact with multiple proteins, so it would be expected to see KMT2A interact with PHF5A-PHF14 in general. Co-IPs with KMT2A, are needed to explore the specificity in PCSCs vs. monolayer cultures.

Reviewer #3, expertise in PCSCs and mass-spec (Remarks to the Author):

The manuscript "Identification of an RNA polymerase-associated PHF5A-PHF14-HMG20A-RAI1-KMT2A protein subcomplex as a novel epigenetic target for attenuating the stemness and tumorigenicity of pancreatic cancer stem cells" of M. Abdel Mouti et al. deals with a relevant topic, indeed the epigenetic targeting represent a new emerging area of research. The work is original and obtained results suggest a new potential epigenetic target for the treatment of pancreatic cancer. The methodology used is rigorous and results support the conclusion, however, the paper needs minor revision. In particular:

- Introduction section, it would be useful to mention the differences between PDAC e PASC, since the used in vitro models belong to these PC neoplasms.
- Results section, "Small molecule screening ..." paragraph, it is not clear why a different type of cells, i.e. FG cells, were used for the small molecule screening; authors should comment on this.
- Results section, "Optimizing the platform for characterizing the transcriptional ..." the expression of intracellular self-renewal and pluripotency genes in the enriched CSCs (lane 194) should be compared with that of the sorted cells having no CSC surface markers, which represent a better control than HPDE6c7.
- Methods section, "Cell Lines and Cell Culture" paragraph, it would be useful to specify which is the genetic background of all the used cell lines, in particular as refereed to the typical mutated/deleted genes of pancreatic cancer (i.e. K-ras, p53, p16, DPC4). Moreover, culture of FG cells should be mentioned.
- Methods section, "LC-MS/MS Analysis" paragraph, is should be described the method used for the normalized total spectral counts.
- Methods section, Lane 903: only OICR-9429 is indicated, whilst in into the result section (lane 326) it is mentioned a library of 142 compounds. The library which was used should be indicated into the method.

Reviewer #4, expertise in Small Molecule Chemical Screening (Remarks to the Author):

This manuscript, by Mouti et al, describes the identification of a potential sensitivity of pancreatic cancer stem cells by way of an RNA polymerase complex that includes KMT2A. Overall, I found the manuscript interesting and well-written and I believe that it will be of interest to the community.

This submission features a large amount of data used to characterize stem cells derived from pancreatic tumor cell lines and to define epigenetic complexes related to PHF5A that are associated with the stem-like character of these cells. The data presented here are somewhat outside of this reviewer's expertise, so no specific comments on this

section will be provided. However, I did find the logic and results of the experiments reasonably clear.

The authors then present a screen of known epigenetic regulators, notably evaluated using several methods in parallel, to identify the WDR5-binding compound OICR-9429 as effective in decreasing expression of a stem cell marker. Additional work in the manuscript using an alternative WDR5-binding compound MM-102 is presented, including an in vivo study. My review of this paper focuses mainly on aspects of the use of the small molecules.

Relatively minor issues include the following:

- Small molecule screening is performed in FG cells (page 9 and methods), but FG is not defined.
- No references in the text are not provided for either the OICR or MM molecule (more on this below).

Larger issues are noted as follows:

1. The authors should clarify certain mechanistic aspects of how the WDR5 inhibitor function in this context. Ongoing work by the Tansey lab (e.g., 2022 Scientific Reports) suggests that inhibition of histone methylation is not the most relevant driver of the pharmacological

effects of WDR5 binding compounds, which would include OICR-9492 and MM-102. Thus, additional support of the role of the WDR5 binding compounds in inhibiting KMT2A leading to the observed phenotypes should be provided:

a. Unlike KMT2A, WDR5 was not identified in the PHF5A and related complexes. While it is, of course, known that WDR5 inhibitors can affect KMT2A function (see manuscript reference 42, for example), for them to do so in this context, it seems to this reviewer that WDR5 should have been a component of the active complexes.

b. Some data should be referenced or added to clarify the ability of the compounds to inhibit the HMT action of KMT2A. I am unable to find direct inhibition info on OICR-9429 for example, and no data is presented in the current work to validate the KMT2A inhibition of this compound and its association with the effects in the screen. MM-102 is reported (Wang, et al 2013 J.Amer.Chem.Soc) to inhibit KMT2A/MLL1 with an IC₅₀ of 400 nM in biochemical assays. It is notable, based on that 400 nM number for MM-102, that quite high doses of the compound are needed to elicit an effect on histone methylation (Figure 6 b) in this manuscript. What is the source of this drop-off? Many agents display a higher effective dose moving from in vitro data, but 400 nM → 75 μM implies a significant barrier to cellular action (excessive protein binding, cellular permeability, efflux, etc). This is relevant, as it seems that this data drove selection of doses in additional phenotypic experiments. It is also relevant, as these high doses raise the possibility of confounding effects not driven by inhibition of the WDR5-KMT2A interface.

c. The use of complementary in vitro data with a superior molecule compared with MM-102 is highly encouraged. OICR-9429 is a well-regarded probe for in vitro experiments regarding WDR5 (see chemicalprobes.org), but I could only find more limited information on MM-102. References or data should be supplied to support the specificity of the MM-102 compound. Additionally, repeating some of the in vitro work with the OICR compound or perhaps the more completely characterized MM-401 compound from reference 42 would lend weight to the role of a WDR5 inhibitor in this work. Without such data, the prospect of additional off-target effects to contribute to the action of MM-102 at the high doses used has to be considered.

d. This reviewer notes that a direct inhibitor of the enzymatic action of KMT2A, if such is available, could be used to further differentiate the potential mechanisms (e.g., the HMT activity vs potential scaffolding functions) and add clarity.

2. Certain aspects of the in vivo work could also use additional context:

a. No pharmacokinetic data are presented or referenced to support the use of MM-102 in vivo. Since relatively high concentrations (50-75 μ M) appear to be needed in cellular experiments, do the exposures obtained in the animal studies approach these levels? Given the lack of difference in 30 and 50 mg/kg/day doses of the molecule (Figure 6l), some context with respect to drug concentrations achieved in blood or tumors is needed.

i. Indeed, this reviewer suggests that an alternative inhibitor molecule might be used for further follow-ups for in vivo (note that this may not be necessary for the current paper). OICR-9429 appears to be ill-suited for in vivo work (chemicalprobes.org). However, additional WDR5 inhibitors have been reported, at least some of which feature defined pharmacokinetics and even oral availability (e.g., Fesik, et al 2022 J. Medicinal Chem.).

b. Assessing the level of inhibition of KMT2A activity achieved in the tumors would better associate the role of KMT2A with the in vivo phenotypic observations.

In summary, the identification of a potentially new sensitivity of pancreatic cancer could have a significant impact, given the severity of the disease. However, the choice of a relatively poor-quality molecule to evaluate the potential of this target opportunity somewhat mutes the impact. Additional data would significantly improve the manuscript and add weight to the conclusions of the authors. This data might include some or all of the following:

- Confirmatory in vitro data with a higher quality small molecule
- Additional biological experiments that correlate with the small molecule results. Off the top of my head, these might include for example mutations that abrogate the histone methylase activity of KMT2A or inhibit its interaction with WDR5.
- Target engagement data in vivo to associate the inhibition of tumor growth after dosing with MM-102 with inhibition of KMT2A.
- Pharmacokinetic data to indicate that the concentrations of MM-102 achieved with in

vivo dosing are sufficiently above the effective KMT2A inhibition concentrations in vitro to correlate with the observed effects.

General Response

We would like to thank the Reviewers for their helpful feedback and insightful suggestions. Responses from the Reviewers have been encouraging and constructive. We provide here below a point-by-point response to the Reviewers' comments, accompanied by additional experimental data and analyses. We have incorporated the suggested information as blue-colored text in the revised manuscript. We have also updated the figures by integrating the new data requested. Collectively, the additions recommended by the Reviewers have further improved the timeliness, impact, and breadth of our manuscript.

A point-by-point response to Reviewers' comments

Reviewer #1, expertise in PCSCs:

Cancer stem cells (CSC) are critical players in tumor initiation, progression, recurrence, and metastasis. Several biomarkers for detecting CSC populations in pancreatic cancer have been found. Several studies were conducted to investigate the processes of CSC maintenance. This study focused on identifying the mechanism of histone methyltransferase KMT2A and its subcomplex formation in pancreatic CSC maintenance. The established pancreatic CSCs were validated and used to identify PHF5A binding partners to maintain stemness potential. They identified RNA polymerase-associated PHF5A-PHF14-HMG20A-RAI1-KMT2A transcriptional subcomplex involved in the maintenance of pancreatic CSCs. Specially KMT2A showed self-renewal and tumorigenicity of pancreatic CSCs. The KMT2A inhibitor MM-102 showed decreased tumorigenicity and stemness potential of pancreatic CSCs. This manuscript is well-written; however, several conceptual and technical questions must be answered.

Author response_Reviewer 1_ General comment:

We thank the reviewer for recognizing our work. Likewise, we appreciate the reviewer's insightful comments and remarks, which we aimed to address as adequately as possible in our revised manuscript.

Reviewer 1_Comment 1:

The 3D model of the CSC establishment depicted a population pool of CSCs. According to current research, numerous heterogeneous populations of CSC have distinct mechanisms for self-renewal and tumorigenicity. Identify this PHF5A-PHF14-HMG20A-RAI1-KMT2A complex in distinct CSC populations (CD44+, CD133+, Aldefluor, and SP).

Author response_Reviewer 1_Comment 1:

We thank the reviewer for this suggestion. We have isolated CD44⁺ and CD133⁺ CSC populations from L3.6sl and A13A monolayer cells, respectively, using a magnetic-activated cell sorting (MACS) technique. This was followed by nuclear PHF5A Co-IP in CD44⁺ L3.6sl and CD133⁺ A13A tumorspheres and western blot analysis which revealed the physical association between PHF5A, PHF14, HMG20A, RAI1, and KMT2A in both CSC populations, providing further support for the role of PHF5A-PHF14-HMG20A-RAI1-KMT2A protein subcomplex in PCSCs. We have added this information to Supplementary Fig. S3f, g and the Results section (page 13, lines 430-436) of our revised manuscript.

Supplementary Fig. S3f, g

Western blot analyses of PHF5A Co-IP in nuclear protein extracts from CD44⁺ L3.6sl (**f**) and CD133⁺ A13A (**g**) tumorspheres.

To validate our MACs-sorted PCSCs, Aldefluor and Hoechst 33342 exclusion assays were conducted along with gene expression analysis of self-renewal and pluripotency genes. In both assays, however, we have not observed significant enrichment for aldehyde dehydrogenase-bright (ALDH^{br}) or side-population (SP) cells in PCSCs versus their parental monolayers. Given these results, we did not specifically study the formation of the protein subcomplex in ALDH^{br} or SP cells. We have added this information to the Discussion section (page 20, lines 664-670) of our revised manuscript.

Based on our PCSC marker characterization experiments (Revised manuscript, Fig. 1b-e), we also need to highlight the following:

In ABCG2⁺ L3.6pl and L3.6sl tumorspheres:

- CD44 is expressed in all cells. This is illustrated by flow cytometry dot plots of CD44 enrichment in L3.6pl (100%; Rebuttal Fig. 1a) and L3.6sl (99.7%; Rebuttal Fig. 1b) tumorspheres.
- EpCAM is expressed by all cells (100%) of L3.6pl (Rebuttal Fig. 1c) and L3.6sl (Rebuttal Fig. 1d) tumorspheres.

Hence, CSC-enriched L3.6pl and L3.6sl tumorspheres used for our proteomic, genomic, transcriptomic, and functional analyses could be phenotypically identified as ABCG2⁺ CD44⁺ EpCAM⁺ cancer stem-like cells. In the present study, we referred to those cells as ABCG2⁺ since their respective parental monolayer cells were sorted by MACs based on their surface expression of ABCG2 which showed significant enrichment in L3.6pl and L3.6sl tumorspheres as compared to L3.6pl and L3.6sl monolayers in our flow cytometry screening experiment (Revised Manuscript, Fig. 1b, c).

Rebuttal Fig. 1

a-d, Flow cytometry dot plots of CD44 enrichment in L3.6pl (**a**) and L3.6sl (**b**) tumorspheres and EpCAM enrichment in L3.6pl (**c**) and L3.6sl (**d**) tumorspheres.

In SSEA4⁺ A13A and A13B tumorspheres:

- CD133 is expressed by all cells, as shown in the flow cytometry dot plots of CD133 enrichment in A13A (99.8%; Rebuttal Fig. 2a) and A13B (99.7%; Rebuttal Fig. 2b) tumorspheres.
- EpCAM is expressed by all cells (100%) of A13A and A13B tumorspheres (Rebuttal Fig. 2c, d).
- CD24 is expressed by nearly all cells (99.8%) of A13A tumorspheres (Rebuttal Fig. 2e) and the vast majority of A13B tumorsphere cells (94.4%; Rebuttal Fig. 2f).

Therefore, CSC-enriched A13A and A13B tumorspheres used in our analyses could be phenotypically identified as SSEA4⁺ CD133⁺ EpCAM⁺ CD24⁺ cancer stem-like cells. Those cells were referred to as SSEA4⁺ since their respective parental monolayer cells were MACs sorted based on their surface expression of SSEA4 which showed significant enrichment in A13A and A13B tumorspheres as compared to A13A and A13B monolayer cells in our flow cytometry screening experiment (Revised manuscript, Fig. 1d, e).

Rebuttal Fig. 2

a-f, Flow cytometry dot plots of CD133 enrichment in A13A (**a**) and A13B (**b**) tumorspheres, EpCAM enrichment in A13A (**c**) and A13B (**d**) tumorspheres, and CD24 enrichment in A13A (**e**) and A13B (**f**) tumorspheres.

Reviewer 1_Comment 2:

Previous studies have found that PHF5A interacts with a number of subcomplexes in pluripotent stem cells and pancreatic cancer stem cells. How the PHF5A-PHF14-HMG20A-RAI1-KMT2A complex differs from earlier mechanisms for pancreatic CSC maintenance?

Author response_Reviewer 1_Comment 2:

In hESCs, PHF5A associates with components of the PAF1 complex, including PAF1, LEO, CTR9, and CDC73. PHF5A plays an important role in regulating the stability of PHF5A-PAF1-LEO-CTR9-CDC73 complex and its recruitment to the regulatory regions of pluripotency genes. In addition, PHF5A promotes RNA Pol II pause release and transcriptional elongation of self-renewal and pluripotency-associated genes (Strikoudis, A. et al. *Nat. Cell Biol.* 2016, PMID 27749823).

In PCSC subpopulations studied by Karmakar et al, however, PHF5A does not interact with LEO1, CTR9, or CDC73, but is instead part of a protein subcomplex containing PAF1 and DDX3. Inhibiting the activity of DDX3 by RK-33, a small molecule inhibitor of the helicase activity of DDX3, reduced the localization of DDX3 and PAF1 on the *Nanog* promoter. Functionally, RK-33 decreased the expression of PAF1 and CSC markers in PCSCs,

but only had a minimal effect on their expression in non-CSCs. DDX3 inhibition also impaired the sphere-forming capacity of PCSCs and triggered their apoptosis, with less pronounced cell death observed in normal human fibroblasts (Karmakar, S. *et al. Gastroenterology* 2020, PMID 32781084).

Here, we report a previously unrecognized physical association between PHF5A, PHF14, HMG20A, RAI1, and KMT2A. The present study also identified KMT2A as an epigenetic regulator of PCSCs and a key member of the PHF5A-PHF14-HMG20A-RAI1-KMT2A subcomplex. Targeting the protein subcomplex with KMT2A-WDR5 inhibitors, such as OICR-9429, MM-401, and MM-201 impaired the self-renewal capacity of PCSCs and induced their apoptosis. In preclinical studies, MM-102 attenuated the tumorigenicity of PCSCs in murine xenograft models. However, our LC-MS/MS analysis of nuclear PHF5A Co-IPs in ABCG2⁺ L3.6pl tumorspheres indicates that PHF5A does not associate with any of the nuclear binding partners identified by Strikoudis *et al.* or by Karmakar *et al.*, suggesting that hESCs and different subpopulations of PCSCs maintain their stemness features and functions through unique protein-protein interactions that involve PHF5A to fulfil cell type-specific requirements and functions.

We have added this information in a shortened form to the Discussion section (page 19, lines 620-630, and page 20, lines 645-649) of our revised manuscript.

Reviewer 1_Comment 3:

This work shows that the PHF5A-PHF14-HMG20A-RAI1-KMT2A-mediated pathway is important in pancreatic cell line models. Using multiplexed immunofluorescence, demonstrate the expression of PHF5A-PHF14-HMG20A-RAI1-KMT2A complex in pancreatic cancer tissues solely in the CSC population.

Author response_Reviewer 1_Comment 3:

We thank the reviewer for this suggestion. The authors agree that demonstrating the expression of PHF5A-PHF14-HMG20A-RAI1-KMT2A protein subcomplex by multiplexed immunofluorescence in patient-derived PC tissues can further support our proteomics data of the subcomplex formation in PCSCs derived from PC cell lines. However, multiplexed immunofluorescence specifically in CSCs is limited by our ability to correctly and reliably identify the CSC population due to the lack of a universal CSC marker (Huang, R. & Rofstad,

E.K. Oncotarget. 2017, PMID 27343550). This is further supported by our flow cytometry screening data of putative PCSC surface markers which showed that enriched CSC markers differed between pancreatic tumorspheres derived from different PC cell lines. In addition, immunofluorescence microscopy does not provide a general overview of marker expression but focuses on selected subareas in the tumor tissue.

Alternatively, we used multiplexed immunofluorescence staining followed by flow cytometry which gives a broader overview of marker expression in the broader cell population. The reason for selecting immunofluorescence followed by flow cytometry also had technical advantages for us since the lab had started establishing multiplexed stainings of CSC markers by flow cytometry and was more familiar with the technical needs for conducting such analyses. Here, the tumor samples were dissociated into single cells then MACs-sorted using anti-CD45-conjugated microbeads to deplete CD45⁺ cells corresponding to leukocytes, while collecting CD45⁻ cells that include cancer cells (Figure a). Thereafter, cells were subjected to multiplexed immunostaining with fluorophore (Alexa Fluor 488, Alexa Fluor 568, or Alexa Fluor 647)-conjugated antibodies targeting cancer stem cell markers (ABCG2, CD44, and CD133) and PHF5A, PHF14 and KMT2A. The flow cytometry data (Figure b) for double and triple positive cells are shown as pseudodot plots from patient 1 PDAC. The gating in red shows the third marker in double positive cell populations. Data from both patient PDACs indicated the presence of a subpopulation of cancer cells co-expressing PHF5A, PHF14 and KMT2A in cells that are positive for different cancer stem cell markers (ABCG2, CD44, and CD133). The percentages of double-positive and triple-positive cells relative to the total cell population is shown in column charts for both PDAC patients 1 (Figure c) and 2 (Figure d). Collectively, our data indicate that PDAC tumors contain cancer stem cell marker expressing cells that also co-express PHF5A, PHF14 and KMT2A, and the frequency of these cells ranges from approximately 1% to 8% of CD45⁻ cells, depending on the tumor and markers analyzed. These data provide further support on the co-expression of the PHF5A complex factors in cells expressing markers (ABCG2, CD44, CD133) that have been shown by other publications to be associated with CSC characteristics (PMID: 17283135; 18371365; 30130664; 29320425; 30273655).

a**b**
a**b****Reviewer 1_Comment 4:**

What is the pattern of KMT2A expression in pancreatic tumor samples?

Author response_Reviewer 1_Comment 4:

The expression of *KMT2A* is significantly ($***p=8.2e^{-11}$) elevated in PDAC patients as compared to the normal pancreas. Data were obtained from TCGA PAAD (PDAC patients, $n = 183$) and GTEx (normal pancreas, $n = 328$).

The expression statistics of *KMT2A* in PDAC patient tissue (Unit = TPM; $n = 183$):

Min = 0.7058
 First Quartile = 5.9089
 Median = 7.5423
 Mean = 7.9658
 Third Quartile = 9.7584
 Max = 18.9735
 Standard Deviation = 3.257851

The expression statistics of *KMT2A* in normal pancreatic tissue (Unit = TPM; $n = 328$):

Min = 1.687
 First Quartile = 4.938
 Median = 5.998
 Mean = 6.169
 Third Quartile = 7.289
 Max = 15.080
 Standard Deviation = 2.006811

Revised Fig. 5

g, Box plot comparing gene expression levels of *KMT2A* in PDAC tumors versus normal pancreatic tissues. Data were obtained from TCGA PAAD (PDAC patients, n = 183) and GTEx (normal pancreas, n = 328) clinical datasets. Statistical analysis was performed using a two-tailed *t* test.

These data have been added to Figure 5g and the Results section (page 11, lines 374-376) of our revised manuscript.

Reviewer 1_Comment 5:

What is the justification for comparing HPDE adherent culture to pancreatic CSC 3D culture? The author must utilize the same cancer adherent cell as a control to identify the mechanism.

Author response_Reviewer_Comment 5:

We thank the reviewer for this suggestion. According to findings of Tai et al., HPDE cells retain pancreatic ductal epithelial stem cell characteristics (Tai, M.H. et al. *Pancreas* 2003, PMID 12499933), so they were used as a control in our study to assess the expression of self-renewal and pluripotency genes in MACS-sorted PCSC 3D cultures.

As suggested, we also compared the expression levels of self-renewal and pluripotency genes in CSC marker-enriched pancreatic tumorspheres to their respective adherent cultures lacking CSC enrichment. Our data show a significant increase in the expression of *SOX2*, *NANOG*, *KLF4*, *STAT3*, and *POU5F1* in ABCG2⁺ L3.6pl (Revised manuscript, Supplementary Fig. S1g) and ABCG2⁺ L3.6sl (Revised manuscript, Supplementary Fig. S1h) tumorspheres as compared to their respective ABCG2-depleted (ABCG2⁻) monolayer cells.

g**h**
Revised Supplementary Fig. S1

g, h, qRT-PCR analysis of expression levels of self-renewal and pluripotency genes in ABCG2⁺ L3.6pl (**g**) and ABCG2⁺ L3.6sl (**h**) tumorspheres as compared to their respective CSC marker-depleted adherent cultures. *ACTB* was used for the normalization of mRNA expression levels. Data are presented as the mean value \pm SEM (n = 6). P values were calculated using a two-tailed *t* test with Welch's correction for unequal variances.

However, we only observed a significant increase in the expression of *POU5F1* in SSEA4⁺ A13A (Revised manuscript, Supplementary Fig. S1i) and *KLF4* in SSEA4⁺ A13B (Revised manuscript, Supplementary Fig. S1j) tumorspheres as compared to their respective SSEA4-depleted (SSEA4⁻) monolayer cells. The elevated expression of some self-renewal and pluripotency-associated genes in SSEA4⁻ monolayer cultures versus SSEA4⁺ tumorspheres of A13A and A13B cells might be linked to their aberrant expression as oncogenic factors in non-CSC monolayer cell cultures (Seymour, T. *et al.* Int. J. Mol. Sci. 2015, PMID 26580604). Several studies have shown that CSCs and non-CSCs coexist in tumors in a dynamic equilibrium, and that both types of cells can interconvert in response to environmental factors (Marjanovic, N.D. *et al.* Clin. Chem. 2013, PMID 23220226; Yang, G. *et al.* Br. J. Cancer 2012, PMID 22472879) or upregulation of signaling cascades that promote the acquisition of cancer stem cell-like phenotypes by non-CSCs (Li, H.J. *et al.* Cancer Discov. 2012, PMID 22763855). These could potentially serve as compensatory mechanisms in A13A and A13B cells that are both derived from PDAC tumors, one of the most aggressive and lethal human cancer subtypes, in response to the depletion of the CSC population.

Revised Supplementary Fig. S1

i, j, qRT-PCR analysis of expression levels of self-renewal and pluripotency genes in SSEA4⁺ A13A (**i**) and SSEA4⁺ A13B (**j**) tumorspheres as compared to their respective CSC marker-depleted adherent cultures. *ACTB* was used for the normalization of mRNA expression levels. Data are presented as the mean value \pm SEM (n=6). P values were calculated using a two-tailed *t* test with Welch's correction for unequal variances.

We have added these data to Supplementary Fig. S1g-i and the Results section (page 7, lines 213-221) of our revised manuscript.

Reviewer 1_Comment 6:

All of the self-renewal biomarkers were examined at the mRNA level. The self-renewal markers showing considerable variance may have been evaluated at the protein level.

Author response_Reviewer 1_Comment 6:

We thank the reviewer for this suggestion. We have performed flow cytometry analysis to assess SOX2 protein expression levels in CSC marker-enriched pancreatic tumorspheres, since SOX2 is the self-renewal marker that shows the most considerable variance (significant increase) in ABCG2⁺ and SSEA4⁺ pancreatic tumorspheres as compared to HPDE6c7. In line with our gene expression data, we observed a significant increase in the mean fluorescence intensity (MFI) of SOX2, a readout of SOX2 protein expression levels, in all CSC-enriched pancreatic tumorspheres versus HPDE6c7 (Fig. 1r). Representative flow cytometry histograms

of SOX2 protein expression in ABCG2⁺ L3.6pl, ABCG2⁺ L3.6sl, SSEA4⁺ A13A, and SSEA4⁺ A13B tumorspheres versus HPDE6c7 are shown in Fig. 1s, t, u, and v, respectively.

r

s

t

u

v

Revised Fig. 1

r, Mean fluorescence intensity of SOX2 in ABCG2⁺ L3.6pl, ABCG2⁺ L3.6sl, SSEA4⁺ A13A, and SSEA4⁺ A13B tumorspheres as compared to HPDE6c7. Data are representative of 3 independent experiments and shown as the mean value \pm SEM. Statistical analysis was performed using a two-tailed *t*-test with Welch's correction for unequal variances.

s-v, Flow cytometry histograms of intracellular SOX2 protein expression in ABCG2⁺ L3.6pl (s), ABCG2⁺ L3.6sl (t), SSEA4⁺ A13A (u), and SSEA4⁺ A13B (v) tumorspheres versus HPDE6c7.

We have added these data to Fig. 1r-v, Results section (page 6, 203-212), and the Methods section (page 25, lines 831-834) of our revised manuscript.

Reviewer 1_Comment 7:

The self-renewal of CSC populations was demonstrated using a serial dilution or limited dilution assay. KMT2A inhibitor treatment and serial dilution experiment is required to demonstrate its impact on pancreatic CSC self-renewal.

Author response_REVR 1_Comment 7:

We thank the reviewer for this suggestion. We have examined the effect of KMT2A inhibition on the self-renewal capacity of PCSCs using a two-fold serial dilution assay. This showed a significant decrease (5-6 fold) in the total number of pancreatic tumorspheres formed by ABCG2⁺ L3.6sl cells (n = 3) when treated with 75 μ M MM-102 versus DMSO with increasing serial dilutions (1:2) of treated cells (Revised manuscript, Fig. 6d), suggesting an important role for KMT2A in the maintenance of PCSC stemness features.

Revised Fig. 6.

d, Sphere limiting dilution analysis to examine the sphere-forming capacity of cells treated with 75 μ M MM-102 versus DMSO. Experiments were conducted in triplicate. Statistical analysis was performed using two-way ANOVA with multiple comparisons with Tukey correction.

We have added these data to Fig. 6d, Results section (page 15, lines 475-478), and the Methods section (page 32, lines 1053-1058) of our revised manuscript.

Reviewer #2, expertise in epigenetic regulation in pancreatic cancer

In this manuscript, Mai et al. investigate the molecular mechanisms that maintain the unique characteristics of pancreatic cancer stem cells. They first establish culture conditions to generate CSCs-enriched pancreatic tumorspheres with four pancreatic cancer immortalized cell lines, L3.6pl, L3.6sl, A13A, and A13B. Using large-scale proteomic assay, they identify a physical association between PHF5A and PHF14 in PCSCs. Subsequently, they reveal a transcriptional subcomplex, PHF5A-PHF14-HMG20A-RAI1, as a regulator of PCSC characteristics. In the second part of the study, the authors use a small molecule compound screening platform and demonstrate that inhibition of KMT2A decreases SSEA4-expressing cells and tumorsphere formation. Since KMT2A is co-immunoprecipitated PHF5A, the authors propose that KMT2A-mediated H3K4me3 which is recognized by the PHF5A-PHF14-HMG20A-RAI1 protein subcomplex.

The manuscript is clearly written and most of the conclusions are supported by reasonable experimental evidence. Although the interaction between PHF5A and PHF14 is demonstrated with Co-IPs coupled with LC/MS-MS analysis, the question of whether the PHF5A-PHF14-HMG20A-RAI1-KMT2A protein subcomplex functions as an epigenetic regulator in pancreatic cancer stem cells is interesting. Unfortunately, the authors fail to provide enough experimental evidence to support this claim. Several key points require clarification or additional evidence to make the work suitable for publication. In particular, the physical interaction between PHF5A and PHF14 needs to be verified and characterized. Furthermore, the inhibitory effect of KMT2A is unique to suppress PCSCs is unconvincing.

Author response_Reviewer 2_General comment:

We thank the reviewer for providing a constructive feedback on our work. We also appreciate the reviewer's helpful remarks and suggestions for strengthening the manuscript, which have been incorporated into our revised version.

Reviewer 2_Comment 1:

The authors propose that PHF14 is a novel binding partner of PHF5A in PCSCs based on LC-MS/MS analysis of PHF5A Co-IP in ABCG2+ L3.6pl tumorspheres. Even though protein-protein docking with HDOCK favors a potential physical association between PHF14 and PHF5A, none of the experiments and analyses address whether the interaction is specific to PCSCs and whether the interaction is a direct physical interaction. Given that PHF5A and PHF14 belong to the PHD-finger superfamily and may act as a chromatin-associated protein, it is possible that PHF5A and PHF14 simply bind to the same chromatin regions rather than interact specifically with each other. A Nuclease (like benzonase) treatment should be included to support the physical

interaction. A more specific measurement (like Isothermal titration calorimetry) to quantify the interaction between PHF5A and PHF14 will also support their conclusion.

In addition, such interaction should also be evaluated in the monolayer cultures which show no enrichment of PCSCs.

Author response_Reviewer 2_Comment 1:

We thank the reviewer for these suggestions. In all of our nuclear Co-IP experiments for LC-MS/MS and western blot analyses in this study, an enzymatic shearing cocktail (Active motif, cat. 54001) was used to digest DNA in nuclear protein extracts at 4 °C for 90 minutes which allows a safe release of undissociated protein complexes from DNA (Revised manuscript, Fig. 4a). Data from these experiments verify the physical association between PHF5A and PHF14 in different PCSCs (Revised manuscript, Fig. 2l, m).

As suggested, we have evaluated the physical interaction between PHF5A and PHF14 in monolayer cultures lacking CSC enrichment, including L3.6pl, L3.6sl, HPDE6c7, and the differentiated mouse embryonic fibroblasts NIH-3T3 cells by western blot analysis of PHF5A Co-IPs in DNase-digested nuclear protein extracts from these monolayer cultures. This revealed a physical association between PHF5A and PHF14 in L3.6pl, L3.6sl (Revised manuscript, Supplementary Fig. S2d), and HPDE6c7 (Revised manuscript, Supplementary Fig. S2e) cells, but not in NIH-3T3 cells (Revised manuscript, Supplementary Fig. S2f). These findings indicate that PHF5A and PHF14 are not physically associated in all cell types, and suggest that the physical association between PHF5A and PHF14 is cell type-dependent to regulate specific biological properties and functions.

Revised Supplementary Fig. S2

d-f, Western blot analysis of PHF5A Co-IPs from nuclear protein extracts of L3.6pl, L3.6sl (d), HPDE6c7 (e), and NIH-3T3 (f) monolayer cultures.

We have now added these data to Supplementary Fig. S2 and the Results section (page 9, lines 285-293) of our revised manuscript.

In clinical datasets, we observed a strong correlation between the expression levels of PHF5A and PHF14 in normal pancreas (n = 328) and PDAC patient samples (n = 182) with a correlation coefficient of 0.78 and 0.57, respectively, which suggests potential physical association and cooperation.

Pearson's product-moment correlation analysis between PHF5A and PHF14 in normal pancreas tissue (a; n=328) and PDAC tissues (b; n = 182). Data were obtained from the GTEx and TCGA PAAD projects, respectively. **Abbreviations:** CI, confidence interval.

Reviewer 2_Comment 2:

ChIP-seq for PHF5A and PHF14 resulted in less than 300 targeted regions being identified. It would be important to show quality controls for the specificity of the PHF5A and PHF14 antibodies.

Author response_Reviewer 2_Comment 2:

We thank the reviewer for highlighting this point. Quality control testing for anti-PHF5A (Proteintech, cat. 15554-1-AP) and anti-PHF14 (Proteintech, cat. 24787-1-AP) antibodies was

performed according to the guidelines set by ENCODE and modENCODE consortia for ChIP-seq experiments (Landt, S.G. *et al.* Genome Res. 2012, PMID: 22955991).

I- Western blot analysis:

A- Detection of PHF5A and PHF14 in a fractionated protein extract at their expected molecular weights (kDa).

Western blot **a** demonstrates strong immunoreactivity bands for PHF5A and PHF14 proteins in the nuclear fraction of cell protein lysate from SSEA4⁺ A13A cells. Lamin B1 was used as a nuclear marker. Arrows indicate the bands corresponding to the expected sizes of proteins:

- PHF14 (140-150 kDa).
- PHF5A (12-14 kDa).
- Lamin B1 (66 kDa).

B. Co-immunoprecipitation of PHF5A and PHF14 in nuclear protein extracts.

- Western blot **b** demonstrates Co-IP of PHF5A by anti-PHF5A antibody. Arrow indicates the band corresponding to the expected size of PHF5A protein (12-14 kDa) in the nuclear input and anti-PHF5A Co-IP eluate of SSEA4⁺ A13A cells.

Note: *Light chain of the control IgG and PHF5A antibodies was detected at 25 kDa.

b

- Western blot **c** demonstrates Co-IP of PHF14 by anti-PHF14 antibody. Arrow indicates the band corresponding to the expected size of PHF14 protein (140-150 kDa) in the nuclear input and anti-PHF14 Co-IP eluate of ABCG2⁺ L3.6pl cells.

c

Similar results were obtained in separate Co-IP experiments which indicate the specificity of the antibodies compared to the presence of background bands.

II- ChIP-PCR and ChIP-qPCR analyses:

In addition to western blotting, we have specifically addressed ChIP quality of the antibodies by investigating the *FLT4* locus as a positive control of PHF5A-PHF14-KMT2A binding by performing PHF5A ChIP, PHF14 ChIP, KMT2A ChIP, and IgG ChIP in the proximity of the *FLT4* locus, and performing PCR of the samples for 30 cycles followed by 1% agarose gel electrophoresis. We also used PHF5A knockout (KO) cells for PHF5A ChIP that we derived by CRISPR/Cas9-mediated KO editing. We observed that PHF5A, PHF14, and KMT2A are all bound to the *FLT4* locus compared to IgG ChIP (enrichment > 100-fold), as demonstrated by the appearance of amplified DNA bands with the expected size upon 1% agarose gel electrophoresis which indicates the specificity of the antibody pulldown (example of gel in left panel). Furthermore, the PHF5A KO cells did not show enrichment of the *FLT4* locus, indicating that the PHF5A antibody specifically pulls down PHF5A on the *FLT4* locus (quantification of DNA bands in right panel).

III- Motif enrichment analysis

We have performed a motif enrichment analysis following the guidelines set by the ENCODE (Landt, S.G., *et al.* Genome Res. 2012, PMID: 22955991). A set of high-quality peaks (IDR threshold = 0.01) was defined between replicates. Then, motif enrichment analysis (HOMER de novo and known enrichment) was performed using HOMER (Hypergeometric Optimization of Motif EnRichment) (version 20201202) using default parameters (Heinz, S., *et al.* Mol Cell. 2010, PMID: 20513432). In the tables shown below for Homer Known Motif Enrichment Results for PHF5A (**a**) and PHF14 (**b**) ChIP experiments, we observed target sequences that contain known motifs of PHF5A (**Table a**) and PHF14 (**Table b**) that meet ENCODE guidelines of >10% motif representation and exceeding the 4% enrichment standard.

a

Homer Known Motif Enrichment Results (PHF5A/)

Homer de novo Motif Results

Gene Ontology Enrichment Results

Known Motif Enrichment Results (txt file)

Total Target Sequences = 85, Total Background Sequences = 23441

Rank	Motif	Name	P-value	log P-value	q-value (Benjamini)	# Target Sequences with Motif	% of Targets Sequences with Motif	# Background Sequences with Motif	% of Background Sequences with Motif	Motif File	SVG
1		BPC6(BBRBPC)/col-BPC6-DAP-Seq(GSE60143)/Homer	1e-26	-6.214e+01	0.0000	43.0	50.59%	1589.3	6.78%	motif file (matrix)	svg
2		bZIP-IRF(bZIP,IRF)/Th17-BatF-ChIP-Seq(GSE39756)/Homer	1e-18	-4.314e+01	0.0000	33.0	38.82%	1336.2	5.70%	motif file (matrix)	svg
3		FRS9(ND)/col-FRS9-DAP-Seq(GSE60143)/Homer	1e-16	-3.843e+01	0.0000	49.0	57.65%	3934.1	16.78%	motif file (matrix)	svg
4		GAGA-repeat/Arabidopsis-Promoters/Homer	1e-5	-1.315e+01	0.0005	55.0	64.71%	9212.0	39.30%	motif file (matrix)	svg
5		BPC1(BBRBPC)/colamp-BPC1-DAP-Seq(GSE60143)/Homer	1e-5	-1.284e+01	0.0005	37.0	43.53%	4933.0	21.05%	motif file (matrix)	svg
6		GFX(?)/Promoter/Homer	1e-2	-5.489e+00	0.6928	2.0	2.35%	26.7	0.11%	motif file (matrix)	svg

b**Homer Known Motif Enrichment Results (PHF14/)**Homer de novo Motif ResultsGene Ontology Enrichment ResultsKnown Motif Enrichment Results (txt file)

Total Target Sequences = 85, Total Background Sequences = 23139

Rank	Motif	Name	P-value	log P-value	q-value (Benjamini)	# Target Sequences with Motif	% of Targets Sequences with Motif	# Background Sequences with Motif	% of Background Sequences with Motif	Motif File	SVG
1		BPC6(BBRBPC)/col-BPC6-DAP-Seq(GSE60143)/Homer	1e-28	-6.629e+01	0.0000	45.0	52.94%	1615.1	6.98%	motif file (matrix)	svg
2		bZIP:IRF(bZIP,IRF)/Th17-BatF-ChIP-Seq(GSE39756)/Homer	1e-18	-4.150e+01	0.0000	33.0	38.82%	1393.7	6.03%	motif file (matrix)	svg
3		FRS9(ND)/col-FRS9-DAP-Seq(GSE60143)/Homer	1e-17	-4.071e+01	0.0000	51.0	60.00%	4029.3	17.42%	motif file (matrix)	svg
4		GAGA-repeat/Arabidopsis-Promoters/Homer	1e-6	-1.512e+01	0.0001	58.0	68.24%	9416.4	40.71%	motif file (matrix)	svg
5		BPC1(BBRBPC)/colamp-BPC1-DAP-Seq(GSE60143)/Homer	1e-5	-1.369e+01	0.0002	40.0	47.06%	5349.6	23.13%	motif file (matrix)	svg
6		E2F7(E2F)/Hela-E2F7-ChIP-Seq(GSE32673)/Homer	1e-2	-4.987e+00	1.0000	3.0	3.53%	104.0	0.45%	motif file (matrix)	svg
7		GFX(?)/Promoter/Homer	1e-2	-4.784e+00	1.0000	2.0	2.35%	37.3	0.16%	motif file (matrix)	svg

Reviewer 2_Comment 3:

It is well demonstrated that KMT2A mediates H3K4me3 and actively associates with RNA polymerase II. It is therefore expected to find that inhibition of KMT2A with either OICR-9429 or MM-102 can lead to decreasing tumorsphere formation and increasing cell death. The authors make the argument that a physical association between PHF5A-PHF14-HMG20A-RAI1 protein subcomplex and KMT2A, in which KMT2A enzyme catalyzes methylated histone marks recognized by PHF5A-PHF14-HMG20A-RAI1 protein subcomplex members. This claim is not supported by examining changes in PHF5A/PHF14 deposition upon MM-102 treatment (Supplementary figure 3e).

Author response_Reviewer 2_Comment 3:

Thanks to the reviewer for allowing us to discuss this point further. There are two possible mechanisms of how KMT2A's enzymatic activity relates to PHF5A-PHF14-HMG20A-RAI1 protein subcomplex: firstly, as we have proposed, KMT2A enzyme catalyzes methylated histone marks that are recognized by PHF5A-PHF14-HMG20A-RAI1 protein subcomplex members, which would impact their chromatin binding. In light of this, PHF5A and PHF14 are two histone readers which have been reported to interact with histone H3 (Zheng, Y.Z. *et al.* Cancer Res. 2018, PMID 29700004; Huang, Q. *et al.* Acta Biochim Biophys. 2013, PMID 23688586). Our biotin-modified histone peptide pull-down experiments (Revised manuscript, Supplementary Fig. S3d) demonstrate the binding affinity of PHF5A and

PHF14 for di/tri-methylated lysine 4 of histone H3 in ABCG2⁺ L3.6sl tumorspheres (Revised manuscript, Supplementary Fig. S3e). The inhibition of the HMT activity of KMT2A in PCSCs using MM-102, a peptidomimetic antagonist of KMT2A-WDR5 interaction, decreases H3K4me3 levels in a concentration-dependent manner, with no changes observed in H3K4me2 levels (Revised manuscript, Fig. 6c). In this context, it is of note that reduced H3K4me3 levels achieved at a concentration of 75 μ M of MM-102 were associated with impaired self-renewal and induction of apoptosis in PCSCs, despite unchanged levels of H3K4me2, suggesting that KMT2A epigenetically regulates stem cell phenotypes and functions in PCSCs mainly through the deposition of H3K4me3 marks. To validate our hypothesis, we treated PCSCs with MM-102 at a concentration of 50 μ M. This concentration is not associated with an apoptotic activity, which is critical for ensuring the quality of sequencing data. A significant differential peak enrichment for PHF14 was observed in only 7% (12/171) of genes cooccupied by KMT2A, PHF5A, and PHF14. This may be due to the partial inhibition of the enzymatic activity of KMT2A at 50 μ M, resulting in only a subtle decrease in H3K4me3 protein levels which may not be sufficient to significantly modulate the binding of PHF14 and/or PHF5A to their target genomic sites or to impair the viability of cells. This limitation may be overcome, however, by the generation of inducible *KMT2A* knockouts which preserve the stemness of PCSCs across multiple passages through enhanced CSC-enriching conditions, as well as developing more potent small molecule drug inhibitors of KMTA activity.

Revised Supplementary Fig. S3

e, Western blot analysis of pull-downs of biotinylated H3K4me2 and H3K4me3 peptides in nuclear protein extracts from ABCG2⁺ L3.6sl tumorspheres.

As a second point, KMT2A may catalyze the methylation of histone marks in complex with PHF5A-PHF14-HMG20A-RAI1. This may be true for most loci, where inhibition of

KMT2A has no significant effect on the stability of the complex or binding of its components to target genomic sites, but impacts other aspects, such as gene expression and cellular functions. There is also a possibility that not all genes identified by ChIP-seq are regulated by the PHF5A subcomplex, especially when potentially associated with enhancer regions.

We have added the above information in a shortened form to the Discussion section (page 21, 22, lines 695-714) of our revised manuscript.

Reviewer 2_Comment 3 (cont.):

Furthermore, the ChIP-seq for KMT2A identifies only 400 targets which is much less than the targets demonstrated in other studies. It would be important to show quality controls for the KMT2A antibody.

Author response_Reviewer 2_Comment 3 (cont.):

We thank the reviewer for highlighting this point. Our KMT2A ChIP-seq data are specific to the CSC population of cells, so the number of targets we have detected might not strictly align with other studies involving a distinct cell type. However, our quality control tests for anti-KMT2A antibody (Proteintech, cat. 29728-1-AP) include the following:

I- Western blot analysis:

The figure below demonstrates data of our western blot analysis in nuclear protein extracts from NIH-3T3 cells using anti-KMT2A antibody (Proteintech, cat. 29728-1-AP). Arrow indicates the band corresponding to the expected size of KMT2A (180 kDa).

Please also see the Author response_Reviewer 2_Comment 2, where we show an enrichment of KMT2A ChIP-PCR on FLT4 locus that is > 100-fold above IgG, which is considered of sufficient quality for identifying protein binding by ChIP-sequencing.

II- Motif enrichment analysis:

We have performed a motif enrichment analysis following guidelines set by the ENCODE (Landt, S.G., *et al.* Genome Res. 2012, PMID: 22955991). A set of high-quality peaks (IDR threshold = 0.01) was defined between replicates. Then, motif enrichment analysis (HOMER de novo and known enrichment) was performed with HOMER (Hypergeometric Optimization of Motif EnRichment) (version 20201202) using default parameters (Heinz, S., *et al.* Mol Cell. 2010, PMID: 20513432). According to the table below for Homer Known Motif Enrichment Results for KMT2A, our data include target sequences, identified by our KMT2A ChIP-seq analysis, containing known motifs of KMT2A that meet ENCODE guidelines of >10% motif representation and exceeding the 4% enrichment standard.

Homer Known Motif Enrichment Results (./)

Homer de novo Motif Results

Gene Ontology Enrichment Results

Known Motif Enrichment Results (txt file)

Total Target Sequences = 158, Total Background Sequences = 25628

Rank	Motif	Name	P-value	log P-value	q-value (Benjamini)	# Target Sequences with Motif	% of Targets Sequences with Motif	# Background Sequences with Motif	% of Background Sequences with Motif	Motif File	SVG
1		BPC6(BBRBPC)/col-BPC6-DAP-Seq(GSE60143)/Homer	1e-39	-9.021e+01	0.0000	65.0	41.14%	1391.6	5.43%	motif file (matrix)	svg
2		FRS9(ND)/col-FRS9-DAP-Seq(GSE60143)/Homer	1e-23	-5.311e+01	0.0000	72.0	45.57%	3300.6	12.88%	motif file (matrix)	svg
3		bZIP:IRF(bZIP,IRF)/Th17-BatF-ChIP-Seq(GSE39756)/Homer	1e-18	-4.355e+01	0.0000	47.0	29.75%	1612.8	6.29%	motif file (matrix)	svg
4		BPC1(BBRBPC)/colamp-BPC1-DAP-Seq(GSE60143)/Homer	1e-7	-1.698e+01	0.0000	60.0	37.97%	4958.8	19.35%	motif file (matrix)	svg
5		GAGA-repeat/Arabidopsis-Promoters/Homer	1e-6	-1.405e+01	0.0002	84.0	53.16%	8769.5	34.22%	motif file (matrix)	svg
6		ZML2(C2C2gata)/col-ZML2-DAP-Seq(GSE60143)/Homer	1e-3	-8.395e+00	0.0379	12.0	7.59%	566.3	2.21%	motif file (matrix)	svg
7		KLF10(Zf)/HEK293-KLF10.GFP-ChIP-Seq(GSE58341)/Homer	1e-2	-6.846e+00	0.1529	31.0	19.62%	2826.8	11.03%	motif file (matrix)	svg
8		AT1G04880(ARID)/colamp-AT1G04880-DAP-Seq(GSE60143)/Homer	1e-2	-5.604e+00	0.4630	5.0	3.16%	164.1	0.64%	motif file (matrix)	svg
9		CRF10(AP2EREBP)/col100-CRF10-DAP-Seq(GSE60143)/Homer	1e-2	-5.312e+00	0.5511	8.0	5.06%	422.6	1.65%	motif file (matrix)	svg
10		AT4G12670(MYBrelated)/col-AT4G12670-DAP-Seq(GSE60143)/Homer	1e-2	-5.092e+00	0.6183	1.0	0.63%	0.0	0.00%	motif file (matrix)	svg

Reviewer 2_Minor comment 1:

The authors examine the enrichment of CSC surface markers in pancreatic tumorspheres. Even if ABCG2 shows higher expression levels in tumorsphere cultures, other typical cancer stem cell markers, such as CD24, are expressed much less in the tumorsphere culture than in the monolayer cultures for L3.6pl and L3.6sl lines. The enrichment of ABCG2 could simply be a reflection of the cultural conditions. Therefore, it is important to determine whether the selected ABCG2+ cells are indeed functionally cancer stem cells. For example, whether the ABCG2+ are more enriched in tumor-propagating cells compared with monolayer culture cells in transplantation assays. Do the ABCG2+ cells present higher self-renewal properties compared with monolayer culture cells in serial transplantations?

Author response_Reviewer 2_Minor comment 1:

Thanks to the reviewer for highlighting this point. To address this comment, we have performed the following experiments:

1-Assessment of the self-renewal capacity of ABCG2⁺ cells versus parental monolayer. Sphere-forming assay is one of the reliable methods to identify CSCs and evaluate their self-renewal capacity (Bahmad, H.F. *et al.* Front Oncol. 2018, PMID 30211124). We have performed the requested analysis of the self-renewal of ABCG2⁺ cells versus parental monolayer cells. This revealed a significant increase in the relative number of spheres formed by ABCG2⁺ L3.6pl cells as compared to L3.6pl monolayer cells. Statistical analysis was performed using unpaired two-tailed *t*-test, which indicated statistical significance at p -value < 0.0001.

2- Regarding the other aspects of the comment, in our flow cytometry-based screening of enriched CSC markers in day 6 pancreatic tumorspheres, we observed a significant decrease in

the percentage of cells expressing CD24 in L3.6pl (Revised manuscript, Fig. 1b) and L3.6sl (Revised manuscript, Fig. 1c) tumorspheres cultured under CSC-enriching conditions as compared to their respective parental monolayers, suggesting that CD24 is less likely to mark the CSC population in those cells. PCSCs that are CD44⁺ CD24⁺ EpCAM⁺ have been reported to be self-renewing and highly tumorigenic (Li, C. *et al.* Cancer Res. 2007, PMID 17283135). However, stem cell markers are not universal across all types of tumors, but may vary from one tumor type to another and even within different subpopulations in one tumor type due to the heterogeneity of CSCs (Huang, R. *et al.* Cancer Therapeutics 2017, PMID 27343550). As an example, CD44⁺/CD24⁻ and ALDH1 expression have been identified as the primary markers of breast CSCs (Rabinovich, I. *et al.* Eur. J. Histochem. 2018, PMID 30362671).

On the other hand, our flow cytometry data show significant enrichment of ABCG2 in day 6 L3.6pl (Revised manuscript, Fig. 1b) and L3.6sl (Revised manuscript, Fig. 1c) tumorspheres as compared to their respective parental monolayer cultures. Accordingly, L3.6pl and L3.6sl cells were phenotypically sorted by MACs based on their cell surface expression of ABCG2. ABCG2⁺ cells are also EpCAM⁺ and CD44⁺ (Author response_Reviewer 1_Comment 1).

3- To validate that ABCG2⁺ cells possess stem cell-like properties, we also performed the following:

- i) qRT-PCR analysis of self-renewal and pluripotency-associated genes, including *SOX2*, *NANOG*, *KLF4*, *STAT3*, and *POU5F1* in ABCG2⁺ L3.6pl tumorspheres (Revised manuscript, Fig. 1n) and ABCG2⁺ L3.6sl tumorspheres (Revised manuscript, Fig. 1o) versus HPDE6c7. This revealed a significant increase in the expression of *SOX2* and *NANOG* in ABCG2⁺ L3.6pl (Revised manuscript, Fig. 1n) and ABCG2⁺ L3.6sl (Revised manuscript, Fig. 1o) tumorspheres, in addition to *KLF4* in ABCG2⁺ L3.6sl (Revised manuscript, Fig. 1o) tumorspheres as compared to HPDE6c7.

ii)

n

o

ii) qRT-PCR analysis of self-renewal and pluripotency-associated genes in ABCG2⁺ L3.6pl tumorspheres (Revised manuscript, Supplementary Fig. S1g) and ABCG2⁺ L3.6sl tumorspheres (Revised manuscript, Supplementary Fig. S1h) versus their respective ABCG2⁻ monolayer cells. Here, we observed a significant upregulation of all tested self-renewal and pluripotency genes, including *SOX2*, *NANOG*, *KLF4*, *STAT3*, and *POU5F1* in ABCG2⁺ L3.6pl (Revised manuscript, Supplementary Fig. 1g) and ABCG2⁺ L3.6sl (Revised manuscript, Supplementary Fig. S1h) tumorspheres as compared to their respective ABCG2⁻ monolayer cultures, suggesting that ABCG2 is a marker of the CSC population in PC cells.

g

h

iii) Analysis of SOX2 protein expression in ABCG2⁺ pancreatic tumorspheres versus HPDE6c7 using flow cytometry. This revealed a significant increase in the mean fluorescence intensity (MFI) of SOX2, a readout of SOX2 protein expression levels, in ABCG2⁺ L3.6pl and ABCG2⁺ L3.6sl tumorspheres versus HPDE6c7 (Revised Fig. 1r), as demonstrated by representative flow cytometry histograms of SOX2 protein expression in ABCG2⁺ L3.6pl (Fig. 1s) and ABCG2⁺ L3.6sl (Fig. 1t) tumorspheres versus HPDE6c7.

r

s

t

Revised Fig. 1

r, Mean fluorescence intensity of SOX2 in ABCG2⁺ L3.6pl, ABCG2⁺ L3.6sl, SSEA4⁺ A13A, and SSEA4⁺ A13B tumorspheres as compared to HPDE6c7. Data are representative of 3 independent experiments and shown as the mean value \pm SEM. Statistical analysis was performed using a two-tailed *t*-test with Welch's correction for unequal variances.

s, t, Flow cytometry histograms of intracellular SOX2 protein expression in ABCG2⁺ L3.6pl (**s**) and ABCG2⁺ L3.6sl (**t**) tumorspheres versus HPDE6c7.

We have added these data to Fig. 1r-v, Results section (page 6, 203-212), and the Methods section (page 25, lines 831-834) of our revised manuscript.

Reviewer 2_Minor comment 2:

Specific interaction of KMT2A with PHF5A-PHF14-HMG20A-RAI1 protein subcomplex (Figure h and i) needs to be shown more convincingly. It is unclear if this interaction is direct and specific toward KMT2A in PCSCs. KMT2A is likely to interact with multiple proteins, so it would be expected to see KMT2A interact with PHF5A-PHF14 in general. Co-IPs with KMT2A, are needed to explore the specificity in PCSCs vs. monolayer cultures.

Author response_Reviewer 2_Minor comment 2:

We thank the reviewer for this suggestion. We have performed nuclear KMT2A Co-IP in L3.6pl monolayer and ABCG2⁺ L3.6pl tumorspheres followed by western blot analysis of complex formation. In L3.6pl monolayer, KMT2A only associates with HMG20A and WDR5 (Revised manuscript, Supplementary Fig. S3h). In ABCG2⁺ L3.6pl tumorspheres, however, a physical association between KMT2A, PHF5A, PHF14, HMG20A, RAI1, and WDR5 was observed (Revised manuscript, Supplementary Fig. S3i), indicating the specificity of the protein complex to the CSC population. These results have been added to Supplementary Fig. S3h, i and the Results section (page 13, 14, lines 438-444) of our revised manuscript.

Revised Supplementary Fig. S3

h and **i**, Western blot analyses of KMT2A Co-IPs in nuclear protein extracts from L3.6pl monolayer (**h**) and ABCG2⁺ L3.6pl tumorspheres (**i**).

Reviewer #3, expertise in PCSCs and mass-spec

The manuscript "Identification of an RNA polymerase-associated PHF5A-PHF14-HMG20A-RAI1-KMT2A protein subcomplex as a novel epigenetic target for attenuating the stemness and tumorigenicity of pancreatic cancer stem cells" of M. Abdel Mouti et al. deals with a relevant topic, indeed the epigenetic targeting represent a new emerging area of research. The work is original and obtained results suggest a new potential epigenetic target for the treatment of pancreatic cancer. The methodology used is rigorous and results support the conclusion, however, the paper needs minor revision.

Author response_ Reviewer 3_General comment:

We thank the reviewer for the feedback provided on our work. Likewise, we thank the reviewer for the insightful comments and useful suggestions, which we aimed to address as adequately as possible in the revised manuscript.

Reviewer 3_Comment 1:

Introduction section, it would be useful to mention the differences between PDAC e PASC, since the used in vitro models belong to these PC neoplasms.

Author response_Reviewer 3_Comment 1:

We thank the reviewer for this suggestion. Our revised manuscript now highlights the main differences between PDAC and PASC in the Introduction (page 3, lines 104-110) of our revised manuscript.

Reviewer 3_Comment 2:

Results section, "Small molecule screening ..." paragraph, it is not clear why a different type of cells, i.e. FG cells, were used for the small molecule screening; authors should comment on this

Author response_Reviewer 3_Comment 2:

We thank the reviewer for highlighting this point. FG is the parental PC cell line derived from metastatic pancreatic adenocarcinoma, from which L3.6pl and L3.6sl cells were derived following 3 rounds of selective enrichment in athymic mice (Bruns, C.J. *et al.* Neoplasia 1999, PMID 10935470). This cell line shares the same genetic background of L3.6pl and L3.6sl cells (*KRAS*; Simple; p.Gly12Asp (c.35G>A)). We used FG cells, representing a heterogeneous

population of PC cells, in our small molecule screening to investigate the effects of epigenetic modulators on CSC and non-CSC populations.

We have now added this information to the Results section (page 11, lines 348-353) of our revised manuscript.

Reviewer 3_Comment 3:

Results section, "Optimizing the platform for characterizing the transcriptional ..." the expression of intracellular self-renewal and pluripotency genes in the enriched CSCs (lane 194) should be compared with that of the sorted cells having no CSC surface markers, which represent a better control than HPDE6c7.

Author response_Reviewer 3_Comment 3:

We thank the reviewer for this suggestion. We assessed the expression levels of self-renewal and pluripotency-associated genes in CSC marker-enriched pancreatic sphere-forming cultures versus their parental CSC marker-depleted monolayer cultures. Our data showed a significant increase in the expression of *SOX2*, *NANOG*, *KLF4*, *STAT3*, and *POU5F1* in ABCG2⁺ L3.6pl (Revised manuscript, Supplementary Fig. S1g) and ABCG2⁺ L3.6sl (Revised manuscript, Supplementary Fig. S1h) tumorspheres as compared to their respective ABCG2⁻ depleted (ABCG2⁻) monolayer cells.

Revised Supplementary Fig. S1

g, h, qRT-PCR analysis of expression levels of self-renewal and pluripotency genes in ABCG2⁺ L3.6pl (**g**) and ABCG2⁺ L3.6sl (**h**) tumorspheres as compared to their respective CSC marker-depleted adherent cultures. *ACTB* was used for the normalization of mRNA expression levels. Data are presented as the mean value \pm SEM (n = 6). P values were calculated using a two-tailed *t* test with Welch's correction for unequal variances.

However, we only observed a significant increase in the expression of *POU5F1* in SSEA4⁺ A13A tumorspheres (Revised manuscript, Supplementary Fig. S1i) and *KLF4* in SSEA4⁺ A13B tumorspheres (Revised manuscript, Supplementary Fig. S1j) tumorspheres as compared to their respective SSEA4-depleted (SSEA4⁻) monolayer cells. The elevated expression of some self-renewal and pluripotency-associated genes in SSEA4⁻ monolayer cultures versus SSEA4⁺ tumorspheres of A13A and A13B cells might be linked to their aberrant expression as oncogenic factors in non-CSC monolayer cell cultures (Seymour, T. *et al.* Int. J. Mol. Sci. 2015, PMID 26580604). Several studies have shown that CSCs and non-CSCs coexist in tumors in a dynamic equilibrium and that both types of cells can interconvert in response to environmental factors (Marjanovic, N.D. *et al.* Clin. Chem. 2013, PMID 23220226; Yang, G. *et al.* Br. J. Cancer 2012, PMID 22472879) or upregulation of signaling cascades that promote the acquisition of cancer stem cell-like phenotypes by non-CSCs (Li, H.J. *et al.* Cancer Discov. 2012, PMID 22763855). These could potentially serve as compensatory mechanisms in A13A and A13B cells that are both derived from PDAC tumors, one of the most aggressive and lethal human cancer subtypes, in response to the depletion of the CSC population.

Revised Supplementary Fig. S1

i, j, qRT-PCR analysis of expression levels of self-renewal and pluripotency genes in SSEA4⁺ A13A (**i**) and SSEA4⁺ A13B (**j**) tumorspheres as compared to their respective CSC marker-depleted adherent cultures. *ACTB* was used for the normalization of mRNA expression levels. Data are presented as the mean value \pm SEM (n=6). P values were calculated using a two-tailed *t* test with Welch's correction for unequal variances.

We have added these data to Supplementary Fig. S1g-i and the Results section (page 7, lines 213-221) of our revised manuscript.

Reviewer 3_Comment 4:

Methods section, "Cell Lines and Cell Culture" paragraph, it would be useful to specify which is the genetic background of all the used cell lines, in particular as referred to the typical mutated/deleted genes of pancreatic cancer (i.e. K-ras, p53, p16, DPC4). Moreover, culture of FG cells should be mentioned.

Author response_Reviewer 3_Comment 4:

We thank the reviewer for these suggestions. The genetic backgrounds of cell lines used in this study have been specified in the revised manuscript:

- L3.6pl, L3.6sl, and FG PC cell lines, all with a *KRAS*; Simple; p.Gly12Asp (c.35G>A) genetic background (page 23, lines 761-762).

- The genetic backgrounds for A13A and A13B cell lines are copy number gain of *GATA-6* and *cTAGE1*; *KRAS* G12V; *Tp53* WT; *SMAD4* WT, and copy number gain of *GATA-6* and *cTAGE1*, respectively (page 23, lines 764-766)
- The human pancreatic ductal epithelial cell line HPDE6c7 (normal *KRAS*, *p53*, *c-myc*, and *p16*^{INK4A} genotypes) immortalized by the human papilloma virus (page 24, lines 779-780).

We have also provided a description of the culture conditions for FG cells (page 23, lines 766-771).

Reviewer 3_Comment 5:

Methods section, "LC-MS/MS Analysis" paragraph, is should be described the method used for the normalized total spectral counts

Author response_ Reviewer 3_Comment 5:

We thank the reviewer for this suggestion. We have described the steps involved in normalizing the total spectral counts in the Methods section (page 28, lines 927-930) of our revised manuscript as follows:

- 1) calculation of total number of spectra in each biosample.
- 2) calculation of the average number of spectra across all biosamples.
- 3) multiplying each spectrum count in each biosample by the average count over the biosample's total spectral count.

Reviewer 3_Comment 6:

Methods section, Lane 903: only OICR-9429 is indicated, whilst in into the result section (lane 326) it is mentioned a library of 142 compounds. The library which was used should be indicated into the method.

Author response_ Reviewer 3_Comment 6:

We thank the reviewer for bringing this point to our attention. In our revised manuscript, we have mentioned in the Methods section (page 32, lines 1040-1041) of our revised manuscript that 142 small molecule compounds were added individually to each well, including OICR-9429. We also list the compounds used in the library in Revised Supplementary Table 1.

Reviewer #4, expertise in Small Molecule Chemical Screening

This manuscript, by Mouti et al, describes the identification of a potential sensitivity of pancreatic cancer stem cells by way of an RNA polymerase complex that includes KMT2A. Overall, I found the manuscript interesting and well-written and I believe that it will be of interest to the community.

This submission features a large amount of data used to characterize stem cells derived from pancreatic tumor cell lines and to define epigenetic complexes related to PHF5A that are associated with the stem-like character of these cells. The data presented here are somewhat outside of this reviewer's expertise, so no specific comments on this section will be provided. However, I did find the logic and results of the experiments reasonably clear.

The authors then present a screen of known epigenetic regulators, notably evaluated using several methods in parallel, to identify the WDR5-binding compound OICR-9429 as effective in decreasing expression of a stem cell marker. Additional work in the manuscript using an alternative WDR5-binding compound MM-102 is presented, including an in vivo study. My review of this paper focuses mainly on aspects of the use of the small molecules.

In summary, the identification of a potentially new sensitivity of pancreatic cancer could have a significant impact, given the severity of the disease. However, the choice of a relatively poor-quality molecule to evaluate the potential of this target opportunity somewhat mutes the impact. Additional data would significantly improve the manuscript and add weight to the conclusions of the authors. This data might include some or all of the following:

- *Confirmatory in vitro data with a higher quality small molecule.*
- *Additional biological experiments that correlate with the small molecule results. Off the top of my head, these might include for example mutations that abrogate the histone methylase activity of KMT2A or inhibit its interaction with WDR5.*
- *Target engagement data in vivo to associate the inhibition of tumor growth after dosing with MM-102 with inhibition of KMT2A.*
- *Pharmacokinetic data to indicate that the concentrations of MM-102 achieved with in vivo dosing are sufficiently above the effective KMT2A inhibition concentrations in vitro to correlate with the observed effects.*

Author response_Reviewer 4_General comment:

We appreciate the constructive feedback and valuable remarks provided by the reviewer, which we aimed to address as adequately as possible in our revised manuscript.

Relatively minor issues include the following:

Reviewer 4_Minor comment 1:

Small molecule screening is performed in FG cells (page 9 and methods), but FG is not defined.

Author response_Reviewer 4_Minor comment 1:

We thank the reviewer for bringing this point to our attention. In our revised manuscript, we have defined FG as a human cell line of metastatic pancreatic adenocarcinoma (Morgan, R.T. *et al.* Int. J. Cancer 1980, PMID 6989766), from which spontaneous hepatic metastases (pancreas injection; L3.6pl) and experimental hepatic metastases (spleen injection; L3.6sl) were obtained following three cycles of selective enrichment in nude mice (Bruns, C.J. *et al.* Neoplasia 1999, PMID 10935470).

We have added this information to the Results section (page 5, lines 175-177) of our revised manuscript.

Reviewer 4_Minor comment 2:

No references in the text are not provided for either the OICR or MM molecule (more on this below).

Author response_Reviewer 4_Minor comment 2:

We thank the reviewer for bringing this point to our attention. In our revised manuscript, we have included the following references for each of OICR-9429 (page 36, ref. 21-23) and MM-102 (page 36, Ref. 26, 27):

References for OICR-9429:

21. Nguyen, N. et al. Recruitment of MLL1 complex is essential for SETBP1 to induce myeloid transformation. *iScience* **25**, 103679 (2021).

22. Zhang, J. et al. Targeting WD repeat domain 5 enhances chemosensitivity and inhibits proliferation and programmed death-ligand 1 expression in bladder cancer. *Journal of experimental & clinical cancer research : CR* **40**, 203 (2021).

23. Aho, E.R. et al. Displacement of WDR5 from Chromatin by a WIN Site Inhibitor with Picomolar Affinity. *Cell Rep.* **26**, 2916-2928 (2019).

References for MM-102:

26. Karatas, H, et al. High-affinity, small-molecule peptidomimetic inhibitors of MLL1/WDR5 protein-protein interaction. *J. Am. Chem. Soc.* **135**, 669-82 (2013).

27. Shimoda, H., Doi, S., Nakashima, A., Sasaki, K., Doi, T. & Masaki, T. Inhibition of the H3K4 methyltransferase MLL1/WDR5 complex attenuates renal senescence in ischemia reperfusion mice by reduction of p16^{INK4a}. *Kidney Int.* **96**, 1162-1175 (2019).

Larger issues are noted as follows:

Reviewer 4_Major comment 1a:

The authors should clarify certain mechanistic aspects of how the WDR5 inhibitor function in this context. Ongoing work by the Tansey lab (e.g., 2022 Scientific Reports) suggests that inhibition of histone methylation is not the most relevant driver of the pharmacological effects of WDR5 binding compounds, which would include OICR-9492 and MM-102. Thus, additional support of the role of the WDR5 binding compounds in inhibiting KMT2A leading to the observed phenotypes should be provided:

Unlike KMT2A, WDR5 was not identified in the PHF5A and related complexes. While it is, of course, known that WDR5 inhibitors can affect KMT2A function (see manuscript reference 42, for example), for them to do so in this context, it seems to this reviewer that WDR5 should have been a component of the active complexes.

Author response_Reviewer 4_Major comment 1a:

We thank the reviewer for this suggestion. To address this comment and the comment provided by reviewer 2 regarding the specificity of the PHF5A-PHF14-HMG20A-RAI1-KMT2A protein subcomplex to PCSCs, we have performed KMT2A Co-IP in nuclear protein extracts from L3.6pl monolayer and ABCG2⁺ L3.6pl tumorspheres followed by western blot analysis of complex formation. In L3.6pl monolayer, KMT2A only associates with HMG20A and WDR5 (Revised manuscript, Supplementary Fig. S3h). In ABCG2⁺ L3.6pl tumorspheres, however, a physical association between KMT2A, PHF5A, PHF14, HMG20A, RAI1, and WDR5 was observed (Revised manuscript, Supplementary Fig. S3i), indicating the following:

1. WDR5 is a component of the active subcomplex.
2. PHF5A-PHF14-HMG20A-RAI1-KMT2A-WDR5 protein subcomplex is specific to PCSCs.

Revised Supplementary Fig. S3

h and **i**, Western blot analyses of KMT2A Co-IPs in nuclear protein extracts from L3.6pl monolayer (**h**) and ABCG2⁺ L3.6pl tumorspheres (**i**).

These results have been added to the Results section (page 13, 14, lines 438-444) of our revised manuscript.

Reviewer 4_Major comment 1b:

Some data should be referenced or added to clarify the ability of the compounds to inhibit the HMT action of KMT2A. I am unable to find direct inhibition info on OICR-9429 for example, and no data is presented in the current work to validate the KMT2A inhibition of this compound and its association with the effects in the screen.

Author response_Reviewer 4_Major comment 1b:

We thank the reviewer for this suggestion. We have revised the manuscript to include references for the inhibition of the HMT action of KMT2A by OICR-9429 (page 36, ref. 21-23). OICR-9429 mainly acts by blocking WDR5-KMT2A interaction, which impairs the HMT activity of KMT2A. In our study, treatment of ABCG2⁺ L3.6pl cells with different concentrations of OICR-9429 (0 - 50 μ M) resulted in a decrease in H3K4me3 levels starting at

a concentration of 10 μM , thus validating KMT2A inhibition by OICR-9429. These results have been added to Supplementary Fig. S3a and THE Results section (page 11, lines 366-369) of our revised manuscript.

Revised Supplementary Fig. S3a

Western blot analysis of H3K4me3 levels in ABCG2⁺ L3.6sl cells treated with different concentrations of OICR-9429 for 5 days. Total histone H3 was used as a loading control for extracted histone proteins.

Reviewer 4_Major comment 1b (cont.):

MM-102 is reported (Wang, et al 2013 J.Amer.Chem.Soc) to inhibit KMT2A/MLL1 with an IC₅₀ of 400 nM in biochemical assays. It is notable, based on that 400 nM number for MM-102, that quite high doses of the compound are needed to elicit an effect on histone methylation (Figure 6 b) in this manuscript. What is the source of this drop-off? Many agents display a higher effective dose moving from in vitro data, but 400nm -> 75 μM implies a significant barrier to cellular action (excessive protein binding, cellular permeability, efflux, etc). This is relevant, as it seems that this data drove selection of doses in additional phenotypic experiments. It is also relevant, as these high doses raise the possibility of confounding effects not driven by inhibition of the WDR5-KMT2A interface.

Author response_Reviewer 4_Major comment 1b (cont.):

We thank the reviewer for highlighting this point. The specificity of MM-102 was demonstrated by Karatas et al. using a synthetic analogue of MM-102, known as C-MM-102, that shares its physicochemical properties and structure, with the exception of replacing L-arginine with D-arginine. The synthetic analogue, however, failed to modulate the expression of *HoxA9*, one of KMT2A target genes, at a concentration of 50 μM , suggesting the specificity of the effects induced by MM-102, even at a high drug concentration, to their binding to WDR5 and inhibition of the HMT activity of KMT2A (Karatas, H. et al. J. Am. Chem. Soc. 2013, PMID 23210835).

To avoid confounding effects not driven by inhibition of the WDR5-KMT2A interface, we have performed additional experiments using OICR-9429 and MM-401 which inhibit the enzymatic activity of KMT2A by blocking the physical interaction between KMT2A and WDR5. Our tumorsphere formation assay demonstrated impaired self-renewal capacity of SSEA4⁺ A13A cells when treated with 10 μ M of each of OICR-9429 and MM-401 versus DMSO (control; Supplementary Fig. S3b). We have added these data to Revised Supplementary Fig. S3b and the Results section (page 12, lines 380-382) of our revised manuscript.

Revised Supplementary Fig. S3

b, Column chart demonstrating the effects of OICR-9429 and MM-401 on the sphere-forming capacity of SSEA4⁺ A13A cells as compared to DMSO-treated cells (control). Data are presented as the mean value \pm SEM (n = 6). Statistical analysis was performed using one-way ANOVA with multiple comparisons.

We have also examined the effects of 10 μ M of each of OICR-9429 and MM-401 on the viability of SSEA4⁺ A13A cells which were pretreated with each compound for 5 days before staining with PE-Annexin V and 7-AAD for flow cytometry analysis. This revealed a significant decrease in the percentages of viable and early apoptotic SSEA4⁺ A13A cells treated with either OICR-9429 or MM-401 as compared to DMSO (control; Supplementary Fig. S3c, lower panel) which is demonstrated by the lower percentages of viable (Annexin^{Lo}/7AAD^{Lo}) and early apoptotic (Annexin^{Hi}/7AAD^{Lo}) SSEA4⁺ A13A cells treated with either OICR-929 or

MM-401 versus DMSO in the representative flow cytometry pseudocolor plots (Supplementary Fig. S3c, upper panel). The analysis also showed a significant increase in the percentages of late apoptotic and necrotic SSEA4⁺ A13A cells treated with either OICR-9429 or MM-401 as compared to DMSO (control; Supplementary Fig. S3c, lower panel). These data are demonstrated by the higher percentages of late apoptotic (Annexin^{Hi}/7AAD^{Hi}) and necrotic (Annexin^{Lo}/7AAD^{Hi}) SSEA4⁺ A13A cells treated with either OICR-929 or MM-401 versus DMSO (control) in the representative flow cytometry pseudocolor plots (Supplementary Fig. S3c, upper panel). We have added these data to Supplementary Fig. S3c and the Results section (page 12, lines 382-396).

c

Supplementary Fig. S3

c, upper panel: Representative pseudocolor plots of flow cytometry analysis of cell viability and apoptosis following PE-Annexin V and 7-AAD staining of SSEA4⁺ A13A cells treated with DMSO, OICR-9429 and MM-401 for 5 days. **c, lower panel:** Graphical presentation of the percentages (%) of viable, early apoptotic, necrotic, and late apoptotic, and necrotic SSEA4⁺ A13A cells treated with either DMSO, OICR-9429, or MM-401 for 5 days, as

determined by flow cytometry analysis of PE-Annexin V and 7-AAD-stained SSEA4⁺ A13A cells. Data are presented as the mean value \pm SEM (n = 6). Statistical analysis was performed using two-way ANOVA with multiple comparisons.

In addition, treatment of PCSCs with increasing concentrations of MM-102 inhibit the HMT activity of KMT2A in a dose-dependent manner, as demonstrated by the gradual decrease in H3K4me3 levels with increasing drug concentrations (Revised manuscript, Fig. 6c). Based on the data obtained from our *in vitro* functional assays in PCSCs, the observed phenotypes, including reduced tumorsphere formation and induction of apoptosis, are correlated with lower levels of H3K4me3 achieved at 75 μ M of MM-102, suggesting a biochemical dependence on the maintenance of H3K4me3 levels for cell survival. Therefore, our findings do not support the hypothesis that the confounding effects of MM-102 are not related to the inhibition of the WDR5-KMT2A interface at high doses, as we observed an impairment of the HMT activity of KMT2A, which is dependent on its association with WDR5, as demonstrated by the decrease in H3K4me3 protein levels.

In terms of concentration, others have also demonstrated impaired KMT2A activity and subsequent reduction in H3K4me3 levels only at high concentrations (up to 50 μ M) of MM-102 (Shimoda, H. et al. *Kidney international* 2019, PMID 31570196; Furth et al. *Cell Reports* 2022, PMID 35584667). We believe that the concentration of the drug inhibitor is influenced by several factors, including the stability of the physical association between KMT2A and WDR5, access of the drug to its target binding site, and expression levels of WDR5 (Karatas, H. *et al.* *J. Am. Chem. Soc.* 2013, PMID 23210835), which consequently result in the observed discrepancies between reported IC₅₀ values in different cell types.

Reviewer 4_Major comment 1c:

The use of complementary in vitro data with a superior molecule compared with MM-102 is highly encouraged. OICR-9429 is a well-regarded probe for in vitro experiments regarding WDR5 (see chemicalprobes.org), but I could only find more limited information on MM-102. References or data should be supplied to support the specificity of the MM-102 compound.

Author response_Reviewer 4_Major comment 1c:

We thank the reviewer for these suggestions. We provide here below a point-by-point response to different aspects of the comment.

1. Complementary in vitro data with a superior drug molecule and references to support the specificity of MM-102: Please check our Author response_Reviewer 4_Major comment 1b (cont.).
2. References for MM-102 compound: We have included references for the MM-102 compound in our revised manuscript (page 36, Ref. 26, 27).
3. Data supporting the specificity of MM-102: Our western blot analysis shows that the inhibition of the HMT activity of KMT2A using MM-102 decreases H3K4me3 levels in a concentration-dependent manner, without any changes observed in H3K4me2 levels even at high concentrations of MM-102 (Revised manuscript, Fig. 6c), suggesting its specificity to the inhibition of the H3K4me3 writing activity of KMT2A.

Reviewer 4_Major comment 1c (cont.):

Additionally, repeating some of the in vitro work with the OICR compound or perhaps the more completely characterized MM-401 compound from reference 42 would lend weight to the role of a WDR5 inhibitor in this work. Without such data, the prospect of additional off-target effects to contribute to the action of MM-102 at the high doses used has to be considered.

Author response_Reviewer 4_Major comment 1c (cont.):

We thank the reviewer for this suggestion. In our revised manuscript, we have performed additional in vitro experiments with OICR-9429 and MM-401 to validate the role of WDR5 inhibitors in this work. Please check our Author response_Reviewer 4_Major comment 1b (cont.)

Reviewer 4_Major Comment 1d:

This reviewer notes that a direct inhibitor of the enzymatic action of KMT2A, if such is available, could be used to further differentiate the potential mechanisms (e.g., the HMT activity vs potential scaffolding functions) and add clarity.

Author response_Reviewer 4_Major comment 1d:

We thank the reviewer for this suggestion. We agree with the reviewer that a direct KMT2A inhibitor would be useful for further clarifying the potential mechanisms involved in the process. To the best of our knowledge, there is no direct KMT2A enzymatic inhibitor currently available. This may be due to the relatively low HMT activity of KMT2A which is greatly enhanced by interacting with WDR5. Hence, inhibiting the enzymatic activity of KMT2A mainly involves blocking the interaction between WDR5 and KMT2A (Dou. Y. et al. *Nat. Struct. Mol. Biol.* 2006, PMID 16878130).

Reviewer 4_Major comment 2a:

No pharmacokinetic data are presented or referenced to support the use of MM-102 in vivo. Since relatively high concentrations (50-75 μ M) appear to be needed in cellular experiments, do the exposures obtained in the animal studies approach these levels? Given the lack of difference in 30 and 50 mg/kg/day doses of the molecule (Figure 6l), some context with respect to drug concentrations achieved in blood or tumors is needed.

Author response_ Reviewer 4_Major comment 2a:

We thank the reviewer for this comment. The *in vivo* effects of MM-102 on renal senescence, inflammation, and fibrosis have been tested in mice with ischemia-reperfusion injury at a dose of 15 mg/kg per day for 7 consecutive days (105 mg/kg per week) (Shimoda, H. et al. *Kidney international* 2019, PMID 31770196). However, this study did not provide the drug's pharmacokinetic data *in vivo*. Based on the relatively high concentration of MM-102 (75 μ M) required *in vitro* to achieve KMT2A inhibition and significant functional effects, we performed a pilot study to determine the maximal tolerated dose (MTD) of MM-102 in mice using the following doses: 30, 50, 75, and 100 mg/kg/day every 3 days, in which we have determined 50 mg/kg/day as the MTD. Our study relied on the reduction in tumor volume which was routinely monitored in MM-102 treated mice as compared to vehicle-treated ones as a measure of whether the administered doses are sufficient to induce a therapeutic response, rather than measuring drug concentrations in blood or tumors. It is important, however, to take into account *in vivo* drug concentrations when planning our follow-up studies.

Reviewer 4_Major comment 2ai:

Indeed, this reviewer suggests that an alternative inhibitor molecule might be used for further follow-ups for in vivo (note that this may not be necessary for the current paper). OICR-9429 appears to be ill-suited for in vivo work (chemicalprobes.org). However, additional WDR5 inhibitors have been reported, at least some of which feature defined pharmacokinetics and even oral availability (e.g., Fesik, et al 2022 J. Medicinal Chem.).

Author response_Reviewer 4_Major Comment 2ai:

We thank the reviewer for this suggestion. We agree with the reviewer on using more potent WDR5 inhibitors for our follow-up in *in vivo* experiments in the future.

Reviewer 4_Major comment 2b:

Assessing the level of inhibition of KMT2A activity achieved in the tumors would better associate the role of KMT2A with the in vivo phenotypic observations.

Author response_Reviewer 4_Major comment 2b:

We thank the reviewer for this suggestion. We agree with the reviewer that such experiment would be useful. We have performed western blotting experiments for investigating the H3K4me3 signal in fractions of subcutaneous tumors resected from xenografted mice treated with MM-102 (50 mg/kg/day) versus vehicle (control). As shown in the figure below, our data revealed a decrease in H3K4me3 protein levels in MM-102-treated tumors (especially tumors 2 and 6) as compared to vehicle -treated tumors, which confirmed the effects of KMT2A inhibition to a comparable level as we observed in our *in vitro* studies.

REVIEWER COMMENTS

Reviewer #1 (Remarks to the Author):

Authors performed several experiments to to prove the raised questions. They have performed different CSC population analysis, serial dilution assay, and FACS assay for different CSC markers. The manuscript is extensively revised with all the new data. They have also changed the results and discussion. The present format of the revised manuscript is acceptable for the publication.

Reviewer #2 (Remarks to the Author):

The authors have partially addressed my concerns. However, (1) their new data did not support the conclusion that PHF14 is a novel binding partner of PHF5A in PCSCs. The authors performed co-IP experiments in monolayer cultures lacking CSC enrichment, including L3.6pl, L3.6sl, HPDE6c7, and the differentiated mouse embryonic fibroblasts NIH-3T3 cells. They found a physical association between PHF5A and PHF14 in pancreatic cancer cells L3.6pl, L3.6sl, as well as HPDE6c7 but not in the NIH-3T3 cells. These findings clearly indicate the physical association between PHF5A and PHF14 is non PCSC-specific; (2) The western blot and ChIP-qPCR experiments were not sufficient to show the reliability of PHF5A and PHF14 ChIP-seq. It is expecting to identify the same targets from ChIP-qPCR with the same antibodies they used for ChIP-seq. Their western blot analyses could only demonstrate that the antibodies have specific binding for proteins between size 117-171 kDa, and 10-17 kDa. The authors need to perform ChIP-qPCR or ChIP-seq in PHF5A- or PHF14- knockout cells to illustrate the specificity.

Reviewer #3 (Remarks to the Author):

The authors completely answered to the comments. After revision, the manuscript was much improved.

In my opinion now the manuscript can be accepted for publication.

Reviewer #5 (Remarks to the Author):

Thank you for the significant effort you put into the revised version of the document. It is a very interesting piece of carefully done research.

Did the authors consider adding to the manuscript (i) a comment that summarises Reviewer 4's concerns about the dose of MM-102 used in vivo and (ii) a statement that

ideally this work would be repeated using an improved chemical tool in the future? This response could be based on the reasonable justifications contained in the response to reviewers on these issues.

Manuscript number: NCOMMS-22-46626B

A point-by-point response to Reviewers' comments

Reviewer #1 (Remarks to the Author):

Authors performed several experiments to prove the raised questions. They have performed different CSC population analysis, serial dilution assay, and FACS assay for different CSC markers. The manuscript is extensively revised with all the new data. They have also changed the results and discussion. The present format of the revised manuscript is acceptable for the publication.

Author response_Reviewer 1_General Comment:

We thank the reviewer for the positive response.

Reviewer #2 (Remarks to the Author):

The authors have partially addressed my concerns. However,

Reviewer 2_Comment 1:

Their new data did not support the conclusion that PHF14 is a novel binding partner of PHF5A in PCSCs. The authors performed co-IP experiments in monolayer cultures lacking CSC enrichment, including L3.6pl, L3.6sl, HPDE6c7, and the differentiated mouse embryonic fibroblasts NIH-3T3 cells. They found a physical association between PHF5A and PHF14 in pancreatic cancer cells L3.6pl, L3.6sl, as well as HPDE6c7 but not in the NIH-3T3 cells. These findings clearly indicate the physical association between PHF5A and PHF14 is non PCSC-specific.

Author response_Reviewer 2_Comment 1:

We thank the reviewer for this comment. Our LC-MS/MS analysis of PHF5A nuclear Co-IPs in PCSCs identified PHF14 as one of the top 10 nuclear binding partners of PHF5A based on normalized total spectral counts (Figure 2h). Western blot analysis of nuclease-treated, nuclear Co-IPs of PHF5A (Figure 2i) and PHF14 (Figure 2j) in different PCSC populations confirmed the physical association between PHF5A and PHF14 which was previously unrecognized. Furthermore, our simulation of the molecular docking between PHF5A and PHF14 revealed a favored physical association between the two proteins, with a putative binding energy of -288.23 kcal/mol (Supplementary Figure S2a) which validates our experimental findings.

The physical association between PHF5A and PHF14 is not specific to the CSC population, as it was detected in monolayer cultures lacking CSC enrichment, including L3.6pl, L3.6sl (Supplementary Fig. S2d) and HPDE6c7 (Supplementary Figure S2e). In NIH-3T3 cells, however, PHF5A was not found to bind to PHF14 (Supplementary Figure S2f), suggesting that the physical association between the two proteins is cell type-dependent to regulate specific biological properties and functions.

To conclude, we agree with the reviewer and have added this information to the revised manuscript (page 8, 9, lines 295-304).

“The physical association between PHF5A and PHF14 was then examined in monolayer cultures lacking CSC enrichment, including L3.6pl, L3.6sl, HPDE6c7, and the

differentiated mouse embryonic fibroblasts NIH-3T3 by western blot analysis of nuclear PHF5A Co-IPs in those cells (Supplementary Fig. S2c). As a result, we observed that PHF5A physically associates with PHF14 in monolayer cultures of L3.6pl, L3.6sl (Supplementary Fig. S2d), and HPDE6c7 (Supplementary Fig. S2e). According to these findings, we conclude that the physical association between PHF5A and PHF14 is not specific to PCSCs. In NIH-3T3 cells, however, PHF5A does not bind to PHF14 (Supplementary Fig. S2f), suggesting that the physical association between the two proteins is cell type-dependent to regulate specific cellular properties and functions.”

Additionally, we would like to point out that we assessed the specificity of the identified PHF5A-PHF14-HMG20A-RAI1-KMT2A protein subcomplex to PCSCs by western blot analysis of nuclear KMT2A Co-IPs in L3.6pl monolayer and ABCG2⁺ L3.6pl tumorspheres. In L3.6pl monolayer cells, KMT2A was found to only associate with HMG20A and WDR5 (Supplementary Figure S3h). In ABCG2⁺ L3.6pl tumorspheres, however, we observed a physical association between KMT2A, PHF5A, PHF14, HMG20A, RAI1, and WDR5 (Supplementary Figure S3i), indicating the specificity of the identified protein subcomplex to PCSCs.

Reviewer 2_Comment 2:

(2) The western blot and ChIP-qPCR experiments were not sufficient to show the reliability of PHF5A and PHF14 ChIP-seq. It is expecting to identify the same targets from ChIP-qPCR with the same antibodies they used for ChIP-seq. Their western blot analyses could only demonstrate that the antibodies have specific binding for proteins between size 117-171 kDa, and 10-17 kDa. The authors need to perform ChIP-qPCR or ChIP-seq in PHF5A- or PHF14- knockout cells to illustrate the specificity.

Author response_Reviewer 2_Comment 2:

We thank the reviewer for this comment. Please find the requested experimental data in PHF5A knockout cells below.

Firstly, we would like to emphasize that we identified the same targets from ChIP-qPCR with the same antibodies used for ChIP-seq analysis which included anti-PHF5A (Proteintech, cat.

15554-1-AP) and anti-PHF14 (Proteintech, cat. 24787-1-AP) antibodies. This indicates the specific binding of these different proteins to the same genomic regions (e.g. *FLT4* locus).

In our revision, we performed PCR and qPCR analyses of PHF5A ChIP in PHF5A-knockout cells that we derived by CRISPR/Cas9-mediated KO editing, as well as PHF5A ChIP, PHF14 ChIP, KMT2A ChIP, and IgG ChIP in control (non-knockout) cells in the proximity of the *FLT4* locus that we have identified by our PHF5A, PHF14, and KMT2A ChIP-seq analyses in PCSCs. Our ChIP-PCR data show that PHF5A, PHF14, and KMT2A are all bound to the same *FLT4* locus compared to IgG ChIP (enrichment > 100-fold), as demonstrated by the appearance of amplified DNA bands with the expected size upon 1% agarose gel electrophoresis (left panel). However, PHF5A knock-out cells did not show enrichment at the *FLT4* locus, indicating that the PHF5A antibody specifically pulls down PHF5A on the *FLT4* locus. These findings were further supported by our ChIP-qPCR analysis of enrichment at the same *FLT4* locus in control (IgG) and PHF5A ChIP in PHF5A knockout cells, in addition to PHF5A ChIP, PHF14 ChIP, KMT2A ChIP, and IgG ChIP in control cells (right panel). ChIP-qPCR data are presented as the mean value \pm S.D. (n = 3). Statistical analysis was performed by one-way ANOVA with multiple comparisons with Tukey correction. **** marks adjusted P-value < 0.0001, *** is adjusted P-value < 0.001, ** is adjusted P-value < 0.01, and * is adjusted P-value < 0.05.

Reviewer #3 (Remarks to the Author):

The authors completely answered to the comments. After revision, the manuscript was much improved. In my opinion now the manuscript can be accepted for publication.

Author response_Reviewer 3_General Comment:

We thank the reviewer for this positive response.

Reviewer #5 (Remarks to the Author):

Thank you for the significant effort you put into the revised version of the document. It is a very interesting piece of carefully done research.

Author response_Reviewer 5_ General comment:

We thank the reviewer for recognizing our work. We also appreciate the reviewer's comments which we fully addressed in the revised version of our manuscript.

Reviewer 5_ Comment i:

Did the authors consider adding to the manuscript a comment that summarises Reviewer 4's concerns about the dose of MM-102 used in vivo

Author response_Reviewer 5_ Comment i:

We thank the reviewer for this comment. In our revised manuscript, we have included additional information that addresses the concerns of Reviewer 4 regarding the doses of MM-102 used in our *in vivo* experiment (page 15, lines 527-531; page 20, lines 707-709).

Page 15, lines 527-531:

“First, we performed a pilot study to determine the maximal tolerated dose (MTD) of MM-102 in mice (n = 3) using the following doses: 30, 50, 75, and 100 mg/kg/day. The MTD of MM-102 in mice was 50 mg/kg/day, so we used this dose of MM-102 for drug treatment in vivo. However, to avoid potential toxicity upon prolonged drug administration, we included an additional dose of 30 mg/kg/day.”

Page 20, lines 707-709:

“Future follow-up studies will evaluate the in vivo effects of additional KMT2A inhibitors and compare their pharmacokinetics data by measuring drug concentrations achieved in blood and tumors in order to select the best candidate for clinical evaluation.”

Reviewer 5_ Comment ii:

A statement that ideally this work would be repeated using an improved chemical tool in the future? This response could be based on the reasonable justifications contained in the response to reviewers on these issues.

Author response_Reviewer 5_Comment ii:

We thank the reviewer for this comment. In our revised manuscript, we have stated that future follow-up studies will be conducted to evaluate the *in vivo* effects of additional KMT2A inhibitors and compare their pharmacokinetics by measuring drug concentrations achieved in blood and tumors in order to select the best candidate for clinical evaluation (page 20, lines 707-709).

Page 20, lines 707-709:

“Future follow-up studies will evaluate the in vivo effects of additional KMT2A inhibitors and compare their pharmacokinetics data by measuring drug concentrations achieved in blood and tumors in order to select the best candidate for clinical evaluation.”

REVIEWERS' COMMENTS

Reviewer #2 (Remarks to the Author):

The authors performed ChIP-qPCR experiments in PHF5A- knockout cells to prove the specificity of PHF5A ChIPseq. They have also changed the statement about the specific interaction between PHF5A and PHF14. The revised manuscript is acceptable for publication.

Author response_Reviewer 2_General Comment:

We thank the reviewer for this positive response.